# From Individual Calibration to Reliable Classifiers:
# ALD Parameterization with mPAIC Guarantees

**Deming Sheng** [1]   **Ricardo Henao** [1]

## Abstract

Modern neural classifiers can achieve remarkable predictive performance, yet often suffer from *miscalibration*. In this paper, we introduce a unified calibration framework applicable to arbitrary distribution-based classifiers. The proposed calibration objective guarantees a *monotone Probably Approximately Individually Calibrated (mPAIC)* predictor, which theoretically implies the properties of a *Probably Approximately Calibrated Classifier (PACC)* with explicit error bounds. To enable stable and effective optimization, we further devise a *Decoupled Dual-Stream Optimization (DDSO)* strategy with gradient detachment to reconcile discriminative representation learning and continuous calibration. Notably, our framework bridges calibration paradigms, supporting flexible deployment either as an end-to-end *pre-calibration* objective or as a lightweight *post-calibration* adapter. Extensive experiments across nine real-world datasets demonstrate that our approach consistently outperforms strong baselines, achieving superior performance on both *accuracy* and multi-level *calibration*.

## 1. Introduction

Recent advances in neural networks (He et al., 2016; Devlin et al., 2019; Alexey et al., 2021) have dramatically improved classifier accuracy, facilitating their deployment across diverse fields (Chen et al., 2024; Gong et al., 2025; Liu et al., 2025). Despite these gains, their predicted probabilities are frequently *miscalibrated*, often manifesting as overconfident forecasts (Marx et al., 2023; Vashistha & Farahi, 2025), where predicted probabilities do not align with empirical frequencies (*e.g.*, among predictions made with confidence $0.7$, approximately $70\%$ should be correct).

One potential source of this *miscalibration* lies in the use of the standard Softmax function. From a probabilistic perspective, Softmax can be interpreted as a parameter estimator for a Gumbel distribution (Gumbel, 1935) with a fixed *scale* parameter. This theoretical constraint (McFadden, 1972), imposes a rigid assumption of homoscedasticity, implying that the latent utility distributions across all classes and samples share an identical and constant variance. Consequently, Softmax lacks the necessary degrees of freedom to capture heteroscedasticity (*i.e.*, the varying levels of aleatoric uncertainty inherent in different samples), thereby undermining calibration performance (see Section 2.2 for theoretical analysis and Section 4.4 for empirical results).

To mitigate this issue, various calibration strategies have been proposed, broadly categorized into *post-calibration* and *pre-calibration*. *Post-calibration* methods typically adjust the output probabilities of a trained model using a held-out validation set. A widely adopted technique is Temperature Scaling (TS, Guo et al. 2017), which rescales logits by a single global learnable scalar $T > 0$ before applying the Softmax function. However, TS is fundamentally restricted by its monotonic linear transformation, rendering it incapable of rectifying complex, non-monotonic biases (*e.g.*, simultaneous under- and over-confidence). Furthermore, the single global parameter fails to address the varying patterns of *miscalibration* between different classes. In contrast, Histogram Binning (HB, Zadrozny & Elkan 2001) addresses these limitations by partitioning the prediction space into $M$ disjoint bins. By replacing the predicted probabilities of samples in each bin with the bin's empirical accuracy, HB performs independent interval adjustment, effectively correcting non-monotonic biases. However, HB is critically sensitive to the hyperparameter $M$, which is a value that if it is too small results in coarse-grained adjustments (underfitting), whereas a value that is too large leads to sparsely populated bins with high statistical variance (overfitting).

Beyond *post-calibration*, *pre-calibration* methods incorporate calibration objectives directly into training. A representative direction is distribution matching via differentiable criteria (*e.g.*, MMD-based regularization, Marx et al. 2023), which avoids discretization artifacts and supports end-to-end optimization. While effective, *pre-calibration* methods are typically tied to training and are less flexible at infer-

[1]Duke University. Correspondence to: Ricardo Henao <ricardo.henao@duke.edu>.

*Proceedings of the $43^{rd}$ International Conference on Machine Learning*, Seoul, South Korea. PMLR 306, 2026. Copyright 2026 by the author(s).

ence time: they may require careful tuning of regularization strength, can introduce optimization tension between discrimination and calibration, and cannot be trivially attached to an already-trained model.

To address the above limitations, we first systematically explore alternative distributional classifiers to identify the structural causes of *miscalibration* inherent to Softmax. Based on these insights, we propose the asymmetric Laplace Distribution (ALD, Kotz et al. 2012) classifier ($\mathbf{H}_{\text{ALD}}$) and the corresponding *individualized calibration* with ALD classifier ($\mathbf{H}_{\text{ICALD}}$). Our specific contributions are summarized as follows.

- We introduce a unified calibration framework applicable to arbitrary distribution-based classifiers (*e.g.*, $\mathbf{H}_{\text{ICSoftmax}}$ and $\mathbf{H}_{\text{ICALD}}$). The proposed calibration objective guarantees a *monotone Probably Approximately Individually Calibrated (mPAIC)* predictor, which theoretically implies the properties of a *Probably Approximately Calibrated Classifier (PACC)* with explicit error bounds.

- We also propose a *Decoupled Dual-Stream Optimization (DDSO)* strategy that uses gradient detachment to reconcile the conflicting objectives of *discrimination* and *calibration*. By preventing feature collapse and synchronizing convergence rates, our approach improves *calibration* without compromising *discriminative accuracy*. Crucially, this framework bridges the gap between paradigms, allowing for flexible use as either an end-to-end *pre-calibration* objective or a lightweight *post-calibration* adapter.

- We conduct extensive experiments on nine real-world datasets using seven performance metrics, spanning both *accuracy* and multi-level *calibration* assessments. Our method consistently demonstrates superior performance and robustness compared to the other *pre-* and *post-calibration* baselines.

## 2. Methods

### 2.1. Notation

We use capital letters $X$, $Y$ and $Q$ for random variables, lowercase $y$ and $q$ for fixed values, bold lowercase $\mathbf{x}$ for vectors and $\mathcal{X}$, $\mathcal{Y}$ and $\mathcal{Q}$ for their domains. We denote a classification model by $\mathbf{H}$. Specifically, a Softmax-based classifier is $\mathbf{H}_{\text{Softmax}}$, a classifier based on the ALD is $\mathbf{H}_{\text{ALD}}$.

### 2.2. Softmax and ALD Classifiers

Given a classification dataset $\mathcal{D} = \{(\mathbf{x}_n, y_n)\}_{n=1}^{N}$ composed of $N$ samples, where $\mathbf{x}_n$ represents input features (*e.g.*, tabular data, text, or images) and $y_n \in \{1, \ldots, C\}$ denotes the class label among $C$ classes, the dominant approach employs a Softmax-based classifier, denoted $\mathbf{H}_{\text{Softmax}}$, to predict the class probability distribution. Although typically interpreted as a normalization

step, $\mathbf{H}_{\text{Softmax}}$ can be rigorously formulated as a distribution-based classifier under the Random Utility Theory (McFadden, 1972), in the sense that it assumes a latent utility $U_c$ for each class $c$:

$$U_c = z_c + \epsilon_c, \quad \epsilon_c \sim \text{Gumbel}(\mu = 0, \beta = 1), \quad (1)$$

where $z_c$ is the logit output from a neural network (or linear model) and $\epsilon_c$ is an independent and identically distributed (*i.i.d.*) random noise term that captures unobserved stochasticity. In this context, $\mathbf{H}_{\text{Softmax}}$ essentially functions as a parameter estimator that predicts the *location* parameter ($\mu = z_c$) of a Gumbel distribution (Gumbel, 1935), while implicitly fixing its *scale* parameter ($\beta = 1$). Consequently, the predicted probability $\Pr(y = c)$ corresponds to the probability that the $c$-th latent variable achieves the maximum utility, which yields the standard Softmax function (see Appendix A.1 for the detailed proof).

This perspective highlights a critical limitation: by fixing its scale parameter, $\mathbf{H}_{\text{Softmax}}$ lacks the flexibility to model sample-specific uncertainty (see Appendix C.5 for a case study). However, the probabilistic interpretation of $\mathbf{H}_{\text{Softmax}}$ shows a promising direction: the potential to explicitly predict the distribution parameters for classification tasks. Naturally, a fundamental question arises: *Can we design a superior classifier grounded in a more flexible distribution?* Driven by this motivation, we propose the asymmetric Laplace Distribution (ALD, Kotz et al. 2012) classifier, $\mathbf{H}_{\text{ALD}}$. To validate this choice, we provide a comprehensive comparison with other candidate distributions (*e.g.*, Normal, Logistic, Exponential, LogNormal, and Weibull) in Appendix C.1, where $\mathbf{H}_{\text{ALD}}$ demonstrates superior adaptability. Specifically, this advantage arises from the ALD's flexible parameterization, which more effectively captures both heteroscedasticity and the asymmetry (skewness) of class distributions.

Formally, $\mathbf{H}_{\text{ALD}}$ operates by estimating class-wise cumulative probabilities at a fixed threshold and normalizing them to define the categorical posterior. Given an input $\mathbf{x}$, the neural network predicts a set of ALD parameters $\{\theta_c(\mathbf{x}), \sigma_c(\mathbf{x}), \kappa_c(\mathbf{x})\}$ for each class $c \in \{1, \ldots, C\}$. The unnormalized score $s_c(\mathbf{x})$ is defined as the value of the cumulative distribution function (CDF) of the ALD evaluated at a fixed scalar threshold $t_0 \in \mathbb{R}$ (*e.g.*, $t_0 = 0$; see Appendix C.1 for a case study on the choice of $t_0$):

$$s_c(\mathbf{x}) = F_{\text{ALD}}\big(t_0; \theta_c(\mathbf{x}), \sigma_c(\mathbf{x}), \kappa_c(\mathbf{x})\big). \quad (2)$$

Notably, we treat $t_0$ as a fixed hyperparameter rather than as a learnable parameter to ensure model *identifiability*. Since the CDF of the ALD relies primarily on the relative distance between the threshold and the location parameter (*i.e.*, $t_0 - \theta_c(\mathbf{x})$), simultaneously learning both would introduce *translation invariance*, rendering the optimization problem

ill-posed (*i.e.*, infinite pairs of $(t_0, \theta_c(\mathbf{x}))$ could yield the same probability). Alternatively, fixing $t_0$ effectively acts as an anchor in its latent space, which encourages the model to learn meaningful changes in $\theta_c(\mathbf{x})$. Then, these scores are subsequently normalized across all classes to yield the final categorical probability distribution $\hat{\mathbf{p}}(\mathbf{x})$ with elements

$$\hat{p}_c(\mathbf{x}) = \frac{s_c(\mathbf{x})}{\sum_{j=1}^{C} s_j(\mathbf{x})}, \qquad c = 1, \ldots, C. \quad (3)$$

Crucially, this probabilistic framework is not restricted to the ALD. It is distribution-agnostic and can be generalized to any parametric family. In practice, one replaces $F_{\text{ALD}}$ with the CDF of a chosen distribution and predicts its corresponding parameters, while maintaining the rest of the architecture. For example, employing a normal distribution $\mathcal{N}(\mu_c, \eta_c^2)$, involves predicting the mean $\mu_c$ and standard deviation $\eta_c$ for each class and substituting $F_{\mathcal{N}}$ into (2). Complementary experiments and discussion of these variants are provided in Section 4.4 and Appendix C.1.

**Training objective** We adopt the negative log-likelihood (cross-entropy) loss. Let $y_{nc} \in \{0, 1\}$ denote the one-hot encoded ground-truth label for instance $n$ and class $c$, and let $\hat{p}_{nc}$ be the predicted probability for class $c$. The loss is

$$\mathcal{L}_{\text{NLL}} = -\frac{1}{N} \sum_{n=1}^{N} \sum_{c=1}^{C} y_{nc} \log \hat{p}_{nc}(\mathbf{x}_n). \quad (4)$$

### 2.3. ALD-MMD Classifier

The loss in (4) is set to fit labels well; however, predicted probabilities $\hat{p}_c(\mathbf{x})$ can be *miscalibrated*. As noted by Marx et al. (2023), modern neural classifiers often achieve high accuracy while exhibiting poor probability calibration. We first formalize a *Probably Approximately Calibrated Classifier (PACC)*, then introduce a kernel-based calibration penalty that drives a classifier toward satisfying this calibration, and finally combine it with (4).

**Definition 2.1** (PACC). Let $\mathbf{H}$ be a multi-class classifier and let $\widehat{\text{KCE}}[k, \mathbf{H}, \mathcal{D}]$ denote its empirical kernel calibration error computed with kernel $k(\cdot, \cdot)$ on an *i.i.d.* dataset $\mathcal{D} = \{(\mathbf{x}_n, y_n)\}_{n=1}^{N}$. We say that $\mathbf{H}$ is an $(\epsilon, \delta)$-Probably Approximately Calibrated Classifier (PACC) under $k$ if

$$\Pr\left[ \widehat{\text{KCE}}[k, \mathbf{H}, \mathcal{D}] \geq \epsilon \right] \leq \delta.$$

PACC is defined primarily via the KCE (Marx et al., 2023) because histogram-binned metrics such as expected calibration error (ECE, Naeini et al. 2015) introduce discretization bias and are non-differentiable, which impedes end-to-end optimization. In contrast, KCE replaces bins with smooth

kernels, yielding differentiable and low-variance estimators and casting calibration as distribution matching via maximum mean discrepancy (MMD, Gretton et al. 2012). Moreover, the kernel view provides a simple and unified way to measure different notions of calibration by specifying the forecast variable, the target variable, and a conditioning variable $Z$, which recover *canonical*, *top-label* and *marginal* calibration (Vaicenavicius et al., 2019). Conceptually, these notions differ in which aspect of the prediction we require to match the ground truth and what information we condition on: (*i*) *canonical*: $\widehat{Y}, Y, \hat{\mathbf{p}}(X)$; (*ii*) *top-label*: $\mathbb{I}\{\widehat{Y} = Y^*\}, \mathbb{I}\{Y = Y^*\}, \hat{p}_{Y^*}(X)$; (*iii*) *marginal*: $\mathbb{I}\{\widehat{Y} = y\}, \mathbb{I}\{Y = y\}, \hat{p}_y(X)$ for all $y \in \mathcal{Y}$. Here, $X$ is the input of the classifier, $Y$ is the ground-truth label, and $\widehat{Y}$ is a forecast label drawn from the predictive probability mass function $\hat{\mathbf{p}}(X)$ by a classifier. $Y^* = \arg\max_{y \in \mathcal{Y}} \hat{p}_y(X)$ is the top-label and $\hat{p}_{Y^*}(X)$ is its confidence. Finally, $\mathbb{I}\{\cdot\}$ denotes the indicator function.

To make $\mathbf{H}_{\text{ALD}}$ *PACC* at the *canonical* level, we follow Marx et al. (2023) and regularize training through distribution matching between the joint target law $(Y, Z)$ and the joint forecast law $(\widehat{Y}, Z)$, where $Z \equiv \hat{\mathbf{p}}(X)$. Similarly, *top-label PACC* can be obtained by matching the joint laws of $(\mathbb{I}\{Y = Y^*\}, \hat{p}_{Y^*}(X))$ and $(\mathbb{I}\{\widehat{Y} = Y^*\}, \hat{p}_{Y^*}(X))$. Since both pairs share the same conditioning variable $Z$, matching these joint laws provides a tractable surrogate for conditional canonical calibration by encouraging $\Pr(Y|Z) \approx \Pr(\widehat{Y}|Z)$ (Marx et al., 2023). An integral probability metric (IPM, Müller 1997) with witness class $\mathcal{F}$ measures the discrepancy between the two joint laws:

$$D(\mathcal{F}, P, Q) = \sup_{f \in \mathcal{F}} \left| \mathbb{E}_P[f(Y, Z)] - \mathbb{E}_Q[f(\widehat{Y}, Z)] \right|, \quad (5)$$

where $P$ is the joint law of $(Y, Z)$ and $Q$ is that of $(\widehat{Y}, Z)$. Choosing $\mathcal{F}$ as the unit ball of a reproducing kernel Hilbert space (RKHS) $\mathcal{H}_k$, yields the maximum mean discrepancy

$$\text{MMD}(\mathcal{H}_k, P, Q) = \left\| \mathbb{E}_P[\phi(Y, Z)] - \mathbb{E}_Q[\phi(\widehat{Y}, Z)] \right\|_{\mathcal{H}_k}, \quad (6)$$

where $\phi : \mathcal{Y} \times \Delta^{C-1} \to \mathcal{H}_k$ is the RKHS feature map induced by the kernel $k$ and MMD is an IPM where $\mathcal{F}$ is the RKHS unit ball. Following Gretton et al. (2012), the plug-in unbiased estimator of $\text{MMD}^2$ (*i.e.*, squared kernel calibration error, SKCE) with $N$ *i.i.d.* instances in $\mathcal{D}$ is

$$\mathcal{L}_{\text{SMMD}} = \widehat{\text{MMD}}^2(\mathcal{H}_k, \mathcal{D}, Q) = \frac{\sum_{i=1}^{N} \sum_{j \neq i}^{N} h_{ij}}{N(N-1)}, \quad (7)$$

where

$$\begin{aligned} h_{ij} = {} & k\big((y_i, z_i), (y_j, z_j)\big) + k\big((\hat{y}_i, z_i), (\hat{y}_j, z_j)\big) \\ & - k\big((y_i, z_i), (\hat{y}_j, z_j)\big) - k\big((y_j, z_j), (\hat{y}_i, z_i)\big). \end{aligned} \quad (8)$$

and $k(\cdot, \cdot)$ is a factorized kernel on targets and forecasts,

$$k\big((y,z),(y',z')\big) \;=\; k_Y(y,y')\,k_Z(z,z'), \qquad (9)$$

with $k_Y$ a kernel on the finite label space (*e.g.*, $k_Y(y,y') = \mathbb{I}\{y = y'\}$) and $k_Z$ a differentiable kernel on the probability simplex. Since $Z \equiv \hat{\mathbf{p}}(X)$ is the predicted class-probability vector, it lies in $\Delta^{C-1}$ (*i.e.*, it has nonnegative entries that sum to one). Here, we use the radial basis function (RBF, Scholkopf et al. 1997) kernel,

$$k_Z(z,z') = \exp\left(-\tfrac{\|z-z'\|_2^2}{2\tau^2}\right), \qquad \tau > 0, \quad (10)$$

with the bandwidth $\tau$ chosen by the median heuristic (Gretton et al., 2012).

**Training objective** To complete the loss, we regularize the negative log-likelihood with the squared MMD:

$$\mathcal{L}_{\text{ALD-MMD}} \;=\; \mathcal{L}_{\text{NLL}} \;+\; \alpha\,\mathcal{L}_{\text{SMMD}}, \qquad \alpha \geq 0. \quad (11)$$

Minimizing (11) decreases the empirical kernel calibration error ($\widehat{\text{KCE}}$) and thus encourages the *PACC* property, with suitable choices of $\alpha$ and kernel (Marx et al., 2023). Moreover, (11) is distribution-agnostic and applies unchanged to $\mathbf{H}_{\text{Softmax}}$ and other parametric classifiers.

## 2.4. ICALD Classifier

Another route to improve the calibration of $\mathbf{H}_{\text{ALD}}$ is *individualized calibration with ALD (ICALD)*, which we denote by $\mathbf{H}_{\text{ICALD}}$. Unlike $\mathbf{H}_{\text{ALD}}$, $\mathbf{H}_{\text{ICALD}}$ augments its input with a random anchor $q \sim \mathcal{U}(0,1)$, learns ALD parameters as functions of $(\mathbf{x}, q)$ rather than $\mathbf{x}$ alone, and aligns the resulting *top-label* confidence with the target level $q$.

**Training objective** Draw $q_i \sim \mathcal{U}(0,1)$ and let $\hat{\mathbf{p}}(\mathbf{x}_i, q_i) = [\hat{p}_{i1}, \ldots, \hat{p}_{iC}]$ be the predicted class-probability vector produced by $\mathbf{H}_{\text{ICALD}}$. Let $y_i^* = \arg\max_{c \in \mathcal{Y}} \hat{p}_{ic}$ denote the top-label, with confidence $\hat{p}_{i,y_i^*}$. We then penalize the deviation $|\hat{p}_{i,y_i^*} - q_i|$:

$$\mathcal{L}_{\text{ICALD}} = \mathcal{L}_{\text{NLL}} \;+\; \beta\,\frac{1}{N}\sum_{i=1}^{N}|\hat{p}_{i,y_i^*} - q_i|, \quad \beta \geq 0, \ (12)$$

where the expectation over the anchor randomness is approximated by resampling $q_i$ at each iteration. Following Zhao et al. (2020), we further enforce that the *top-label* confidence $\hat{p}_{i,y_i^*}$ is *non-decreasing* with respect to the anchor $q_i$. Any continuous but non-monotonic mapping can be converted into a monotonic one by sorting its outputs over anchor levels (see Appendix A.2). Moreover, $\beta$ is a hyperparameter that balances discrimination and calibration.

While Section 2.3 enforces calibration via *distribution matching* (KCE/MMD) under different conditioning choices (*e.g.*, (11) promotes *canonical*-level *PACC*), $\mathbf{H}_{\text{ICALD}}$ instead

enforces *individualized calibration* on the top-label confidence, which in turn implies small KCE and hence *top-label*-level *PACC* (Theorem 2.5 below). To connect the confidence-alignment penalty in (12) with *PACC* guarantees, we briefly recall the notion and theorem of *individual calibration* for predictive CDFs and adapt it to our multi-class setting.

**Definition 2.2** (Probably Approximately Individually Calibrated (**PAIC**, Zhao et al. 2020))**.** A predictive CDF model $F(Y|\mathbf{x})$ is said to be $(\epsilon, \delta)$-PAIC if for all $\mathbf{x} \in \mathcal{X}$, $Y \in \mathcal{Y}$, and $q \in [0,1]$, the following holds:

$$\Pr\left[\int_0^1 |\Pr\left[F(Y|\mathbf{x}) \leq q\right] - q|\ dq \leq \epsilon\right] \geq 1 - \delta.$$

**Definition 2.3** (Monotonic Probably Approximately Individually Calibrated (**mPAIC**, Zhao et al. 2020))**.** A predictive CDF model $F(Y|\mathbf{x}, q)$ is said to be $(\epsilon, \delta)$-MPAIC if for all $\mathbf{x} \in \mathcal{X}$, $Y \in \mathcal{Y}$, and $q \in [0,1]$, the following holds:

$$\Pr\left[\int_0^1 |\Pr\left[F(Y|\mathbf{x}, q) \leq q\right] - q|\ dq \leq \epsilon\right] \geq 1 - \delta.$$

**Theorem 2.4** (**mPAIC** $\Rightarrow$ **PAIC**, Zhao et al. 2020)**.** *If a predictive CDF model $F(Y|\mathbf{x})$ is $(\epsilon, \delta)$-MPAIC, then for any $\epsilon' > \epsilon$, it is also $\left(\epsilon', \delta \cdot \frac{1-\epsilon}{\epsilon'-\epsilon}\right)$-PAIC.*

In our multi-class setting, we instantiate the CDF-based framework of Zhao et al. (2020) using the *top-label confidence* as the PIT-style (Rosenblatt, 1952) calibration quantity. Specifically, we interpret the scalar predictive CDF $F(Y|\mathbf{x})$ in Definition 2.2 as the analogue of the top-label confidence $\hat{p}_{y^*}(\mathbf{x})$ produced by $\mathbf{H}_{\text{ALD}}$, and the anchor-conditioned CDF $F(Y|\mathbf{x}, q)$ in Definition 2.3 as the analogue of $\hat{p}_{y^*}(\mathbf{x}, q)$ produced by $\mathbf{H}_{\text{ICALD}}$. Introducing the anchor $q$ makes calibration queryable and level-conditioned, allowing the classifier to cover all confidence levels in $(0,1)$ by sampling $q \sim \mathcal{U}(0,1)$ during training. Moreover, the monotonicity constraint in $q$ matches the requirement in Definition 2.3, so optimizing (12) naturally targets an *mPAIC* predictor. By Theorem 2.4, this further implies *PAIC*, providing theoretical support for *individualized calibration* at the *top-label* confidence level (*i.e.*, predictions made with confidence near $q$ are correct with probability about $q$).

**Theorem 2.5** (**PAIC** $\Rightarrow$ **PACC** (top-label))**.** *Let $\mathbf{H}$ be a multi-class classifier with predictive probabilities $\hat{\mathbf{p}}(X)$ and top-label confidence $\hat{p}_{Y^*}(X)$, where $Y^* := \arg\max_{c \in \mathcal{Y}} \hat{p}_c(X)$. Assume that $\hat{p}_{Y^*}(X)$ is $(\epsilon, \delta)$-PAIC in the sense of Definition 2.2. Let $k$ be a bounded characteristic kernel and let $\widehat{\text{KCE}}[k, \mathbf{H}, \mathcal{D}]$ denote the empirical KCE computed at the top-label level on an i.i.d. sample $\mathcal{D}$ of size $n$. Then there exist a kernel-dependent constant $a_k > 0$ and a sample term $b_{k,n} \to 0$ as $n \to \infty$ such that*

$$\Pr\left[\widehat{\text{KCE}}[k, \mathbf{H}, \mathcal{D}] \geq a_k\,\sqrt{\epsilon} + b_{k,n}\right] \leq \delta + o_n(1).$$

*Equivalently,* $\mathbf{H}$ *is* $(\epsilon', \delta')$-PACC *(Definition 2.1) at the* top-label *level with* $\epsilon' = a_k \sqrt{\epsilon} + b_{k,n}$ *and* $\delta' = \delta + o_n(1)$.

*Remark* 2.6. The constants $a_k$ and the explicit form of $b_{k,n}$ (from empirical process concentration for KCE/MMD) are provided in Appendix A.3. In particular, for bounded kernels with $\sup_z k(z,z) \leq \kappa^2$, one can take $a_k = \kappa$ and $b_{k,n} = O(n^{-1/2})$ up to logarithmic factors.

Our main theoretical result in Theorem 2.5 bridges *individual calibration* (Zhao et al., 2020) with *distribution-matching* calibration route such as KCE/MMD (Marx et al., 2023). In particular, if the *top-label confidence* $\hat{p}_{Y^*}(X)$ is $(\epsilon, \delta)$-*PAIC*, then the corresponding *top-label* KCE is also small with high probability, up to the kernel-dependent constant $a_k$ and the finite-sample term $b_{k,n}$. Consequently, (12) promotes calibration by aligning the anchor-conditioned *top-label* confidence $\hat{p}_{Y^*}(\mathbf{x}, q)$ to the queried level $q \sim \mathcal{U}(0,1)$, together with a monotonicity constraint in $q$. This directly targets an *mPAIC* predictor (Definition 2.3), which implies *PAIC* (Definition 2.2) via Theorem 2.4; applying Theorem 2.5 then further yields *top-label*-level *PACC* (Definition 2.1). In summary, minimizing (12) induces the implication chain *mPAIC* $\Rightarrow$ *PAIC* $\Rightarrow$ *PACC*, providing a distributional justification for the confidence-alignment objective (Proofs of Theorems 2.4 and 2.5 are given in the Appendices A.2 and A.3, respectively).

## 2.5. Learning with the ICALD Classifier

Although the theoretical learning objective is established in Section 2.4, applying it directly in practice presents non-trivial optimization challenges. The primary difficulty arises from the conflicting objectives of discriminative feature learning and calibration. Standard training often leads to *asynchronous convergence* (Kendall et al., 2018), where the feature extractor converges significantly faster than the calibration head, particularly on simpler datasets. This imbalance can cause the backbone to overfit to discriminative tasks before the calibration module effectively learns the confidence alignment.

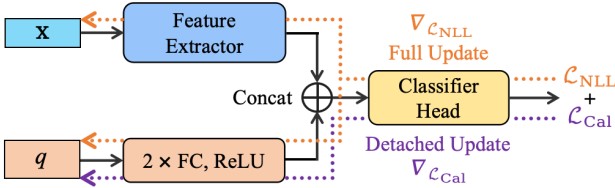

*Figure 1.* **Schematic of DDSO.** DDSO uses gradient detachment to decouple feature learning from calibration. Feature Extractor varies by input modality and FC is a fully connected layer. Classifier Head outputs distribution-specific parameters (see Section 2.2).

To reconcile these objectives, we propose the *Decoupled Dual-Stream Optimization (DDSO)* strategy, as illustrated in Figure 1. This strategy specifically addresses the con-

vergence mismatch and mitigates the risk of overfitting the backbone by separating computational pathways for feature extraction and calibration. Formally, within each training iteration, the protocol operates through two parallel streams described below.

**The Discriminative Stream (orange)** This stream processes inputs utilizing a fixed high-confidence query ($q = 1$). Optimizing $\mathcal{L}_{\text{NLL}}$ in (4) updates all model parameters via end-to-end backpropagation. This process serves as an optimization anchor that forces the backbone to learn sharp discriminative semantic representations essential for high classification accuracy.

**The Calibration Stream (purple)** In parallel, we stochastically sample anchor queries $q \sim \mathcal{U}(0,1)$. Crucially, we implement a *gradient detachment* strategy on the extracted features within this stream. This design serves two purposes. First, it acts as a structural regularizer against *feature collapse*: by blocking calibration gradients from flowing back to the backbone, we prevent the model from minimizing calibration error for low-confidence queries by degrading feature quality (*i.e.*, learning ambiguous representations), thereby guaranteeing that calibration improvements do not compromise discriminative accuracy. Second, it mitigates the convergence mismatch by allowing multiple anchors to be sampled per input within a single iteration. Since this stream updates only a lightweight calibration head, the additional computation is minimal, which accelerates calibration learning and helps it track the rapidly converging backbone without increasing training epochs.

The empirical benefits of this framework are presented systematically in Section 4.4, with detailed experimental results and ablation studies provided in Appendix C.3.

**Flexibility for Pre- & Post-Calibration** ICALD supports both training-time (*pre-calibration*) and inference-time (*post-calibration*) calibration, whereas most existing methods operate at a single stage. For *post-calibration*, we first optimize the base model (comprising the backbone and classifier head), denoted as $m_{\Phi}^{\text{Base}}$, using a standard task-specific loss (*e.g.*, cross entropy as in (4)). Subsequently, we introduce a lightweight adapter module $m_{\Psi}^{\text{Post}}$, typically parameterized as an multilayer perceptron (MLP). This module receives two inputs: the frozen logits (or features) from the base model and a stochastic anchor $q$ sampled from a uniform distribution $\mathcal{U}(0,1)$. Similar to the *pre-calibration* phase, we optimize exclusively the adapter parameters $\Psi$ via the ICALD calibration objective in (12). The adapter outputs adjustment factors $\gamma$, which modulate the fixed base parameters through element-wise multiplication:

$$\underbrace{m_{\Theta}^{\text{Pre}}(\mathbf{x}, q)}_{\hat{\mathbf{p}}_{\text{calibrated}}} = \underbrace{m_{\Psi}^{\text{Post}}(\mathbf{x}, q)}_{\text{Adjustment } \gamma} \odot \underbrace{m_{\Phi}^{\text{Base}}(\mathbf{x})}_{\hat{\mathbf{p}}_{\text{uncalibrated}}}. \quad (13)$$

In summary, the ICALD framework offers superior adapt-

ability for real-world deployment provided that it can be trained end-to-end for optimal feature-calibration alignment (*pre-calibration*) or attached as a lightweight adapter to efficiently correct already-trained models (*post-calibration*).

## 3. Related Work

**Classification and Softmax Function.** Modern multi-class classifiers (He et al., 2016; Vaswani et al., 2017; Devlin et al., 2019; Brown et al., 2020; Alexey et al., 2021) usually predict categorical probabilities by applying a Softmax transformation to logits and optimizing cross-entropy. Although this paradigm achieves strong discriminative performance across domains (Chen et al., 2024; Gong et al., 2025; Liu et al., 2025), its confidence estimates are often miscalibrated and overconfident (Guo et al., 2017; Marx et al., 2023; Vashistha & Farahi, 2025). Prior work has analyzed Softmax confidence from uncertainty and latent-utility perspectives (Pearce et al., 2021; McFadden, 1972), and has explored alternatives such as categorical relaxations (Jang et al., 2017), modified normalization functions (Banerjee et al., 2020), and Softmax variants in attention (Qin et al., 2022). Our work takes a complementary direction: instead of treating Softmax as a fixed normalization layer, we formulate classification through parametric distributions, including Normal, Logistic, Exponential, LogNormal, Weibull (Norton et al., 2021), and especially the asymmetric Laplace distribution (Sheng & Henao, 2025). This provides a drop-in distribution-based alternative to Softmax with explicit parameters for modeling sample-specific uncertainty.

**Calibration Methods.** Calibration methods aim to align predicted confidence with empirical correctness. Post-hoc methods learn a calibration map on top of a trained classifier, including Histogram Binning (Zadrozny & Elkan, 2001), isotonic regression (Zadrozny & Elkan, 2002), Bayesian Binning into Quantiles (Naeini et al., 2015), Platt Scaling (Platt et al., 1999), Temperature Scaling (Guo et al., 2017), and Dirichlet calibration (Kull et al., 2019). These methods are simple and often preserve the original classifier, but their expressiveness is limited by the chosen calibration map and the availability of validation data. Pre-calibration methods instead modify the training objective. Kernel-based approaches define differentiable calibration objectives through KDE or MMD-style distribution matching (Zhang et al., 2020; Kumar et al., 2018; Widmann et al., 2019; Marx et al., 2023). Other recent approaches improve calibration by modifying loss weights, logits, features, or training geometry, including class-wise loss scaling $\mathbf{H}_{\text{CWLS}}$ (Jung et al., 2023), Tilt-and-Average $\mathbf{H}_{\text{TNA}}$ (Cho & Youn, 2024), Feature Clipping $\mathbf{H}_{\text{FC}}$ (Tao et al., 2025), Dual Focal Loss $\mathbf{H}_{\text{DFL}}$ (Tao et al., 2023), and uncertainty-aware balanced softmax variants such as $\mathbf{H}_{\text{BSCE-GRA}}$ (Lin et al., 2025). Compared with these methods, our framework introduces an individualized calibration objective with an explicit implication chain from mPAIC to PAIC and then to PACC, while remaining compatible with both pre-calibration and post-calibration settings.

**Uncertainty-aware Predictive Distributions.** A related line of work models predictive uncertainty by placing distributions over class probabilities or predictive targets. Evidential Deep Learning $\mathbf{H}_{\text{EDL}}$ (Sensoy et al., 2018) predicts Dirichlet evidence for classification uncertainty, while Natural Posterior Networks $\mathbf{H}_{\text{NatPN}}$ (Charpentier et al., 2022) use a Bayesian posterior formulation for exponential-family predictive distributions. These methods provide useful uncertainty estimates, especially for uncertainty quantification and OOD-related settings, but they do not directly optimize individualized calibration of top-label confidence. In contrast, our $\mathbf{H}_{\text{ICALD}}$ framework combines a flexible distribution-based classifier with anchor-conditioned confidence alignment, directly targeting individualized calibration while preserving competitive discriminative performance.

## 4. Experiments

### 4.1. Datasets

We evaluate our methods on a comprehensive suite of datasets spanning tabular, image, and text modalities. For tabular data, we adopt the selection methodology introduced by Marx et al. (2023). Specifically, we utilize five UCI datasets: BREAST-CANCER (Wolberg et al., 1995), HEART-DISEASE (Janosi et al., 1988), ONLINE-SHOPPERS (Sakar & Kastro, 2018), DRY-BEAN (Koklu & Ozkan, 2020), and ADULT (Becker & Kohavi, 1996). For image classification, we adopt the standard CIFAR-10 and CIFAR-100 datasets (Krizhevsky et al., 2009). For text classification, we use the Stanford Sentiment Treebank (Socher et al., 2013), specifically the binary (SST-2) and fine-grained (SST-5) versions.

We follow standard practice by conducting each experiment across 5 random train/test splits. For each split, we randomly assign 70% of the data for training, 10% for validation (used for early stopping), and 20% for testing. Additional details regarding the datasets can be found in Appendix B.1. The source code to reproduce these experiments is available at https://github.com/demingsheng/HICALD.

### 4.2. Baselines

We compare the proposed calibration framework, including both *pre-calibration* and *post-calibration* variants (*i.e.*, $\mathbf{H}_{\text{IC-}}^{\text{Pre}}$ and $\mathbf{H}_{\text{IC-}}^{\text{Post}}$), against the standard Softmax classifier $\mathbf{H}_{\text{Softmax}}$ as well as the ALD-parameterized classifier $\mathbf{H}_{\text{ALD}}$. To ensure a fair comparison, all methods share the same feature extractor (backbone) and differ only in the output param-

eterization and/or the calibration procedure. For calibration baselines, we consider three representative methods: ($i$) MMD-based *pre-calibration* (Marx et al., 2023), which incorporates an additional distribution-matching objective during training; ($ii$) Temperature Scaling (TS, Guo et al. 2017), a *post-calibration* approach that rescales logits using a single temperature parameter; and ($iii$) Histogram Binning (HB, Zadrozny & Elkan 2001), a *post-calibration* method that maps predicted confidences to empirical accuracies via binning. The detailed descriptions and implementation notes for each baseline are presented in Appendix B.3.

### 4.3. Metrics

**Predictive Accuracy Metrics.** To provide a comprehensive assessment of model robustness, particularly across imbalanced domains, we report three key metrics: Accuracy (ACC, Makridakis 1993), Area Under the receiving operating Characteristic (AUC, Lobo et al. 2008), and Average Precision (AP, Zhu 2004). While Accuracy measures overall correctness, AUC and AP offer deeper insights into the model's discriminative capability and robustness against class imbalance. For multi-class classification tasks, we calculate AUC and AP using a *One-vs-Rest (OvR)* strategy and report the *Macro-Average* scores. This approach ensures that performance on the minority classes contributes equally to the final evaluation, preventing metrics from being dominated by majority classes.

**Calibration Metrics.** To rigorously evaluate the reliability of our model's confidence estimates, we utilize a comprehensive set of metrics covering *average-level*, *group-level*, and *distribution-level* calibration. We first report the Expected Calibration Error (ECE, Naeini et al. 2015) as the standard *average-level* metric, which approximates the expected absolute difference between model confidence and empirical accuracy using a binning strategy. Extending this to *group-level* analysis, we report GroupX_ECE and GroupY_ECE to assess calibration separately across different subpopulations (*e.g.*, Majority and Minority classes). Finally, we employ the Kernel Calibration Error (KCE, Marx et al. 2023) for the *distribution-level* calibration evaluation to address the potential biases associated with the binning schemes used in the metrics aforementioned. Detailed definitions and formulations for all metrics are provided in Appendix B.2.

### 4.4. Results

**Softmax *vs*. Other Distribution-based Classifiers.** We first compare $H_{ALD}$ against $H_{Softmax}$ and other distribution-based classifiers including Normal ($H_{Norm}$), Logistic ($H_{Logistic}$), Exponential ($H_{Exp}$), LogNormal ($H_{LogNorm}$), and Weibull ($H_{Weibull}$). For each distribution, we use the optimal threshold $t_0$ identified in our preliminary analysis (see Appendix C.1 for the sensitivity analysis on $t_0$). Figure 2

visualizes the performance comparison using Critical Difference (CD) diagrams (Demšar, 2006). Figure 2(a) presents the overall ranking, while Figure 2(b) highlights the calibration performance (ECE). The complete set of CD diagrams for all individual metrics (Figure 4) and a detailed analysis are provided in Appendix C.2.

In terms of overall performance shown in Figure 2(a), we can observe that $H_{ALD}$ achieves the lowest (best) average rank of 3.00, whereas $H_{Exp}$ exhibits the highest (worst) average rank of 4.82. The results also clearly demonstrate that $H_{ALD}$ consistently outperforms other common parametric distributions such as $H_{Logistic}$ (3.91) and $H_{Norm}$ (4.01). Crucially, the standard baseline $H_{Softmax}$ is a laggard with an overall rank of 4.21. Its weakness is particularly pronounced in calibration, as shown in Figure 2(b), $H_{Softmax}$ drops to the last rank (4.58) in terms of ECE. This indicates that while $H_{Softmax}$ may provide reasonable classification decisions, its confidence estimates are unreliable. In contrast, $H_{ALD}$ maintains the top position in ECE rank (3.07), demonstrating superior reliability.

A potential reason for the inferior performance of $H_{Exp}$ and $H_{Softmax}$ lies in their limited expressiveness, as both rely on single-parameter distributions. Specifically, $H_{Exp}$ models only the *rate* parameter $\lambda$, while $H_{Softmax}$ corresponds to a Gumbel distribution with a fixed scale parameter ($\beta = 1$), leaving only the *location* parameter $\mu$ to be learned (see Appendix A.1 for the proof). This fixed scale restricts the model's ability to adjust uncertainty, leading to poor accuracy or calibration. The increased expressiveness of flexible distribution-based classifiers confers considerable modeling advantages (see Appendix C.2 for a detailed analysis). In particular, the ALD—parameterized by *location*, *scale*, and *asymmetry*—provides a robust inductive bias, enabling the classifier to achieve both superior predictive accuracy and reliable confidence estimates.

Meanwhile, the additional computational burden introduced by these flexible classifiers is negligible (see Appendix C.1). In particular, we report the parameter count and the *relative* wall-clock training time normalized by $H_{Softmax}$. Although $H_{ALD}$ roughly triples the number of parameters, its relative time cost increases only marginally to $1.020\times$. Similarly, other two-parameter distribution-based classifiers remain comparably efficient, with relative time costs ranging from $1.003\times$ to $1.017\times$. These results suggest that $H_{ALD}$ can serve as a practical surrogate for $H_{Softmax}$, improving both accuracy and calibration with minimal additional runtime.

**NLL *vs*. Full Grad *vs*. Detachment.** To further understand how calibration-oriented optimization affects the learned representation, we analyze the *Fisher Ratio* of the backbone features across datasets. The *Fisher Ratio* is a classical measure of class separability derived from Fisher's linear discriminant analysis (Fisher, 1936), which quantifies the

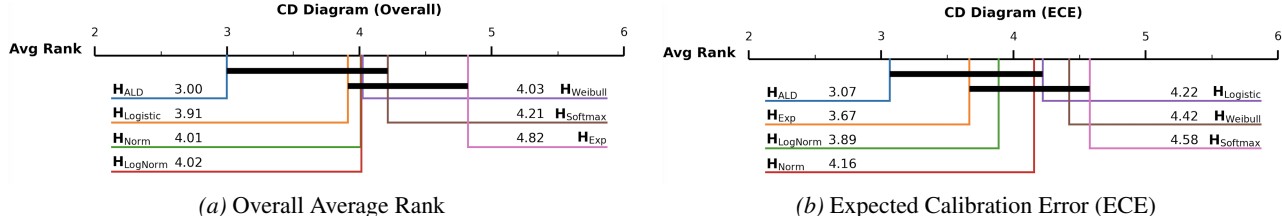

*(a)* Overall Average Rank            *(b)* Expected Calibration Error (ECE)

*Figure 2.* **Critical Difference (CD) diagrams comparing the average ranks of different classifiers.** (a) The overall average rank across all experimental settings ($N = 45$). (b) The average rank specifically for Expected Calibration Error (ECE). The horizontal axis represents the average rank (lower/left is better). The critical difference (CD = 1.34) is calculated using the Nemenyi test (Nemenyi, 1963) at a significance level of $\alpha = 0.05$. Classifiers connected by a thick horizontal bar are **not** statistically significantly different.

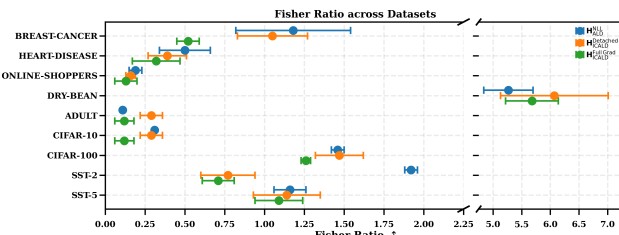

*Figure 3.* **Effect of *DDSO strategy* on Representation Separability.** *Fisher Ratio* of backbone features across datasets and under different optimization strategies.

relative strength of the inter-class scatter *versus* the intra-class scatter. Concretely, given the penultimate features (before class activation), a larger *Fisher Ratio* indicates that samples from different classes form tighter and more distinguishable clusters in the embedding space, while a smaller value suggests feature overlap and reduced discriminability.

Figure 3 reports the *Fisher Ratio* for three training variants (taking $\mathbf{H}_{\text{ALD}}$ as an example): (*i*) NLL Only ($\mathbf{H}_{\text{ALD}}^{\text{NLL}}$), which optimizes the standard likelihood objective in (4) without a calibration stream; (*ii*) Full Grad ($\mathbf{H}_{\text{ALD}}^{\text{Full Grad}}$), which introduces the ICALD calibration loss in (12) while allowing its gradients to update the backbone; and (*iii*) Detached ($\mathbf{H}_{\text{ALD}}^{\text{Detached}}$), which blocks calibration gradients from backpropagating into the backbone. Overall, we observe that $\mathbf{H}_{\text{ALD}}^{\text{Full Grad}}$ often yields a noticeably lower *Fisher Ratio* than $\mathbf{H}_{\text{ALD}}^{\text{NLL}}$, especially on datasets such as BREAST-CANCER and SST-2, indicating that directly propagating calibration gradients can distort discriminative features and reduce class separability. In contrast, $\mathbf{H}_{\text{ALD}}^{\text{Detached}}$ (*DDSO strategy*) consistently preserves (and in some cases improves) the *Fisher Ratio* relative to $\mathbf{H}_{\text{ALD}}^{\text{Full Grad}}$, recovering separability close to $\mathbf{H}_{\text{ALD}}^{\text{NLL}}$ on most datasets. Notably, on DRY-BEAN and ADULT, $\mathbf{H}_{\text{ALD}}^{\text{Detached}}$ even leads to higher *Fisher Ratios*, suggesting that decoupling the calibration objective can act as an implicit regularizer that stabilizes representation learning. These results support the key motivation of our *DDSO* strategy:

while calibration objectives improve probabilistic reliability, their gradients can interfere with discriminative representation learning. By isolating calibration updates from the backbone via gradient detachment, DDSO preserves feature separability while still enabling effective confidence calibration. Additional details can be found in Appendix C.3.

**Training and inference cost.** We compare the computational cost of different $\mathbf{H}_{\text{ICALD}}$ training strategies by normalizing $\mathbf{H}_{\text{ICALD}}^{\text{Full Grad}}$ to $1\times$. Under this normalization, $\mathbf{H}_{\text{ICALD}}^{\text{Detached}}$ only requires $0.554\times$ train+inference time, indicating a substantial reduction in cost. This highlights the efficiency advantage of *DDSO*, which achieves strong calibration improvements while avoiding the prohibitive overhead of full-gradient calibration training.

**Overall performance.** Finally, we provide a comprehensive comparison of $\mathbf{H}_{\text{ICALD}}$ against the standard Softmax baseline $\mathbf{H}_{\text{Softmax}}$, representative *post-calibration* methods (TS/HB), and the *pre-calibration* approach based on MMD. Table 1 reports Friedman average ranks (lower is better) with Nemenyi testing across $N = 45$ experimental settings.

Across the aggregated overall rankings, our $\mathbf{H}_{\text{ICALD}}$ variants achieve the strongest aggregate performance: $\mathbf{H}_{\text{ICALD}}^{\text{Pre}}$ attains the best overall rank (4.65), and $\mathbf{H}_{\text{ICALD}}^{\text{Post}}$ also remains highly competitive (5.03), consistently outperforming the $\mathbf{H}_{\text{Softmax}}$ baseline (7.11) and $\mathbf{H}_{\text{ALD}}$ (6.10). Importantly, these gains do not come at the expense of discriminative accuracy. $\mathbf{H}_{\text{ICALD}}^{\text{Pre}}$ ranks first on ACC (5.29) and remains one of the top methods on AP (5.40), demonstrating that individualized calibration preserves predictive strength while improving reliability.

On calibration metrics, $\mathbf{H}_{\text{ICALD}}$ provides consistent improvements across multiple criteria. In particular, $\mathbf{H}_{\text{ICALD}}^{\text{Pre}}$ achieves the best rank on ECE (3.76) and GroupX_ECE (3.90), while $\mathbf{H}_{\text{ICALD}}^{\text{Post}}$ achieves the best GroupY_ECE rank (3.31), indicating a strong reliability across subpopulations. These gains are competitive with distribution-matching baselines: $\mathbf{H}_{\text{Softmax+MMD}}$ yields the best KCE rank (3.29), while $\mathbf{H}_{\text{ALD+MMD}}$ is close (3.49), and $\mathbf{H}_{\text{ICALD}}$ remains competitive

*Table 1.* **Average ranks from the Critical Difference (CD) diagrams across** $N = 45$ **experimental settings.** Lower is better and the best (lowest) rank in each column is  highlighted . The critical difference (CD = 2.48) is computed via the Nemenyi test at $\alpha = 0.05$. The visualization and the underlying values for each metric are reported in Appendix C.4.

| | Model | Overall | ACC | AUC | AP | ECE | GroupX_ECE | GroupY_ECE | KCE |
|---|---|---|---|---|---|---|---|---|---|
| *Base* | $\mathbf{H}_{\text{Softmax}}$ | 7.11 | 5.99 | **5.13** | 5.97 | 9.07 | 6.93 | 8.64 | 7.93 |
| | $\mathbf{H}_{\text{ALD}}$ | 6.10 | 5.97 | 5.56 | 5.49 | 7.08 | 5.40 | 7.29 | 6.00 |
| *Post-cal* | $\mathbf{H}_{\text{Softmax+TS}}$ | 7.74 | 5.99 | 5.29 | 6.20 | 8.51 | 9.22 | 8.84 | 10.91 |
| | $\mathbf{H}_{\text{Softmax+HB}}$ | 8.72 | 8.91 | 9.07 | 9.82 | 8.00 | 8.57 | 7.78 | 8.73 |
| | $\mathbf{H}_{\text{ICSoftmax}}^{\text{Post}}$ | 5.78 | 6.44 | 6.82 | 6.61 | 4.53 | 5.00 | 5.40 | 5.29 |
| | $\mathbf{H}_{\text{ALD+TS}}$ | 7.23 | 5.97 | 5.71 | 5.56 | 7.62 | 10.38 | 6.96 | 9.82 |
| | $\mathbf{H}_{\text{ALD+HB}}$ | 7.78 | 8.22 | 9.60 | 8.96 | 6.02 | 7.83 | 5.71 | 8.04 |
| | $\mathbf{H}_{\text{ICALD}}^{\text{Post}}$ | 5.03 | 5.93 | 6.93 | 5.42 | 4.80 | 4.27 | **3.31** | 4.13 |
| *Pre-cal* | $\mathbf{H}_{\text{Softmax+MMD}}$ | 6.23 | 6.13 | 6.10 | 6.47 | 7.86 | 5.73 | 7.76 | **3.29** |
| | $\mathbf{H}_{\text{ICSoftmax}}^{\text{Pre}}$ | 5.82 | 7.40 | 5.89 | 5.69 | 4.69 | 4.60 | 5.78 | 6.11 |
| | $\mathbf{H}_{\text{ALD+MMD}}$ | 5.80 | 5.76 | 5.89 | 6.42 | 6.07 | 6.17 | 6.84 | 3.49 |
| | $\mathbf{H}_{\text{ICALD}}^{\text{Pre}}$ | **4.65** | **5.29** | 6.01 | **5.40** | **3.76** | **3.90** | 3.69 | 4.24 |

on KCE ($\mathbf{H}_{\text{ICALD}}^{\text{Post}}$: 4.13 and $\mathbf{H}_{\text{ICALD}}^{\text{Pre}}$: 4.24).

In contrast, *post-calibration* baselines (TS/HB) tend to exhibit metric-specific behavior, often improving *average calibration* while providing less consistent gains on *group-* and *distribution-level* metrics, which results in weaker overall ranks. Finally, *pre-calibration* approaches based on distribution alignment (MMD) offer a more principled objective, but are not uniformly strong across all calibration criteria. Overall, $\mathbf{H}_{\text{ICALD}}$ provides a holistic improvement by maintaining competitive discriminative performance while substantially enhancing probabilistic reliability at the average, group, and distribution levels, with gains consistently supported by the CD analysis over diverse datasets and modalities.

**Additional comparison.** To further evaluate the competitiveness of our method against recent calibration and uncertainty estimate baselines, we additionally compare $\mathbf{H}_{\text{ICALD}}^{\text{Pre}}$ with eight representative methods on the five tabular datasets: $\mathbf{H}_{\text{CWLS}}$ (Jung et al., 2023), $\mathbf{H}_{\text{TNA}}$ (Cho & Youn, 2024), $\mathbf{H}_{\text{FC}}$ (Tao et al., 2025), $\mathbf{H}_{\text{DFL}}$ (Tao et al., 2023), $\mathbf{H}_{\text{BSCE-GRA}}$ (Lin et al., 2025), $\mathbf{H}_{\text{EDL}}$ (Sensoy et al., 2018), $\mathbf{H}_{\text{NatPN}}$ (Charpentier et al., 2022), and $\mathbf{H}_{\text{Dirichlet}}$ (Kull et al., 2019). The detailed results are provided in Appendix C.5. Overall, $\mathbf{H}_{\text{ICALD}}^{\text{Pre}}$ achieves the best aggregate average rank among all methods. In particular, it obtains the best overall rank of 2.17, substantially outperforming the standard $\mathbf{H}_{\text{Softmax}}$ baseline (7.23) and remaining stronger than the additional calibration and uncertainty baselines. Across the averaged metric-wise ranks, $\mathbf{H}_{\text{ICALD}}^{\text{Pre}}$ ranks first on ACC, AUC, AP, ECE, GroupX_ECE, Brier score, and KCE, while remaining competitive on GroupY_ECE and Classwise_ECE. This indicates that the proposed individualized calibration objective improves not only average calibration but also discriminative performance and distribution-level calibration. The detailed reliability diagrams in Appendix C.5 further show that $\mathbf{H}_{\text{ICALD}}^{\text{Pre}}$ generally yields smaller confidence-accuracy gaps across datasets.

## 5. Conclusions

We presented a distribution-based classifier built on the asymmetric Laplace distribution $\mathbf{H}_{\text{ALD}}$, and an individualized variant $\mathbf{H}_{\text{ICALD}}$ to improve reliability in multi-class prediction. Our approach bridges *individual calibration* guarantees and *distribution-matching calibration* metrics, and supports both training-time *pre-calibration* and inference-time *post-calibration* through a lightweight adapter. To address the practical optimization difficulty caused by competing discrimination and calibration objectives, we introduced a *Decoupled Dual-Stream Optimization (DDSO) strategy* that prevents feature collapse and synchronizes convergence, enabling consistent calibration improvements without sacrificing accuracy. Extensive experiments with nine real-world datasets with seven complementary metrics demonstrate that $\mathbf{H}_{\text{ICALD}}$ provides a robust and favorable trade-off between predictive performance and calibration.

**Limitations and future work** First, distribution-based classifiers such as $\mathbf{H}_{\text{ALD}}$ introduce additional design choices, including the dependence on the $t_0$ hyperparameter, which adds a tuning dimension absent from standard Softmax training. Second, our theoretical guarantees and training objective primarily focus on individualized calibration at the *top-label confidence* level. Extending the same framework to stronger notions, such as *canonical* calibration remains an important direction. Third, while our method yields consistent improvements, especially on ECE, it increases computational cost compared to standard training. In particular, full-gradient calibration is expensive and although our *DDSO* strategy substantially reduces the overhead, it does not eliminate it entirely. When normalizing $\mathbf{H}_{\text{ICALD}}^{\text{Full Grad}}$ to $1\times$, $\mathbf{H}_{\text{ICALD}}^{\text{Detached}}$ requires only $0.554\times$ train+inference time while maintaining strong calibration improvements. Developing more efficient calibration objectives and training strategies to further reduce overhead remains a promising avenue for future work.

## Impact Statement

This work improves the reliability of probabilistic predictions by developing calibration-aware multi-class classifiers. Better-calibrated confidence estimates can support safer decision-making in high-stakes applications and improve reliability across subpopulations. Potential risks include over-reliance on confidence scores under dataset bias or distribution shift, so deployment should include monitoring and application-specific safeguards.

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

# A. Analytical Results

This section collects the main analytical results that motivate our design choices and connect our training objectives to formal calibration guarantees. Appendix A.1 revisits the standard Softmax parameterization from a latent-variable perspective, showing that it arises as the exact discrete-choice probability induced by maximizing class utilities perturbed with i.i.d. standard Gumbel noise. Appendix A.2 instantiates the CDF-based individual calibration framework of Zhao et al. (2020) in multi-class classification via the *top-label confidence*, and proves that monotonic individual calibration (*mPAIC*) implies *PAIC* (Theorem 2.4), together with a discussion of when monotonicity is needed and how monotone rearrangement can be used for rigorous evaluation. Finally, Appendix A.3 further links *PAIC* to kernel-based calibration by establishing a high-probability implication from top-label *PAIC* to *PACC* under bounded characteristic kernels, yielding a finite-sample guarantee for the empirical kernel calibration error.

## A.1. Softmax as a Gumbel-Based Latent Variable Model

In this section, we will show the proof for that the Softmax function is the natural consequence of maximizing a latent utility with standard Gumbel-distributed noise.

**Definition A.1** (**Gumbel Distribution, Gumbel 1935**). A random variable $Y$ is said to have a Gumbel distribution with parameters $(\mu, \beta)$, denoted $Y \sim \text{Gumbel}(\mu, \beta)$, if its PDF is:

$$f_{\text{Gumbel}}(y; \mu, \beta) = \frac{1}{\beta} \exp\left(-\frac{y - \mu}{\beta} - \exp\left(-\frac{y - \mu}{\beta}\right)\right), \tag{14}$$

where $\mu \in \mathbb{R}$ is the *location* parameter and $\beta > 0$ is the *scale* parameter.

*Proof.* Let the utility for class $c$ be $U_c = z_c + \epsilon_c$, where $z_c$ is is the logit output by the neural network and $\epsilon_c \sim \text{Gumbel}(0, 1)$ are independent and identically distributed (*i.i.d.*) noise terms.

Specifically, it assumes a latent utility $U_c$ for each class $c$:

$$U_c = z_c + \epsilon_c, \quad \epsilon_c \sim \text{Gumbel}(\mu = 0, \beta = 1). \tag{15}$$

According to Definition A.1, the PDF and CDF of the standard Gumbel distribution are given by:

$$f_{\text{Gumbel}}(x) = e^{-(x + e^{-x})}, \tag{16}$$

$$F_{\text{Gumbel}}(x) = e^{-e^{-x}}. \tag{17}$$

The probability that class $c$ is selected corresponds to the probability that $U_c$ is the maximum among all utilities:

$$\Pr(y = c) = \Pr(U_c > U_j, \forall j \neq c). \tag{18}$$

We express this probability by conditioning on $\epsilon_c = t$. For class $c$ to be the maximum, we require $z_c + t > z_j + \epsilon_j$, which implies $\epsilon_j < t + z_c - z_j$ for all $j \neq c$. Integrating over the density of $\epsilon_c$, and exploiting the independence of the error terms, we obtain:

$$\Pr(y = c) = \int_{-\infty}^{\infty} f_{\text{Gumbel}}(t) \prod_{j \neq c} \Pr(\epsilon_j < t + z_c - z_j) \, dt = \int_{-\infty}^{\infty} f_{\text{Gumbel}}(t) \prod_{j \neq c} F_{\text{Gumbel}}(t + z_c - z_j) \, dt. \tag{19}$$

Substituting the PDF and CDF expressions from Eq. (16) and Eq. (17):

$$\Pr(y = c) = \int_{-\infty}^{\infty} e^{-(t + e^{-t})} \prod_{j \neq c} \exp\left(-e^{-(t + z_c - z_j)}\right) \, dt. \tag{20}$$

We simplify the product term by moving the product inside the exponent as a sum:

$$\prod_{j \neq c} \exp\left(-e^{-(t + z_c - z_j)}\right) = \exp\left(-\sum_{j \neq c} e^{-t} e^{-(z_c - z_j)}\right) = \exp\left(-e^{-t} \sum_{j \neq c} \frac{e^{z_j}}{e^{z_c}}\right). \tag{21}$$

Combining this with the PDF term $e^{-(t+e^{-t})}$, and noting that $e^{-e^{-t}} = \exp(-e^{-t} \cdot \frac{e^{z_c}}{e^{z_c}})$ represents the $j = c$ term in the summation, the integral simplifies to:

$$\Pr(y = c) = \int_{-\infty}^{\infty} e^{-t} \exp\left(-e^{-t} \sum_{j=1}^{C} \frac{e^{z_j}}{e^{z_c}}\right) dt. \tag{22}$$

Let $A = \sum_{j=1}^{C} \frac{e^{z_j}}{e^{z_c}}$. The integral becomes:

$$\Pr(y = c) = \int_{-\infty}^{\infty} e^{-t} e^{-Ae^{-t}} dt. \tag{23}$$

We apply integration by substitution using $u = e^{-t}$, which implies $du = -e^{-t}dt$. The limits change from $(-\infty, \infty)$ to $(\infty, 0)$:

$$\Pr(y = c) = \int_{\infty}^{0} e^{-Au}(-du) = \int_{0}^{\infty} e^{-Au} \, du. \tag{24}$$

Solving this exponential integral:

$$\Pr(y = c) = \left[-\frac{1}{A}e^{-Au}\right]_{0}^{\infty} = 0 - \left(-\frac{1}{A}\right) = \frac{1}{A}. \tag{25}$$

Finally, substituting $A$ back yields the standard Softmax function:

$$\Pr(y = c) = \frac{1}{\sum_{j=1}^{C} \frac{e^{z_j}}{e^{z_c}}} = \frac{e^{z_c}}{\sum_{j=1}^{C} e^{z_j}}. \tag{26}$$

This derivation explicitly confirms that the standard Softmax operation is mathematically equivalent to a discrete choice model governed by Gumbel-distributed errors. Thus, $\mathbf{H}_{\text{Softmax}}$ can be rigorously characterized as a Gumbel-based latent variable model with a fixed scale parameter ($\beta = 1$). $\square$

### A.2. mPAIC implies PAIC

In this section, we prove that the *top-label confidence* induced by $\mathbf{H}_{\text{ICALD}}$, namely $\hat{p}_{Y^*}(X)$, is $(\epsilon, \delta)$-*PAIC* when $\mathbf{H}_{\text{ICALD}}$ is trained with $\mathcal{L}_{\text{ICALD}}$ in (12). Furthermore, for any $\epsilon' > \epsilon$, it is also $\left(\epsilon', \delta \cdot \frac{1-\epsilon}{\epsilon'-\epsilon}\right)$-*PAIC*. Note that, in our multi-class setting, we instantiate the CDF-based framework of Zhao et al. (2020) and Sheng & Henao (2026) using the *top-label confidence* as the PIT-style (Rosenblatt, 1952) calibration quantity. Specifically, we interpret the scalar predictive CDF $F(Y|\mathbf{x})$ in Definition 2.2 as the analogue of the top-label confidence $\hat{p}_{y^*}(\mathbf{x})$ produced by $\mathbf{H}_{\text{ALD}}$, and the anchor-conditioned CDF $F(Y|\mathbf{x}, q)$ in Definition 2.3 as the analogue of $\hat{p}_{y^*}(\mathbf{x}, q)$ produced by $\mathbf{H}_{\text{ICALD}}$. Now, we present the proof of Theorem 2.4 step by step. We begin by recalling the definitions of *PAIC* and *mPAIC* in Definitions 2.2 and 2.3.

**Definition 2.2** (Probably Approximately Individually Calibrated (**PAIC**, Zhao et al. 2020)). A predictive CDF model $F(Y|\mathbf{x})$ is said to be $(\epsilon, \delta)$-PAIC if for all $\mathbf{x} \in \mathcal{X}$, $Y \in \mathcal{Y}$, and $q \in [0, 1]$, the following holds:

$$\Pr\left[\int_{0}^{1} |\Pr\left[F(Y|\mathbf{x}) \leq q\right] - q| \, dq \leq \epsilon\right] \geq 1 - \delta.$$

**Definition 2.3** (Monotonic Probably Approximately Individually Calibrated (**mPAIC**, Zhao et al. 2020)). A predictive CDF model $F(Y|\mathbf{x}, q)$ is said to be $(\epsilon, \delta)$-MPAIC if for all $\mathbf{x} \in \mathcal{X}$, $Y \in \mathcal{Y}$, and $q \in [0, 1]$, the following holds:

$$\Pr\left[\int_{0}^{1} |\Pr\left[F(Y|\mathbf{x}, q) \leq q\right] - q| \, dq \leq \epsilon\right] \geq 1 - \delta.$$

**Theorem 2.4** (**mPAIC** $\Rightarrow$ **PAIC**, Zhao et al. 2020). If a predictive CDF model $F(Y|\mathbf{x})$ is $(\epsilon, \delta)$-MPAIC, then for any $\epsilon' > \epsilon$, it is also $\left(\epsilon', \delta \cdot \frac{1-\epsilon}{\epsilon'-\epsilon}\right)$-PAIC.

*Proof.* Consider the random variables $Y \sim F_{Y|\mathbf{x}}$ and $Q \sim \mathcal{U}(0,1)$. We can view the *PAIC* calibration error as a discrepancy measure between the distribution of predicted cumulative probabilities and the uniform distribution. Define the integrated calibration error using the 1-Wasserstein distance (Villani et al., 2008):

$$\text{ECE}(F) = \int_0^1 |\Pr[F(Y|X) \leq q] - q| \, dq = d_{W_1}\left(\mathbb{F}_{F(Y|X)}, \mathbb{F}_U\right), \tag{27}$$

where $\mathbb{F}_{F(Y|X)}$ denotes the true CDF of the predicted cumulative probabilities, and $\mathbb{F}_U$ denotes the CDF of $\mathcal{U}(0,1)$. Intuitively, $d_{W_1}\left(\mathbb{F}_{F(Y|X)}, \mathbb{F}_U\right)$ tries to integrate the difference between the curve $q \mapsto \Pr[F(Y|X) \leq q]$ and the curve $q \mapsto q$. Now define the pointwise and conditional calibration errors as

$$\mathcal{E}(\mathbf{x}, y) = d_{W_1}\left(\mathbb{F}_{F(y|\mathbf{x},Q)}, \mathbb{F}_U\right), \qquad \mathcal{E}(\mathbf{x}) = d_{W_1}\left(\mathbb{F}_{F(Y|\mathbf{x},Q)}, \mathbb{F}_U\right). \tag{28}$$

Then, we can obtain the bound

$$\mathcal{E}(\mathbf{x}, y) \leq \mathbb{E}_{Q \sim \mathcal{U}(0,1)}\left[|F(y|\mathbf{x}, Q) - Q|\right]. \tag{29}$$

According to the Kantorovich-Rubinstein duality for the 1-Wasserstein distance (Villani et al., 2008), for any two distributions $\mu, \nu$:

$$d_{W_1}(\mu, \nu) = \sup_{\|\psi\|_{\text{Lip}} \leq 1} \left(\int \psi \, d\mu - \int \psi \, d\nu\right) = \sup_{\|\psi\|_{\text{Lip}} \leq 1} |\mathbb{E}_\mu[\psi] - \mathbb{E}_\nu[\psi]|, \tag{30}$$

where the supremum is taken over all 1-Lipschitz functions $\psi : \mathbb{R} \to \mathbb{R}$, *i.e.*, functions that satisfy

$$|\psi(x) - \psi(y)| \leq |x - y|, \quad \forall x, y \in \mathbb{R}. \tag{31}$$

Applying this duality to $F(y|\mathbf{x}, Q)$ and $Q$, and choosing the 1-Lipschitz function $\psi(a) = a$, we can obtain:

$$d_{W_1}(\mathbb{F}_{F(y|\mathbf{x},Q)}, \mathbb{F}_U) \leq |\mathbb{E}_{Q \sim \mathcal{U}(0,1)}[F(y|\mathbf{x}, Q)] - \mathbb{E}_{Q \sim \mathcal{U}(0,1)}[Q]| = |\mathbb{E}_{Q \sim \mathcal{U}(0,1)}[F(y|\mathbf{x}, Q) - Q]|. \tag{32}$$

Finally, applying Jensen's inequality ($|\mathbb{E}[A]| \leq \mathbb{E}[|A|]$ over $Q \sim \mathcal{U}(0,1)$) yields:

$$\begin{aligned} d_{W_1}(\mathbb{F}_{F(y|\mathbf{x},Q)}, \mathbb{F}_U) &\leq \left|\mathbb{E}_{Q \sim \mathcal{U}(0,1)}[F(y|\mathbf{x}, Q) - Q]\right| \\ &\leq \mathbb{E}_{Q \sim \mathcal{U}(0,1)}[|F(y|\mathbf{x}, Q) - Q|] = \mathbb{E}_{Q \sim \mathcal{U}(0,1)}[|F(y|\mathbf{x}, Q) - Q|]. \end{aligned} \tag{33}$$

Moreover, the inequality becomes an equality when $q \mapsto F(y|\mathbf{x}, q)$ is nondecreasing monotonically in $q$. Let $V = F(y|\mathbf{x}, Q)$, where $Q \sim \mathcal{U}(0,1)$. Then the CDF of $V$ is given by:

$$\mathbb{F}_V(v) = \Pr(V \leq v) = \Pr(F(y|\mathbf{x}, Q) \leq v). \tag{34}$$

Now, if $F(y|x, q)$ is a monotonically nondecreasing continuous function of $q$, then the mapping $q \mapsto F(y|x, q)$ is measure-preserving (Villani et al., 2008). This implies that:

$$\mathbb{F}_V(v) = \Pr(Q \leq F^{-1}(y|\mathbf{x}, v)), \tag{35}$$

and hence,

$$\mathbb{F}_V^{-1}(q) = F(y|\mathbf{x}, q), \quad \forall q \in [0, 1]. \tag{36}$$

Let $\mathbb{F}_U$ denote the CDF of the uniform distribution $\mathcal{U}(0,1)$, that is,

$$\mathbb{F}_U(u) = \Pr(U \leq u) = u, \quad \text{so} \quad \mathbb{F}_U^{-1}(q) = q, \quad \forall q \in [0, 1]. \tag{37}$$

According to the property for the 1-Wasserstein distance (Villani et al., 2008) between two distributions $\mu$ and $\nu$ on the real line, if $\mathbb{F}_\mu^{-1}$ and $\mathbb{F}_\nu^{-1}$ are their respective quantile functions (*i.e.*, the inverse function of CDF), then:

$$d_{W_1}(\mu, \nu) = \int_0^1 \left|\mathbb{F}_\mu^{-1}(q) - \mathbb{F}_\nu^{-1}(q)\right| \, dq. \tag{38}$$

Applying this identity to $V = F(y|\mathbf{x}, Q)$ and $Q$, we can obtain:

$$d_{W_1}(\mathbb{F}_{F(y|\mathbf{x},Q)}, \mathbb{F}_U) = \int_0^1 |F(y|\mathbf{x}, q) - q| \, dq = \mathbb{E}_{Q \sim \mathcal{U}(0,1)}\left[|F(y|\mathbf{x}, Q) - Q|\right]. \tag{39}$$

Similarly, inequality (40) follows from the same reasoning as inequality (29), the application of Kantorovich–Rubinstein duality followed by Jensen's inequality, with an additional expectation over $Y \sim F_{Y|x}$:

$$\mathcal{E}(\mathbf{x}) \leq \mathbb{E}_{Y \sim F_{Y|\mathbf{x}}, Q \sim \mathcal{U}(0,1)} \left[ |F(Y|\mathbf{x}, Q) - Q| \right]. \tag{40}$$

**Contradiction argument.** We now proceed by contradiction. Suppose, for contradiction, that $F$ is *not* $(\epsilon', \delta')$-*PAIC*. That is, by Definition 2.2, we have:

$$\Pr\left[\mathcal{E}(\mathbf{x}) > \epsilon'\right] > \delta'. \tag{41}$$

The violation of *PAIC* implies the existence of a failure set $\Omega_{\text{fail}} \subset \mathcal{X}$:

$$\Omega_{\text{fail}} := \left\{ \mathbf{x} \in \mathcal{X}, \mathbb{E}_{Y \sim F_{Y|\mathbf{x}}, Q \sim \mathcal{U}(0,1)} \left[ |F(Y|\mathbf{x}, Q) - Q| \right] \geq \epsilon' \right\}, \tag{42}$$

and by the inequality (40), we can know that whenever $\mathcal{E}(\mathbf{x}) \geq \epsilon'$ we have $\mathbf{x} \in \Omega_{\text{fail}}$, thus we can conclude:

$$\Pr[\mathbf{X} \in \Omega_{\text{fail}}] > \delta'. \tag{43}$$

Then, for any $\epsilon < \epsilon'$ and $\mathbf{x} \in \Omega_{\text{fail}}$, by bounding the expectation, we have:

$$\epsilon' \leq \mathbb{E}_{Y \sim F_{Y|\mathbf{x}}, Q \sim \mathcal{U}(0,1)}[|F(Y|\mathbf{x}, Q) - Q|] \tag{44}$$

$$\leq \epsilon \cdot \Pr[|F(Y|\mathbf{x}, Q) - Q| < \epsilon] + \Pr[|F(Y|\mathbf{x}, Q) - Q| \geq \epsilon], \tag{45}$$

where inequality (45) holds because the absolute deviation term $|F(Y|\mathbf{x}, Q) - Q| \in [0, 1]$. Now, letting $p = \Pr[|F(Y|\mathbf{x}, Q) - Q| \geq \epsilon]$, we can solve:

$$\epsilon' \leq \epsilon(1 - p) + p \Rightarrow p \geq \frac{\epsilon' - \epsilon}{1 - \epsilon}. \tag{46}$$

Combining this with the bound over $\mathbf{x} \in \Omega_{\text{fail}}$ (*i.e.*, inequality (43)) and applying the law of total probability, we can obtain:

$$\Pr\left[|F(Y|\mathbf{x}, Q) - Q| \geq \epsilon\right] = \Pr\left[|F(Y|\mathbf{x}, Q) - Q| \geq \epsilon | \mathbf{x} \in \Omega_{\text{fail}}\right] \Pr\left[\mathbf{x} \in \Omega_{\text{fail}}\right] \tag{47}$$

$$+ \Pr\left[|F(Y|\mathbf{x}, Q) - Q| \geq \epsilon | \mathbf{x} \notin \Omega_{\text{fail}}\right] \Pr\left[\mathbf{x} \notin \Omega_{\text{fail}}\right] > \frac{\epsilon' - \epsilon}{1 - \epsilon} \cdot \delta'.$$

**Violation of MPAIC.** By Definition 2.3, $(\epsilon, \delta)$-*mPAIC* requires: $\Pr[|F(Y|\mathbf{x}, Q) - Q| > \epsilon] < \delta$. Thus, equation (47) implies that $F$ is not $(\epsilon, \delta' \cdot \frac{\epsilon' - \epsilon}{1 - \epsilon})$-*mPAIC*.

**Contrapositive and conclusion.** We have shown:

$$\text{Not } (\epsilon', \delta')\text{-}PAIC \Rightarrow \text{Not } (\epsilon, \delta' \cdot \tfrac{\epsilon' - \epsilon}{1 - \epsilon})\text{-}mPAIC, \quad \forall \epsilon' > \epsilon. \tag{48}$$

Taking the contrapositive:

$$(\epsilon, \delta)\text{-}mPAIC \Rightarrow (\epsilon', \delta \cdot \tfrac{1 - \epsilon}{\epsilon' - \epsilon})\text{-}PAIC, \quad \forall \epsilon' > \epsilon. \tag{49}$$

Finally, we can conclude that if $F$ is not $(\epsilon', \delta')$-*PAIC*, then for any $\epsilon < \epsilon'$, it is not $\left(\epsilon, \delta' \cdot \frac{\epsilon' - \epsilon}{1 - \epsilon}\right)$-*mPAIC*, which is equivalent to Theorem 2.4, *i.e.*, if $F$ is $(\epsilon, \delta)$-*mPAIC*, then for any $\epsilon' > \epsilon$, it is also $\left(\epsilon', \delta \cdot \frac{1 - \epsilon}{\epsilon' - \epsilon}\right)$-*PAIC*. $\qquad\square$

**Monotonicity constraints.** Importantly, during training we do *not* need to explicitly enforce monotonicity for the learning objective to remain valid. The loss in (11) can be interpreted as an upper-bound surrogate of the true calibration error, and any violation of monotonicity only makes this bound more conservative (see the derivations from (29) to (33)). In our experiments, we do *not* use *PAIC/mPAIC*-type individual calibration metrics as the primary evaluation criterion. Therefore, we do not enforce monotonicity during training, nor do we rely on it when reporting the main results.

However, monotonicity becomes important if one wishes to *directly compute* individual calibration errors according to Definition 2.2 and Definition 2.3, since those definitions implicitly assume a measure-preserving (monotone) mapping in the anchor variable. For completeness and to facilitate reproducibility, when the learned mapping is continuous but non-monotone, we can apply a standard *monotone rearrangement* post-processing step: sample anchors $q_1, \ldots, q_K \sim \mathcal{U}(0, 1)$, compute $f_i = \hat{p}_{Y^*}(\mathbf{x}, q_i)$, sort $\{f_i\}_{i=1}^K$ into a nondecreasing sequence, and then reassign the sorted values back to the anchors. This procedure preserves the multiset of predicted values while enforcing a nondecreasing dependence on $q$, producing a monotone predictor suitable for rigorous *PAIC/mPAIC* evaluation.

**A.3. PAIC implies PACC**

In this section, we prove our main results in Theorem 2.5, showing that $(\epsilon, \delta)$-*PAIC* of the *top-label confidence* implies a *PACC* guarantee under a bounded kernel. We begin by recalling the definition of *Probably Approximately Calibrated Classifier* (*PACC*, Definition 2.1) and the statement of Theorem 2.5.

**Definition 2.1 (Probably Approximately Calibrated Classifier; PACC).** Let $\mathbf{H}$ be a multi-class classifier and let $\widehat{\mathrm{KCE}}[k, \mathbf{H}, \mathcal{D}]$ denote its empirical kernel calibration error computed with a kernel $k(\cdot, \cdot)$ on an i.i.d. dataset $\mathcal{D} = \{(\mathbf{x}_n, y_n)\}_{n=1}^N$. We say that $\mathbf{H}$ is an $(\epsilon, \delta)$-*Probably Approximately Calibrated Classifier* (PACC) under $k$ if

$$\Pr\left[ \widehat{\mathrm{KCE}}[k, \mathbf{H}, \mathcal{D}] \geq \epsilon \right] \leq \delta. \tag{50}$$

**Theorem 2.5 (PAIC $\Rightarrow$ PACC (top-label)).** Let $\mathbf{H}$ be a multi-class classifier with predictive probabilities $\hat{\mathbf{p}}(X)$ and top-label confidence $\hat{p}_{Y^*}(X)$, where $Y^* := \arg\max_{c \in \mathcal{Y}} \hat{p}_c(X)$. Assume that $\hat{p}_{Y^*}(X)$ is $(\epsilon, \delta)$-*PAIC* in the sense of Definition 2.2. Let $k$ be a bounded characteristic kernel and let $\widehat{\mathrm{KCE}}[k, \mathbf{H}, \mathcal{D}]$ denote the empirical KCE computed at the top-label level on an i.i.d. sample $\mathcal{D}$ of size $n$. Then there exist a kernel-dependent constant $a_k > 0$ and a sample term $b_{k,n} \to 0$ as $n \to \infty$ such that

$$\Pr\left[ \widehat{\mathrm{KCE}}[k, \mathbf{H}, \mathcal{D}] \geq a_k \sqrt{\epsilon} + b_{k,n} \right] \leq \delta + o_n(1). \tag{51}$$

Equivalently, $\mathbf{H}$ is $(\epsilon', \delta')$-*PACC* at the *top-label* level with $\epsilon' = a_k \sqrt{\epsilon} + b_{k,n}$ and $\delta' = \delta + o_n(1)$.

*Proof.* Assume $\hat{p}_{Y^*}(X)$ is $(\epsilon, \delta)$-*PAIC* in the sense of Definition 2.2. Define the per-$\mathbf{x}$ individual calibration error

$$\mathrm{err}(x) := \int_0^1 \left| \Pr\left[ \hat{p}_{Y^*}(X) \leq q \mid X = \mathbf{x} \right] - q \right| dq \in [0, 1]. \tag{52}$$

Then the $(\epsilon, \delta)$-*PAIC* condition implies

$$\Pr_X\left[ \mathrm{err}(X) \leq \epsilon \right] \geq 1 - \delta. \tag{53}$$

We use the RKHS representation of the KCE (Widmann et al., 2019; Marx et al., 2023). In the *top-label* setting, we take the conditioning variable to be

$$Z \equiv \hat{p}_{Y^*}(X) \in [0, 1]. \tag{54}$$

Let $Q \sim \mathcal{U}(0, 1)$ be independent of $(X, Y)$, and define the calibration residual

$$r(z, q) := \Pr\left[ Z \leq q \mid Z = z \right] - q \in [-1, 1]. \tag{55}$$

Let $k$ be a reproducing kernel on $[0, 1] \times [0, 1]$ with feature map $\psi(\cdot)$, and assume that $k$ is bounded from above:

$$\sup_{(z,q) \in [0,1]^2} k\big((z, q), (z, q)\big) \leq \kappa^2. \tag{56}$$

Then the population KCE admits the RKHS form

$$\mathrm{KCE}[k, \mathbf{H}] = \left\| \mathbb{E}_{Z,Q}\big[ \psi(Z, Q)\, r(Z, Q) \big] \right\|_{\mathcal{H}_k}. \tag{57}$$

By Jensen's inequality and the Cauchy–Schwarz inequality in the RKHS, we have

$$\mathrm{KCE}[k, \mathbf{H}] \leq \left( \mathbb{E}_{Z,Q} \|\psi(Z, Q)\|_{\mathcal{H}_k}^2 \right)^{1/2} \left( \mathbb{E}_{Z,Q} r(Z, Q)^2 \right)^{1/2}. \tag{58}$$

By the boundedness of $k$, $\|\psi(z, q)\|_{\mathcal{H}_k}^2 = k((z, q), (z, q)) \leq \kappa^2$, hence $\|\psi(Z, Q)\|_{\mathcal{H}_k} \leq \kappa$ almost surely, and therefore

$$\mathrm{KCE}[k, \mathbf{H}] \leq \kappa \left( \mathbb{E}_{Z,Q} r(Z, Q)^2 \right)^{1/2}. \tag{59}$$

Since $|r(z,q)| \leq 1$, we have $r(z,q)^2 \leq |r(z,q)|$, and thus

$$\mathbb{E}_Q\left[r(z,Q)^2\right] \leq \mathbb{E}_Q\left[|r(z,Q)|\right] = \int_0^1 \left|\Pr[Z \leq q \mid Z = z] - q\right| dq := \text{err}(z). \tag{60}$$

Taking expectation over $Z$ yields

$$\mathbb{E}_{Z,Q}r(Z,Q)^2 \leq \mathbb{E}_Z[\text{err}(Z)]. \tag{61}$$

Consequently,

$$\text{KCE}[k, \mathbf{H}] \leq \kappa\sqrt{\mathbb{E}_Z[\text{err}(Z)]}. \tag{62}$$

Finally, since $\text{err}(\cdot) \in [0,1]$ and $\Pr[\text{err}(Z) \leq \epsilon] \geq 1 - \delta$, we can split over the event $\mathbb{I}\{\text{err}(Z) \leq \epsilon\}$ and its complement to obtain

$$\mathbb{E}_Z[\text{err}(Z)] \leq (1-\delta)\epsilon + \delta \cdot 1 = \epsilon + \delta(1-\epsilon) \leq \epsilon + \delta. \tag{63}$$

Plugging this into (62) yields the population bound

$$\text{KCE}[k, \mathbf{H}] \leq \kappa\sqrt{\epsilon + \delta} \leq \kappa\sqrt{\epsilon} + \kappa\sqrt{\delta}. \tag{64}$$

**Finite-sample concentration.** We combine the above with the concentration bound in Widmann et al. (2019) for the (biased) empirical KCE. In particular, there exists a kernel-dependent constant $B_{p,q} > 0$ such that for any $t > 0$,

$$\Pr\left[\left|\widehat{\text{KCE}}[k, \mathbf{H}, \mathcal{D}] - \text{KCE}[k, \mathbf{H}]\right| \geq 2\sqrt{\frac{B_{p,q}}{n}} + t\right] \leq \exp\left(-\frac{nt^2}{2B_{p,q}}\right). \tag{65}$$

Here, $t > 0$ is a deviation parameter: increasing $t$ tightens the tail probability on the right-hand side. To obtain a $(1 - \alpha)$ high-probability bound, we choose $t$ such that $\exp\left(-\frac{nt^2}{2B_{p,q}}\right) = \alpha$, i.e.,

$$t := \sqrt{\frac{2B_{p,q}}{n}\log\left(\frac{1}{\alpha}\right)}. \tag{66}$$

Then, with probability at least $1 - \alpha$,

$$\left|\widehat{\text{KCE}}[k, \mathbf{H}, \mathcal{D}] - \text{KCE}[k, \mathbf{H}]\right| \leq 2\sqrt{\frac{B_{p,q}}{n}} + \sqrt{\frac{2B_{p,q}}{n}\log\left(\frac{1}{\alpha}\right)}. \tag{67}$$

Combining with the population bound $\text{KCE}[k, \mathbf{H}] \leq \kappa\sqrt{\epsilon + \delta} \leq \kappa\sqrt{\epsilon} + \kappa\sqrt{\delta}$ yields that, with probability at least $1 - \alpha$,

$$\widehat{\text{KCE}}[k, \mathbf{H}, \mathcal{D}] \leq \underbrace{\kappa}_{a_k}\sqrt{\epsilon} + \underbrace{\kappa\sqrt{\delta} + 2\sqrt{\frac{B_{p,q}}{n}} + \sqrt{\frac{2B_{p,q}}{n}\log\left(\frac{1}{\alpha}\right)}}_{b_{k,n}(\alpha)}. \tag{68}$$

Equivalently,

$$\Pr\left[\widehat{\text{KCE}}[k, \mathbf{H}, \mathcal{D}] \geq a_k\sqrt{\epsilon} + b_{k,n}(\alpha)\right] \leq \alpha. \tag{69}$$

Choosing a vanishing sequence $\alpha = \alpha_n$ (e.g., $\alpha_n = 1/n$) yields $\alpha_n = o_n(1)$ and, since

$$b_{k,n}(\alpha) = \kappa\sqrt{\delta} + 2\sqrt{\frac{B_{p,q}}{n}} + \sqrt{\frac{2B_{p,q}}{n}\log\left(\frac{1}{\alpha}\right)},$$

we have $b_{k,n}(\alpha_n) \to 0$ as $n \to \infty$ and in fact $b_{k,n}(\alpha_n) = O(n^{-1/2})$ up to logarithmic factors. Finally, taking a union bound to the *PAIC* failure event (whose probability is at most $\delta$) gives

$$\Pr\left[\widehat{\text{KCE}}[k, \mathbf{H}, \mathcal{D}] \geq a_k\sqrt{\epsilon} + b_{k,n}(\alpha_n)\right] \leq \delta + o_n(1), \tag{70}$$

where for bounded kernels with $\sup_z k(z,z) \leq \kappa^2$ one can take $a_k = \kappa$, and $b_{k,n}(\alpha_n) \to 0$ as $n \to \infty$ with the rate $O(n^{-1/2})$ up to logarithmic factors. $\qquad\square$

*Table 2.* Summary of dataset statistics: $n$ denotes the total number of samples, $d$ represents the feature dimension, and $m$ indicates the number of classes.

| Dataset | $n$ | $d$ | $m$ |
|---|---|---|---|
| **Type I – Tabular Data** | | | |
| BREAST-CANCER | 569 | 30 | 2 |
| HEART-DISEASE | 921 | 23 | 5 |
| ONLINE-SHOPPERS | 12,330 | 28 | 2 |
| DRY-BEAN | 13,612 | 16 | 7 |
| ADULT | 32,561 | 104 | 2 |
| **Type II – Image Data** | | | |
| CIFAR-10 | 60,000 | $32 \times 32 \times 3$ | 10 |
| CIFAR-100 | 60,000 | $32 \times 32 \times 3$ | 100 |
| **Type III – Text Data** | | | |
| SST-2 | 67,349 | - | 2 |
| SST-5 | 11,855 | - | 5 |

## B. Experimental Details

This section provides additional details on the experiments conducted. The experiments were implemented using the PyTorch framework. Detailed information of the datasets, metrics, baselines and implementation details can be found in Appendix B.1, Appendix B.2, and Appendix B.3, respectively.

**Hardware.** All experiments were run on a Linux server with an NVIDIA RTX 5000 Ada GPU (32 GB VRAM) and an Intel Xeon Gold 5320 CPU (12 cores) at 2.20 GHz.

### B.1. Datasets

We follow the evaluation protocol of Marx et al. (2023) and consider a diverse suite of benchmarks spanning tabular, image, and text modalities. All experiments are repeated over 5 random train/validation/test splits, using 70%/10%/20% for training/validation/testing, where the validation split is used for early stopping. Table 2 reports summary statistics of all the used datasets.

**Tabular Datasets (UCI)** We consider five UCI classification datasets, selected following Marx et al. (2023).

BREAST-CANCER (Wolberg et al., 1995): This dataset is widely used for binary classification tasks in medical diagnostics. The features ($d = 30$) are computed from a digitized image of a fine needle aspirate (FNA) of a breast mass. They describe characteristics of the cell nuclei present in the image, such as radius, texture, perimeter, and smoothness. The target variable distinguishes between benign and malignant tumors.

HEART-DISEASE (Janosi et al., 1988): Aggregated from the Cleveland database and other sources, this dataset contains physiological and demographic attributes of patients, including age, sex, chest-pain type, resting blood pressure, and serum cholesterol. The classification task involves predicting the presence and severity of heart disease, categorized into 5 classes (from no disease to four levels of severity).

ONLINE-SHOPPERS (Sakar & Kastro, 2018): This dataset consists of feature vectors belonging to 12,330 online sessions. The input features ($d = 28$) include administrative, informational, and product-related metrics (*e.g.*, bounce rates, exit rates, page values), as well as temporal data (special days, month). The objective is a binary classification task to predict whether a user's browsing session will result in a purchase transaction.

DRY-BEAN (Koklu & Ozkan, 2020): This dataset was created using high-resolution images of seven registered dry bean varieties. The features ($d = 16$) are geometric form factors extracted from the images, such as area, perimeter, major axis length, and compactness. The task is a 7-class classification problem to identify the bean species (*e.g.*, Seker, Barbunya, Bombay).

ADULT (Becker & Kohavi, 1996): A classic dataset used for algorithmic fairness and classification tasks. It leverages census

data to predict whether an individual's annual income exceeds $50,000$. While the raw data contains categorical variables (*e.g.*, education, occupation, marital status), the feature dimensionality $d = 104$ indicates that the categorical attributes have been pre-processed (*e.g.*, via one-hot encoding) for model training.

**Image Datasets (CIFAR)**    For the image datasets, we utilize two standard computer vision benchmarks from the CIFAR (Canadian Institute for Advanced Research, Krizhevsky et al. 2009).

CIFAR-10: This dataset consists of 60,000 color images with a resolution of $32 \times 32$ pixels. It covers 10 mutually exclusive classes: airplanes, automobiles, birds, cats, deer, dogs, frogs, horses, ships, and trucks. It serves as a standard benchmark for evaluating the performance of Convolutional Neural Networks (CNNs).

CIFAR-100: Similar to CIFAR-10 in terms of image size and total sample count, this dataset is significantly more challenging due to its fine-grained nature. It contains 100 classes containing 600 images each. These classes are grouped into 20 superclasses (*e.g.*, the "fish" superclass includes aquarium fish, flatfish, ray, shark, and trout).

**Text Datasets (SST)**    For the text datasets, we employ variants of the Stanford Sentiment Treebank (SST, Socher et al. 2013), a corpus of movie reviews.

SST-2: This version focuses on binary sentiment classification. It contains movie reviews labeled as either positive or negative, with neutral reviews removed to ensure a clear polarity. It is a standard component of the GLUE benchmark for evaluating sentence-level understanding.

SST-5: This dataset provides a more granular analysis of sentiment. Rather than a simple binary split, the target labels encompass 5 classes: very negative, negative, neutral, positive, and very positive. This fine-grained classification tests the model's ability to capture subtle nuances in language.

## B.2. Metrics

We evaluate classifiers from both *predictive performance* and *probabilistic reliability* perspectives. Below we summarize all reported metrics and the exact computation protocol used throughout the paper.

***Predictive Accuracy* Metrics.**    We report **Accuracy (ACC, Makridakis 1993)**, **Area Under the ROC Curve (AUC, Lobo et al. 2008)**, and **Average Precision (AP, Zhu 2004)**. Given a test set $\{(\mathbf{x}_i, y_i)\}_{i=1}^{N}$ with $C$ classes and predicted probabilities $\hat{\mathbf{p}}(\mathbf{x}_i) = [\hat{p}_{i1}, \dots, \hat{p}_{iC}]$, the predicted label is $\hat{y}_i := \arg\max_{c \in \{1,\dots,C\}} \hat{p}_{ic}$. Accuracy is defined as

$$\text{ACC} := \frac{1}{N} \sum_{i=1}^{N} \mathbb{I}\{\hat{y}_i = y_i\}. \tag{71}$$

To account for class imbalance, we also report AUC and AP. For binary classification ($C = 2$), AUC and AP are calculated using the positive-class probability $\hat{p}_{i2}$. For multi-class tasks ($C > 2$), we adopt a *One-vs-Rest (OvR)* scheme and report *macro-averaged* scores: we binarize labels for each class $c$ and compute the AUC/AP for predicting $Y = c$ *versus* $Y \neq c$, then average over $c = 1, \dots, C$. This macro-averaging assigns equal weight to each class and prevents the evaluation from being dominated by majority classes.

***Average-level Calibration* (ECE).**    We use the **Expected Calibration Error (ECE, Naeini et al. 2015)** as the standard average-level calibration metric. Let $\hat{p}_{i,y_i^*}$ denote the *top-label confidence*, where $y_i^* := \arg\max_c \hat{p}_{ic}$. ECE partitions the interval $[0, 1]$ into $B$ bins $\{I_b\}_{b=1}^{B}$ and measures the discrepancy between empirical accuracy and average confidence within each bin:

$$\text{ECE} := \sum_{b=1}^{B} \frac{|S_b|}{N} \left| \text{acc}(S_b) - \text{conf}(S_b) \right|, \tag{72}$$

where $S_b := \{i : \hat{p}_{i,y_i^*} \in I_b\}$, $\text{acc}(S_b) := \frac{1}{|S_b|} \sum_{i \in S_b} \mathbb{I}\{\hat{y}_i = y_i\}$, and $\text{conf}(S_b) := \frac{1}{|S_b|} \sum_{i \in S_b} \hat{p}_{i,y_i^*}$. We use $B = 20$ bins and the $\ell_1$-ECE variant in all experiments.

***Group-level Calibration* (GroupX_ECE and GroupY_ECE).**    To assess calibration robustness across subpopulations, we report two worst-group variants of ECE. First, **GroupY_ECE** measures calibration separately per class label: for each class

$c$, define the group $G_c := \{i : y_i = c\}$ and compute $\mathrm{ECE}(G_c)$ on this subset. We then report the worst-case value

$$\mathrm{GroupY\_ECE} := \max_{c \in \{1, \ldots, C\}} \mathrm{ECE}(G_c). \tag{73}$$

Second, **GroupX_ECE** measures calibration across groups defined over features. For each pair of input dimensions $(j, k)$, we split the test set by median thresholding into four groups

$$
\begin{aligned}
G_{jk}^{++} &= \{i : x_{ij} > \mathrm{med}_j,\ x_{ik} > \mathrm{med}_k\}, \\
G_{jk}^{+-} &= \{i : x_{ij} > \mathrm{med}_j,\ x_{ik} \le \mathrm{med}_k\}, \\
G_{jk}^{-+} &= \{i : x_{ij} \le \mathrm{med}_j,\ x_{ik} > \mathrm{med}_k\}, \\
G_{jk}^{--} &= \{i : x_{ij} \le \mathrm{med}_j,\ x_{ik} \le \mathrm{med}_k\},
\end{aligned}
\tag{74}
$$

where $\mathrm{med}_j$ is the median of feature $j$ on the test set. We keep only groups whose size lies in $[0.25N, 0.75N]$ to avoid extremely small or degenerate subsets, compute ECE on each retained group, and report the maximum:

$$\mathrm{GroupX\_ECE} := \max_{G \in \mathcal{G}_X} \mathrm{ECE}(G). \tag{75}$$

Both GroupX_ECE and GroupY_ECE are *worst-group* metrics, *i.e.*, lower values indicate more uniform reliability across heterogeneous subpopulations.

*Distribution-level Calibration* **(KCE).** To avoid binning artifacts and evaluate calibration from a *distribution matching* perspective, we report the **Kernel Calibration Error (KCE, Marx et al. 2023)**. KCE measures the discrepancy between the joint distributions of the target variable and a forecast variable, conditioned on a chosen *conditioning variable $Z$*. For multi-class classification, following Marx et al. (2023), we adopt the *canonical* conditioning choice

$$Z_i := \hat{\mathbf{p}}(\mathbf{x}_i), \tag{76}$$

*i.e.*, the predicted class-probability vector. Intuitively, *canonical calibration* requires that among samples with similar predicted distributions $\hat{\mathbf{p}}(\mathbf{x})$, the empirical label distribution matches the forecast distribution.

Formally, let $Y_i \in \{1, \ldots, C\}$ be the ground-truth label and define the forecast variable $\widehat{Y}_i$ as a randomized outcome drawn from the predictive distribution

$$\widehat{Y}_i \sim \mathrm{Categorical}\big(\hat{\mathbf{p}}(\mathbf{x}_i)\big). \tag{77}$$

Using the joint variables $(Y, Z)$ and $(\widehat{Y}, Z)$, KCE is defined as the Maximum Mean Discrepancy (MMD, Gretton et al. 2012) under a product kernel

$$k\big((y, z), (y', z')\big) := k_Y(y, y')\, k_Z(z, z'), \tag{78}$$

namely,

$$\mathrm{KCE} := \mathrm{MMD}\Big(P_{(Y,Z)},\ P_{(\widehat{Y},Z)}\Big). \tag{79}$$

In practice, we compute the empirical (biased) MMD estimator (Gretton et al., 2012). We use an RBF kernel for $k_Z$ on $Z$ and an RBF kernel (Scholkopf et al., 1997) on the label simplex for $k_Y$. Specifically, we embed labels by one-hot vectors $\mathbf{e}_y$ and represent forecasts by probability vectors $\hat{\mathbf{p}}(\mathbf{x})$. Let $K_Z \in \mathbb{R}^{N \times N}$ denote the kernel matrix on $\{Z_i\}_{i=1}^N$ and let $K_Y^{(\mathrm{true})}, K_Y^{(\mathrm{pred})}, K_Y^{(\mathrm{cross})} \in \mathbb{R}^{N \times N}$ be the label-space kernel matrices computed from one-hot targets, predicted probabilities, and their cross-term, respectively. Then the empirical KCE is computed as

$$\widehat{\mathrm{KCE}}^2 = \Big\langle K_Y^{(\mathrm{true})}, K_Z \Big\rangle + \Big\langle K_Y^{(\mathrm{pred})}, K_Z \Big\rangle - 2\Big\langle K_Y^{(\mathrm{cross})}, K_Z \Big\rangle, \tag{80}$$

where $\langle A, B \rangle := \frac{1}{N^2} \sum_{i=1}^N \sum_{i'=1}^N A_{ii'} B_{ii'}$. We report

$$\widehat{\mathrm{KCE}} := \sqrt{\max\Big(0, \widehat{\mathrm{KCE}}^2\Big)} \tag{81}$$

for numerical stability. A lower KCE indicates closer matching between $(Y, Z)$ and $(\widehat{Y}, Z)$, hence better *distribution-level* calibration.

## B.3. Implementation Details

To comprehensively assess the performance of the proposed method $\mathbf{H}_{\text{ICALD}}$, we compare it to a diverse set of baselines ranging from standard Softmax classification to advanced distribution-matching techniques. All models were implemented using the PyTorch framework [1].

**Backbone Architecture**    To handle diverse data modalities effectively, we employ specialized backbone architectures as feature extractors for tabular, image, and text inputs.

- **Tabular Data:** For all tabular datasets, we employ a **Multi-Layer Perceptron (MLP)** with residual connections. The architecture consists of three fully connected layers with ReLU activation functions. The width of the hidden layers ($d_{hidden}$) is determined heuristically based on the input dimension $d_{in}$ (*e.g.*, $d_{hidden} \approx 1.6 \times d_{in}$ for medium-dimensional data) to ensure sufficient capacity without overfitting. We apply Dropout with a rate of $0.5$ after each activation to encourage robustness.

- **Image Data:** For vision benchmarks, we use a ResNet backbone (He et al., 2016), specifically **ResNet-50**. For small-resolution inputs (*e.g.*, $32 \times 32$ in CIFAR), we adopt a CIFAR-style stem by replacing the initial $7 \times 7$ convolution (stride 2) with a $3 \times 3$ convolution (stride 1) and removing the initial max-pooling layer, avoiding overly aggressive early downsampling. Following the final residual stage, we apply global average pooling to obtain a 2048-dimensional representation, which is then projected to the target hidden dimension via a linear layer.

- **Text Data:** For natural language processing tasks, we leverage the pre-trained **BERT-base-uncased** model (Devlin et al., 2019). We extract the embedding of the special '[CLS]' token from the last hidden layer (dimension 768) to serve as the aggregate sentence representation. This representation is then passed through a fully connected projection layer with ReLU activation to map it into the shared latent space.

**Optimization and Hyperparameters** All models are trained using the Adam optimizer (Kingma, 2014) with a learning rate of $\eta = 10^{-3}$ and weight decay of $10^{-4}$. We utilize a batch size of 128 across all experiments. To prevent overfitting and ensure fair comparison, we employ an *early stopping* strategy with a patience of 50 epochs.

$\mathbf{H}_{\textbf{Softmax}}$, $\mathbf{H}_{\textbf{ALD}}$ **and other distribution-based classifiers** To ensure a rigorous and fair comparison, all classifier variants investigated in this study—including the standard baseline $\mathbf{H}_{\text{Softmax}}$, our proposed $\mathbf{H}_{\text{ALD}}$, and the alternative distribution-based classifiers defined in Appendix C.1—share an identical backbone architecture and are optimized using the same Negative Log-Likelihood (NLL) loss function, as formulated in (4). Consistent with the generalized framework described in the main text, the procedure for deriving the final categorical probability distribution $\hat{\mathbf{p}}(\mathbf{x})$ remains identical across all variants. Specifically, each classifier employs the same normalization mechanism defined in (2) and (3), differing solely in the specific Cumulative Distribution Function (CDF) utilized to compute the unnormalized class scores.

For $\mathbf{H}_{\text{ICALD}}$, in all experiments we fix $\beta = 0.1$ in (12). To approximate the expectation over the random anchor $q \sim \mathcal{U}(0, 1)$, we use a **mini-batch Monte Carlo** strategy that samples **multiple anchors per input within each iteration**. Concretely, for each mini-batch $\{(\mathbf{x}_i, y_i)\}_{i=1}^{B}$ we draw $K$ *i.i.d.* anchors $\{q^{(j)}\}_{j=1}^{K}$ and form $BK$ augmented pairs $(\mathbf{x}_i, q^{(j)})$ by replicating the batch $K$ times. We then average the confidence-alignment penalty across these $BK$ instances,

$$\frac{1}{BK} \sum_{j=1}^{K} \sum_{i=1}^{B} \left| \hat{p}_{i,y_i^*}(\mathbf{x}_i, q^{(j)}) - q^{(j)} \right|,$$

which is an unbiased Monte Carlo estimator of $\mathbb{E}_q[|\hat{p}_{y^*}(\mathbf{x}, q) - q|]$ for the current mini-batch. Importantly, sampling multiple anchors per input provides a *denser calibration signal per iteration*, which improves the stability and speed of learning the confidence-alignment objective and helps calibration keep pace with the rapidly improving discriminative fit *without requiring additional training epochs*.

We use task-specific mini-batch Monte Carlo with $K$ anchors per input (sampled within each iteration): $K{=}5$ for tabular datasets, $K{=}10$ for image datasets, and $K{=}15$ for text datasets. We apply early stopping on a held-out validation split and the training epochs are set as follows: BREAST-CANCER: 400, HEART-DISEASE: 600, ONLINE-SHOPPERS: 200, DRY-BEAN: 500, ADULT: 100, CIFAR-10: 150, CIFAR-100: 250, SST-2: 50, SST-5: 50.

---

[1] https://pytorch.org/

*Table 3.* Comparison of Calibration Paradigms.

| Method | Pre-calibration | Post-calibration |
|---|:---:|:---:|
| **Temperature Scaling** (Guo et al., 2017) | ✗ | ✓ |
| **Histogram Binning** (Zadrozny & Elkan, 2001) | ✗ | ✓ |
| **ALD-MMD** (Section 2.3) | ✓ | ✗ |
| **ICALD** (Section 2.4) | ✓ | ✓ |

As summarized in Table 3, we rigorously compare our proposed framework against established calibration baselines, categorized into *post-calibration* and *pre-calibration* approaches.

**Post-calibration Baselines** These methods are applied after the base model has been trained and fixed.

- **Temperature Scaling (TS, Guo et al. 2017):** We implement TS as a parametric post-processing step. After training the base classifier, we freeze the network parameters and optimize a single scalar temperature $T > 0$ to minimize the Negative Log-Likelihood (NLL) on the validation set. This effectively rescales the logits to soften (or sharpen) the output distribution without affecting classification accuracy.

- **Histogram Binning (HB, Zadrozny & Elkan 2001):** We employ Histogram Binning with $M = 20$ bins to partition the prediction space. For each bin, the calibrated probability is assigned as the empirical accuracy of the validation samples falling within that bin, providing a non-parametric correction to the confidence estimates.

**Pre-calibration Baselines** We consider *pre-calibration* baselines that directly regularize training to improve probability calibration, rather than applying a post-hoc calibrator at inference time. Following the kernel calibration view of Marx et al. (2023), we encourage calibration by matching the *target* joint law $(Y, Z)$ with the *forecast* joint law $(\widehat{Y}, Z)$ using an integral probability metric (IPM) with an RKHS witness class. With $Z \equiv \hat{\mathbf{p}}(X)$, this yields a squared maximum mean discrepancy (MMD$^2$), equivalently the squared kernel calibration error (SKCE), computed with a factorized kernel $k\big((y, z), (y', z')\big) = k_Y(y, y') \, k_Z(z, z')$. We use $k_Y(y, y') = \mathbb{I}\{y = y'\}$ and an RBF kernel (Scholkopf et al., 1997),

$$k_Z(z, z') = \exp\left(-\tfrac{\|z - z'\|_2^2}{2\tau^2}\right), \qquad \tau > 0, \tag{82}$$

with the bandwidth $\tau$ chosen by the median heuristic (Gretton et al., 2012).

$\mathbf{H}_{\textbf{ALD+MMD}}$ and $\mathbf{H}_{\textbf{Softmax+MMD}}$. For any base classifier $\mathbf{H}$ (*e.g.*, $\mathbf{H}_{\text{ALD}}$ and $\mathbf{H}_{\text{Softmax}}$), we train with the objective (also see (11))

$$\mathcal{L}_{\text{NLL}} \; + \; \alpha \, \mathcal{L}_{\text{SMMD}}, \tag{83}$$

where $\mathcal{L}_{\text{SMMD}}$ is the unbiased U-statistic estimator of $\widehat{\text{MMD}}^2$ between $(Y, Z)$ and $(\widehat{Y}, Z)$. This penalty decreases the empirical KCE and thus promotes the *Probably Approximately Calibrated Classifier (PACC)* property under the chosen kernel. In all experiments, we set $\alpha = 0.1$ for these *pre-calibration* baselines.

# C. Additional Results

This appendix collects additional empirical results that complement the main text, further validate our modeling choices, and provide stronger evidence of robustness across datasets, metrics, and experimental settings. Appendix C.1 formally defines the probability distributions employed in our framework and conducts an ablation study on the hyperparameter $t_0$ to determine the optimal setting for each distribution-based classifier. Appendix C.2 presents a comprehensive comparison between the proposed $\mathbf{H}_{\text{ALD}}$ and the standard Softmax baseline, as well as other distribution families. We analyze their trade-offs in terms of predictive accuracy, calibration quality, and computational complexity, highlighting the advantages of $\mathbf{H}_{\text{ALD}}$ in modeling heteroscedasticity and asymmetry. Appendix C.3 investigates the impact of the *Decoupled Dual-Stream Optimization (DDSO) strategy*, validating its effectiveness in balancing discriminative learning and calibration updates. Appendix C.4 reports the full experimental results comparing our $\mathbf{H}_{\text{ICALD}}$ framework against strong post-hoc and pre-hoc calibration methods across all datasets and metrics. Appendix C.5 provides an expanded comparison with additional calibration and uncertainty-estimation baselines, showing that $\mathbf{H}_{\text{ICALD}}^{\text{Pre}}$ remains competitive against recent calibration-aware training objectives, Dirichlet calibration, evidential deep learning, and Natural Posterior Networks. Appendix C.6 presents case studies that further examine the fixed-scale bias of Softmax, parameter sensitivity, *DDSO* gradient routing, and robustness under corruption-based covariate shift.

## C.1. Hyperparameter $t_0$ for different distribution-based classifiers

In this section, we first provide the definitions for the various probability distributions employed in our proposed classifiers. Subsequently, we present an ablation study on the choice of the fixed scalar threshold $t_0$, which determines the evaluation point for the cumulative distribution function (CDF) in our framework. We consider six parametric distributions: Asymmetric Laplace Distribution (ALD), Normal (Gaussian) Distribution, Logistic Distribution, Exponential Distribution, Log-Normal Distribution, and Weibull Distribution. Their probability density functions (PDFs) are defined as follows.

**Definition C.1** (**Asymmetric Laplace Distribution, Kotz et al. 2012**). A random variable $Y$ is said to have an asymmetric Laplace distribution with parameters $(\theta, \sigma, \kappa)$, denoted $Y \sim \mathcal{AL}(\theta, \sigma, \kappa)$, if its PDF is:

$$f_{\text{ALD}}(y; \theta, \sigma, \kappa) = \frac{\sqrt{2}}{\sigma} \frac{\kappa}{1 + \kappa^2} \begin{cases} \exp\left(\frac{\sqrt{2}\kappa}{\sigma}(\theta - y)\right), & \text{if } y \geq \theta, \\ \exp\left(\frac{\sqrt{2}}{\sigma\kappa}(y - \theta)\right), & \text{if } y < \theta, \end{cases} \tag{84}$$

where $\theta \in \mathbb{R}$, $\sigma > 0$, and $\kappa > 0$, are the *location*, *scale* and *asymmetry* parameters.

**Definition C.2** (**Normal Distribution, Norton et al. 2021**). A random variable $Y$ is said to have a Normal distribution with parameters $(\mu, \sigma)$, denoted $Y \sim \mathcal{N}(\mu, \sigma)$, if its PDF is:

$$f_{\text{Norm}}(y; \mu, \sigma) = \frac{1}{\sigma\sqrt{2\pi}} \exp\left(-\frac{(y - \mu)^2}{2\sigma^2}\right), \tag{85}$$

where $\mu \in \mathbb{R}$ and $\sigma > 0$ are the *location (mean)* and *scale (standard deviation)* parameters, respectively.

**Definition C.3** (**Logistic Distribution, Norton et al. 2021**). A random variable $Y$ is said to have a Logistic distribution with parameters $(\mu, s)$, denoted $Y \sim \text{Logistic}(\mu, s)$, if its PDF is:

$$f_{\text{Logistic}}(y; \mu, s) = \frac{\exp\left(-\frac{y - \mu}{s}\right)}{s\left(1 + \exp\left(-\frac{y - \mu}{s}\right)\right)^2}, \tag{86}$$

where $\mu \in \mathbb{R}$ is the *location* parameter and $s > 0$ is the *scale* parameter.

**Definition C.4** (**Exponential Distribution, Norton et al. 2021**). A random variable $Y$ is said to have an Exponential distribution with parameter $\lambda$, denoted $Y \sim \text{Exp}(\lambda)$, if its PDF is:

$$f_{\text{Exp}}(y; \lambda) = \begin{cases} \lambda \exp(-\lambda y), & \text{if } y \geq 0, \\ 0, & \text{if } y < 0, \end{cases} \tag{87}$$

where $\lambda > 0$ is the *rate* parameter.

**Definition C.5** (**LogNormal Distribution, Norton et al. 2021**). A random variable $Y$ is said to have a LogNormal distribution with parameters $(\mu, \sigma)$, denoted $Y \sim \text{Lognormal}(\mu, \sigma)$, if its PDF is:

$$f_{\text{LogNorm}}(y; \mu, \sigma) = \begin{cases} \frac{1}{y\sigma\sqrt{2\pi}} \exp\left(-\frac{(\ln y - \mu)^2}{2\sigma^2}\right), & \text{if } y > 0, \\ 0, & \text{if } y \leq 0, \end{cases} \tag{88}$$

where $\mu \in \mathbb{R}$ and $\sigma > 0$ are the parameters of the underlying normal distribution.

**Definition C.6** (**Weibull Distribution, Norton et al. 2021**). A random variable $Y$ is said to have a Weibull distribution with parameters $(\lambda, k)$, denoted $Y \sim \text{Weibull}(\lambda, k)$, if its PDF is:

$$f_{\text{Weibull}}(y; \lambda, k) = \begin{cases} \frac{k}{\lambda}\left(\frac{y}{\lambda}\right)^{k-1} \exp\left(-\left(\frac{y}{\lambda}\right)^k\right), & \text{if } y \geq 0, \\ 0, & \text{if } y < 0, \end{cases} \tag{89}$$

where $\lambda > 0$ is the *scale* parameter and $k > 0$ is the *shape* parameter.

**Impact of Threshold $t_0$.** The threshold parameter $t_0$ plays a critical role in our framework, acting as the fixed evaluation point for the class-conditional CDFs. The domain of feasible $t_0$ values depends on the support of the chosen distribution. Specifically:

- For distributions with support on the entire real line ($\mathbb{R}$), including $\mathbf{H}_{\text{ALD}}$, $\mathbf{H}_{\text{Norm}}$, and $\mathbf{H}_{\text{Logistic}}$, we evaluate $t_0$ on the grid $\{-2.0, -1.0, -0.5, 0.0, 0.5, 1.0, 2.0\}$.

- For distributions with support on positive reals ($[0, \infty)$ or $(0, \infty)$), including $\mathbf{H}_{\text{Exp}}$, $\mathbf{H}_{\text{LogNorm}}$, and $\mathbf{H}_{\text{Weibull}}$, we restrict the search grid to positive values $\{0.01, 0.1, 0.5, 1.0, 2.0, 4.0\}$.

**Optimal Selection Analysis.** Table 4 summarizes the average ranking of each candidate $t_0$ in nine datasets (BREAST-CANCER, HEART-DISEASE, ONLINE-SHOPPERS, DRY-BEAN, ADULT, CIFAR-10, CIFAR-100, SST-2 and SST-5). The ranking is computed based on seven performance metrics (Accuracy, AUC, AP, ECE, GroupX_ECE, GroupY_ECE, and KCE), where a lower average rank indicates superior overall performance.

- **Real-Valued Distributions:** For $\mathbf{H}_{\text{Norm}}$ and $\mathbf{H}_{\text{Logistic}}$, a negative threshold $t_0 = -1.0$ yields the best overall performance. Interestingly, for the $\mathbf{H}_{\text{ALD}}$, a slightly positive threshold of $t_0 = 0.5$ achieves the lowest average rank (3.11), suggesting that the flexibility of the asymmetry parameter $\kappa$ allows the model to adapt well to a non-centered threshold.

- **Positive-Support Distributions:** Among the distributions restricted to positive supports, $\mathbf{H}_{\text{LogNorm}}$ and $\mathbf{H}_{\text{Weibull}}$ favor a small threshold of $t_0 = 0.1$. In contrast, $\mathbf{H}_{\text{Exp}}$ performs best with a larger threshold of $t_0 = 2.0$. This divergence indicates that the heavy-tailed nature of the Exponential distribution may require a larger evaluation point to effectively discriminate between classes compared to the more flexible shape configurations of the Weibull and Log-Normal distributions.

Based on these empirical findings, we adopt the $t_0$ values with the lowest average ranks (highlighted in Table 4) as the default hyperparameters for our comparative experiments in the main text.

*Table 4.* Ablation study on the choice of the threshold $t_0$ for different distribution-based classifiers. We report the average rank of each candidate $t_0$ aggregated over all datasets and evaluation metrics to identify the most robust setting for each distribution. The optimal $t_0$ achieving the lowest average rank is highlighted.

| Method | $t_0$ | ACC | AUC | AP | ECE | GroupX_ECE | GroupY_ECE | KCE | Avg Rank |
|--------|-------|-----|-----|-----|-----|------------|------------|-----|----------|
| $\mathbf{H}_{\text{ALD}}$ | −2.0 | 4.67 | 5.56 | 4.44 | 3.89 | 5.56 | 4.67 | 3.00 | 4.56 |
| | −1.0 | 4.89 | 4.44 | 3.89 | 3.89 | 3.89 | 3.22 | 4.89 | 4.11 |
| | −0.5 | 3.56 | 4.44 | 3.89 | 3.56 | 4.00 | 5.11 | 4.78 | 4.22 |
| | 0.0 | 3.78 | 2.89 | 2.67 | 4.11 | 3.78 | 3.00 | 3.67 | 3.44 |
| | **0.5** | 2.67 | 2.78 | 3.56 | 3.56 | 2.78 | 2.89 | 3.22 | **3.11** |
| | 1.0 | 2.67 | 4.44 | 5.11 | 4.33 | 4.00 | 3.22 | 4.56 | 4.00 |
| | 2.0 | 4.33 | 3.11 | 4.22 | 3.89 | 2.11 | 4.33 | 3.33 | 3.67 |
| $\mathbf{H}_{\text{Norm}}$ | −2.0 | 2.22 | 3.78 | 3.78 | 3.22 | 2.67 | 2.56 | 3.00 | 3.00 |
| | **−1.0** | 2.00 | 2.22 | 1.89 | 2.33 | 3.00 | 3.00 | 3.89 | **2.67** |
| | −0.5 | 4.56 | 4.22 | 3.56 | 2.89 | 3.11 | 4.56 | 3.33 | 3.78 |
| | 0.0 | 4.33 | 4.11 | 3.56 | 3.22 | 2.89 | 3.00 | 3.44 | 3.56 |
| | 0.5 | 5.56 | 5.00 | 4.22 | 4.44 | 4.33 | 4.78 | 4.11 | 4.67 |
| | 1.0 | 4.00 | 4.67 | 5.56 | 5.67 | 5.56 | 4.33 | 3.89 | 4.78 |
| | 2.0 | 2.89 | 3.22 | 4.78 | 6.44 | 5.56 | 5.67 | 5.56 | 4.89 |
| $\mathbf{H}_{\text{Logistic}}$ | −2.0 | 2.00 | 3.00 | 2.89 | 5.00 | 3.44 | 5.22 | 3.67 | 3.56 |
| | **−1.0** | 2.56 | 2.22 | 1.89 | 2.33 | 3.22 | 2.11 | 3.00 | **2.44** |
| | −0.5 | 3.11 | 3.56 | 3.00 | 4.00 | 3.11 | 3.78 | 3.56 | 3.44 |
| | 0.0 | 5.33 | 4.89 | 4.22 | 3.33 | 1.67 | 5.00 | 3.78 | 4.00 |
| | 0.5 | 5.00 | 4.67 | 5.67 | 4.00 | 4.22 | 2.78 | 4.44 | 4.44 |
| | 1.0 | 4.11 | 4.67 | 5.33 | 4.00 | 5.44 | 3.22 | 4.44 | 4.44 |
| | 2.0 | 3.44 | 5.00 | 4.44 | 5.00 | 6.00 | 5.56 | 4.67 | 4.89 |
| $\mathbf{H}_{\text{Exp}}$ | 0.01 | 2.67 | 3.11 | 3.89 | 4.44 | 4.11 | 5.11 | 4.22 | 3.89 |
| | 0.1 | 3.22 | 2.89 | 2.89 | 4.33 | 4.67 | 2.78 | 3.89 | 3.56 |
| | 0.5 | 2.00 | 3.67 | 3.67 | 3.11 | 3.44 | 5.11 | 4.44 | 3.67 |
| | 1.0 | 3.11 | 4.11 | 3.33 | 2.11 | 2.44 | 2.56 | 2.89 | 2.89 |
| | **2.0** | 3.11 | 1.89 | 2.22 | 2.89 | 2.00 | 2.78 | 2.22 | **2.44** |
| | 4.0 | 5.11 | 4.67 | 4.33 | 3.67 | 3.56 | 2.44 | 3.22 | 3.89 |
| $\mathbf{H}_{\text{LogNorm}}$ | 0.01 | 4.22 | 3.67 | 5.22 | 4.00 | 4.33 | 5.11 | 3.78 | 4.33 |
| | **0.1** | 1.56 | 3.44 | 3.11 | 2.00 | 2.89 | 2.22 | 2.78 | **2.56** |
| | 0.5 | 3.11 | 2.44 | 2.78 | 2.00 | 2.78 | 3.89 | 4.22 | 3.00 |
| | 1.0 | 3.33 | 3.22 | 2.78 | 2.56 | 2.44 | 2.78 | 2.44 | 2.78 |
| | 2.0 | 3.67 | 3.67 | 3.78 | 4.44 | 3.56 | 3.67 | 3.22 | 3.67 |
| | 4.0 | 3.56 | 3.56 | 3.56 | 5.56 | 4.22 | 3.56 | 4.22 | 4.00 |
| $\mathbf{H}_{\text{Weibull}}$ | 0.01 | 2.44 | 2.00 | 2.67 | 3.89 | 2.44 | 2.56 | 3.56 | 2.78 |
| | **0.1** | 2.67 | 2.44 | 1.67 | 3.67 | 3.78 | 2.67 | 2.11 | **2.67** |
| | 0.5 | 2.89 | 3.56 | 3.11 | 3.33 | 3.44 | 2.22 | 2.22 | 3.00 |
| | 1.0 | 1.89 | 3.44 | 3.78 | 2.11 | 2.44 | 3.44 | 3.00 | 2.89 |
| | 2.0 | 5.44 | 4.00 | 4.89 | 3.44 | 3.67 | 4.44 | 4.78 | 4.33 |
| | 4.0 | 4.89 | 5.00 | 5.00 | 5.00 | 5.00 | 6.00 | 5.22 | 5.11 |

## C.2. Comparison with Softmax and Distribution-based Classifiers

In this section, we analyze the trade-offs between the standard Softmax baseline ($\mathbf{H}_{\text{Softmax}}$) and other distribution-based classifiers. The comparison focuses on three key aspects: predictive *accuracy*, *calibration* quality, and model *complexity*. The detailed numerical results for all metrics are reported in Table 5, while the comprehensive statistical rankings are visualized in Figure 4. Additionally, the computational cost analysis is detailed in Table 6. Notably, it is important to distinguish between the ranking values reported in Table 5 and Figure 4. The "Avg Rank" in Table 5 represents the mean of rankings derived strictly from the 9 representative datasets listed in the table. In contrast, Figure 4 presents the Friedman average ranks calculated across the complete set of $N = 45$ experimental settings (comprising 9 datasets $\times$ 5 random seeds) to facilitate rigorous statistical hypothesis testing (*i.e.*, the Nemenyi test Nemenyi 1963). Although the absolute rank values differ due to the varying sample sizes and calculation scopes, the relative performance trends remain consistent, with $\mathbf{H}_{\text{ALD}}$ demonstrating dominance in both evaluations.

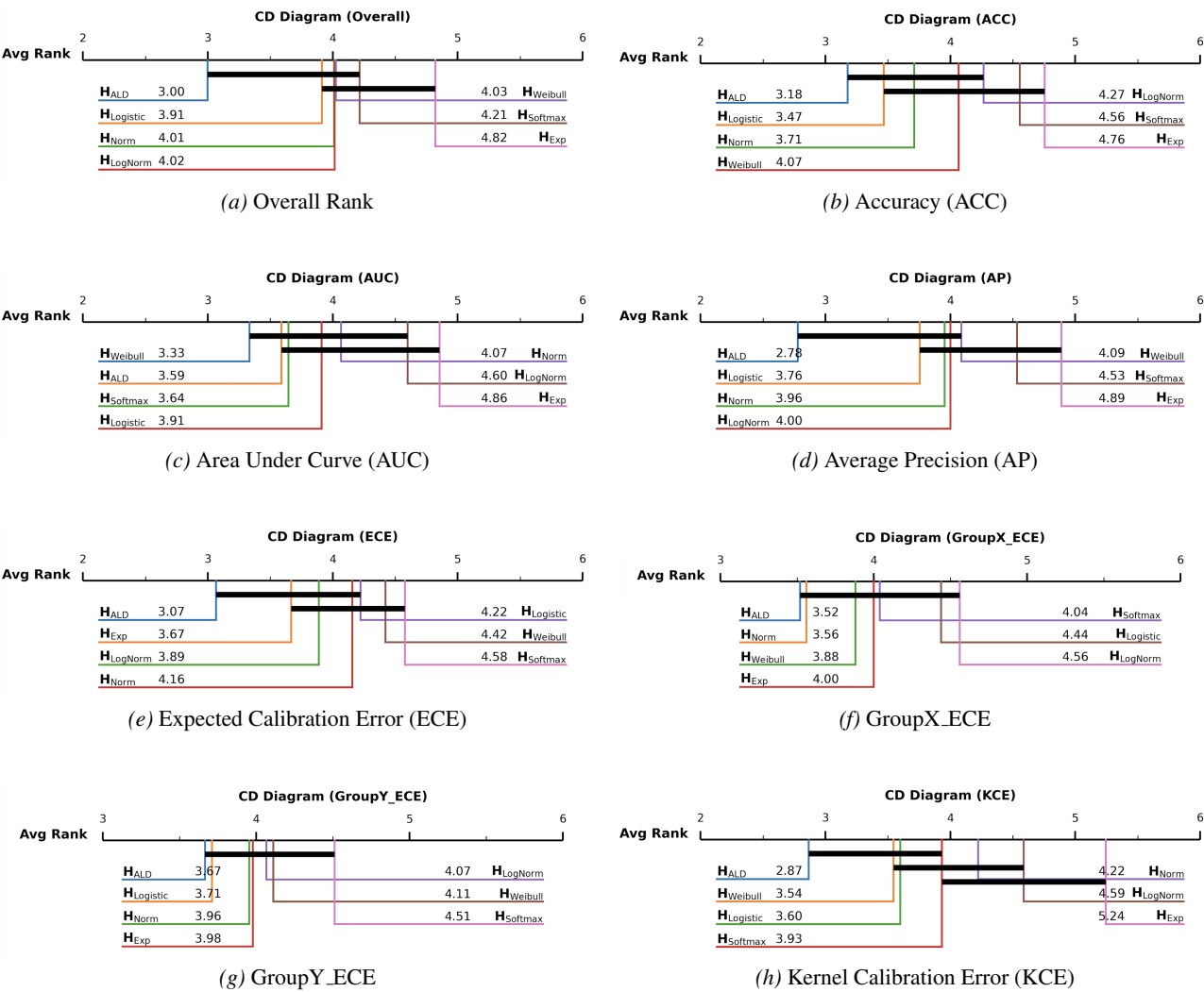

*Figure 4.* **Critical Difference (CD) diagrams comparing the average ranks of different classifiers across all evaluated metrics.** We present the detailed comparisons for: (a) Overall Rank, (b) Accuracy (ACC), (c) Area Under Curve (AUC), (d) Average Precision (AP), (e) Expected Calibration Error (ECE), (f) Group-X ECE, (g) Group-Y ECE, and (h) Kernel Calibration Error (KCE). The horizontal axis represents the average rank (lower/left is better). The critical difference (CD = 1.34) is calculated using the Nemenyi test (Nemenyi, 1963) at a significance level of $\alpha = 0.05$. Classifiers connected by a thick horizontal bar are not statistically significantly different. $\mathbf{H}_{\text{ALD}}$ demonstrates robust performance, achieving the best or competitive rankings across both discriminative and calibration metrics.

*Table 5.* The overall performance comparison of different classifiers. Best values in each column are highlighted in bold yellow.

| Dataset | Model | ACC | AUC | AP | ECE | GroupX_ECE | GroupY_ECE | KCE | Avg Rank |
|---|---|---|---|---|---|---|---|---|---|
| Breast-Cancer | $H_{Softmax}$ | 97.719 ± 0.702 | 99.286 ± 0.320 | 99.205 ± 0.384 | **0.030 ± 0.009** | 0.093 ± 0.023 | 0.060 ± 0.018 | 0.027 ± 0.005 | 4.93 |
| | $H_{ALD}$ | 98.070 ± 1.023 | 99.583 ± 0.340 | 99.429 ± 0.462 | 0.031 ± 0.013 | 0.090 ± 0.031 | 0.057 ± 0.024 | **0.026 ± 0.013** | 3.00 |
| | $H_{Norm}$ | **98.421 ± 1.023** | 99.643 ± 0.349 | 99.512 ± 0.377 | 0.038 ± 0.012 | 0.100 ± 0.029 | 0.063 ± 0.020 | 0.028 ± 0.015 | 4.71 |
| | $H_{Logistic}$ | 97.719 ± 1.190 | 99.550 ± 0.222 | 99.394 ± 0.340 | 0.031 ± 0.014 | 0.095 ± 0.041 | **0.053 ± 0.029** | 0.027 ± 0.016 | 4.29 |
| | $H_{Exp}$ | 97.368 ± 0.961 | 99.464 ± 0.400 | 99.280 ± 0.315 | **0.030 ± 0.009** | 0.093 ± 0.028 | 0.059 ± 0.023 | 0.028 ± 0.009 | 5.14 |
| | $H_{LogNorm}$ | **98.421 ± 1.023** | **99.709 ± 0.159** | **99.598 ± 0.278** | 0.031 ± 0.010 | **0.089 ± 0.023** | 0.057 ± 0.025 | 0.027 ± 0.008 | **2.14** |
| | $H_{Weibull}$ | 97.895 ± 1.053 | 99.590 ± 0.392 | 99.432 ± 0.321 | 0.032 ± 0.009 | 0.093 ± 0.028 | 0.057 ± 0.025 | 0.027 ± 0.008 | 3.79 |
| Heart-Disease | $H_{Softmax}$ | 59.333 ± 3.590 | 79.141 ± 1.374 | 41.130 ± 2.677 | 0.100 ± 0.020 | **0.315 ± 0.039** | 0.443 ± 0.097 | 0.295 ± 0.024 | 3.36 |
| | $H_{ALD}$ | **61.667 ± 3.162** | **80.046 ± 2.029** | 41.338 ± 2.967 | **0.097 ± 0.014** | 0.348 ± 0.022 | 0.503 ± 0.112 | **0.272 ± 0.023** | **2.36** |
| | $H_{Norm}$ | 58.333 ± 2.981 | 77.553 ± 2.606 | 40.310 ± 1.312 | 0.129 ± 0.042 | 0.387 ± 0.104 | 0.460 ± 0.080 | 0.278 ± 0.016 | 5.57 |
| | $H_{Logistic}$ | 59.000 ± 3.590 | 79.231 ± 1.201 | 41.338 ± 1.407 | 0.101 ± 0.005 | 0.370 ± 0.031 | 0.406 ± 0.055 | 0.308 ± 0.061 | 3.71 |
| | $H_{Exp}$ | 59.667 ± 2.667 | 79.132 ± 1.350 | **42.477 ± 0.921** | 0.100 ± 0.018 | 0.328 ± 0.013 | 0.475 ± 0.071 | 0.287 ± 0.013 | 3.36 |
| | $H_{LogNorm}$ | 60.000 ± 4.714 | 77.188 ± 2.603 | 39.436 ± 2.234 | 0.120 ± 0.008 | 0.386 ± 0.014 | 0.483 ± 0.066 | 0.298 ± 0.030 | 4.79 |
| | $H_{Weibull}$ | 59.667 ± 2.449 | 78.588 ± 1.521 | 40.763 ± 2.168 | 0.124 ± 0.024 | 0.356 ± 0.034 | **0.387 ± 0.045** | 0.293 ± 0.036 | 4.86 |
| Online-Shoppers | $H_{Softmax}$ | 89.749 ± 0.339 | 91.915 ± 0.534 | 70.777 ± 1.511 | 0.018 ± 0.002 | 0.055 ± 0.005 | 0.491 ± 0.010 | **0.211 ± 0.011** | 5.43 |
| | $H_{ALD}$ | 89.757 ± 0.326 | 91.830 ± 0.608 | 70.654 ± 2.239 | 0.016 ± 0.002 | 0.051 ± 0.002 | 0.489 ± 0.024 | **0.211 ± 0.015** | 4.86 |
| | $H_{Norm}$ | 89.805 ± 0.352 | **91.994 ± 0.615** | 70.856 ± 1.669 | **0.014 ± 0.002** | **0.050 ± 0.007** | 0.487 ± 0.015 | 0.222 ± 0.015 | **2.43** |
| | $H_{Logistic}$ | **89.911 ± 0.362** | 91.721 ± 0.696 | **71.136 ± 1.592** | 0.017 ± 0.003 | 0.056 ± 0.002 | **0.462 ± 0.007** | 0.225 ± 0.017 | 3.43 |
| | $H_{Exp}$ | 89.773 ± 0.342 | 91.768 ± 0.733 | 70.904 ± 2.139 | 0.018 ± 0.004 | 0.056 ± 0.009 | 0.490 ± 0.021 | 0.240 ± 0.007 | 5.29 |
| | $H_{LogNorm}$ | 89.805 ± 0.243 | 91.767 ± 0.682 | 69.851 ± 1.357 | 0.016 ± 0.003 | 0.055 ± 0.008 | 0.487 ± 0.017 | 0.229 ± 0.025 | 5.07 |
| | $H_{Weibull}$ | 89.886 ± 0.344 | 91.777 ± 0.687 | 70.898 ± 2.057 | 0.016 ± 0.002 | 0.054 ± 0.003 | 0.484 ± 0.010 | 0.222 ± 0.011 | 4.50 |
| Dry-Bean | $H_{Softmax}$ | 92.295 ± 0.383 | 99.438 ± 0.022 | 97.817 ± 0.099 | 0.017 ± 0.004 | 0.045 ± 0.007 | 0.083 ± 0.022 | 0.053 ± 0.002 | 6.21 |
| | $H_{ALD}$ | 92.472 ± 0.463 | 99.428 ± 0.069 | 97.902 ± 0.163 | 0.009 ± 0.004 | 0.032 ± 0.006 | 0.048 ± 0.006 | **0.040 ± 0.002** | 2.79 |
| | $H_{Norm}$ | 92.361 ± 0.262 | 99.436 ± 0.034 | 97.868 ± 0.086 | **0.008 ± 0.002** | 0.032 ± 0.002 | 0.051 ± 0.010 | 0.048 ± 0.004 | 4.00 |
| | $H_{Logistic}$ | **92.795 ± 0.260** | **99.499 ± 0.044** | **98.054 ± 0.100** | 0.009 ± 0.002 | **0.030 ± 0.004** | **0.039 ± 0.004** | **0.040 ± 0.003** | **1.86** |
| | $H_{Exp}$ | 92.413 ± 0.455 | 99.332 ± 0.044 | 97.677 ± 0.165 | 0.012 ± 0.004 | 0.036 ± 0.007 | 0.066 ± 0.015 | **0.040 ± 0.001** | 4.71 |
| | $H_{LogNorm}$ | 92.464 ± 0.383 | 99.442 ± 0.045 | 97.840 ± 0.138 | **0.008 ± 0.003** | 0.033 ± 0.004 | 0.052 ± 0.008 | 0.042 ± 0.002 | 3.64 |
| | $H_{Weibull}$ | 92.472 ± 0.394 | 99.450 ± 0.026 | 97.882 ± 0.098 | 0.009 ± 0.001 | 0.032 ± 0.004 | 0.051 ± 0.010 | 0.044 ± 0.002 | 3.79 |
| Adult | $H_{Softmax}$ | 85.351 ± 0.157 | 90.791 ± 0.259 | 78.118 ± 0.569 | 0.016 ± 0.003 | 0.048 ± 0.006 | 0.409 ± 0.011 | 0.243 ± 0.006 | 4.79 |
| | $H_{ALD}$ | **85.404 ± 0.336** | 90.798 ± 0.385 | 78.222 ± 0.918 | 0.015 ± 0.003 | 0.046 ± 0.002 | **0.398 ± 0.012** | 0.236 ± 0.009 | **2.29** |
| | $H_{Norm}$ | 85.321 ± 0.384 | 90.729 ± 0.314 | **78.369 ± 0.598** | 0.016 ± 0.003 | 0.046 ± 0.002 | 0.402 ± 0.011 | **0.229 ± 0.011** | 3.21 |
| | $H_{Logistic}$ | 85.009 ± 0.367 | 90.369 ± 0.319 | 77.543 ± 0.744 | 0.016 ± 0.003 | 0.049 ± 0.004 | 0.421 ± 0.014 | 0.232 ± 0.010 | 5.57 |
| | $H_{Exp}$ | 85.311 ± 0.305 | **90.862 ± 0.308** | 78.196 ± 0.721 | 0.016 ± 0.002 | **0.042 ± 0.005** | 0.401 ± 0.009 | 0.247 ± 0.006 | 3.43 |
| | $H_{LogNorm}$ | 85.244 ± 0.286 | 90.684 ± 0.330 | 78.251 ± 0.509 | 0.017 ± 0.003 | 0.046 ± 0.005 | 0.405 ± 0.009 | 0.232 ± 0.009 | 4.07 |
| | $H_{Weibull}$ | 85.128 ± 0.483 | 90.711 ± 0.514 | 78.067 ± 1.183 | **0.014 ± 0.001** | 0.044 ± 0.003 | 0.401 ± 0.011 | 0.232 ± 0.008 | 3.64 |
| Cifar-10 | $H_{Softmax}$ | 90.485 ± 0.359 | 99.417 ± 0.049 | 96.458 ± 0.246 | 0.041 ± 0.003 | – | 0.087 ± 0.003 | 0.007 ± 0.002 | 3.92 |
| | $H_{ALD}$ | **90.760 ± 0.573** | 99.415 ± 0.068 | **96.832 ± 0.263** | 0.037 ± 0.006 | – | 0.085 ± 0.006 | **0.005 ± 0.001** | **2.08** |
| | $H_{Norm}$ | 90.500 ± 0.096 | 99.422 ± 0.016 | 96.461 ± 0.108 | 0.039 ± 0.005 | – | 0.106 ± 0.016 | 0.010 ± 0.001 | 4.50 |
| | $H_{Logistic}$ | 90.517 ± 0.507 | **99.433 ± 0.049** | 96.529 ± 0.274 | 0.038 ± 0.003 | – | **0.083 ± 0.010** | 0.009 ± 0.002 | 2.58 |
| | $H_{Exp}$ | 89.555 ± 0.413 | 99.340 ± 0.044 | 96.037 ± 0.145 | 0.039 ± 0.004 | – | 0.102 ± 0.026 | 0.036 ± 0.004 | 5.92 |
| | $H_{LogNorm}$ | 90.302 ± 0.722 | 99.389 ± 0.042 | 96.391 ± 0.288 | **0.036 ± 0.005** | – | 0.105 ± 0.019 | 0.020 ± 0.002 | 4.67 |
| | $H_{Weibull}$ | 90.413 ± 0.547 | 99.425 ± 0.050 | 96.411 ± 0.324 | 0.042 ± 0.003 | – | 0.104 ± 0.004 | 0.007 ± 0.001 | 4.33 |
| Cifar-100 | $H_{Softmax}$ | 67.405 ± 0.484 | **98.599 ± 0.039** | 73.403 ± 0.546 | 0.119 ± 0.006 | – | 0.365 ± 0.036 | **0.008 ± 0.002** | 3.83 |
| | $H_{ALD}$ | 68.313 ± 0.380 | 98.430 ± 0.187 | 74.880 ± 0.500 | 0.109 ± 0.017 | – | **0.299 ± 0.010** | **0.008 ± 0.001** | **1.92** |
| | $H_{Norm}$ | 67.295 ± 0.578 | 98.164 ± 0.255 | 73.102 ± 0.466 | 0.135 ± 0.015 | – | 0.379 ± 0.015 | **0.008 ± 0.002** | 4.92 |
| | $H_{Logistic}$ | 67.590 ± 0.175 | 98.312 ± 0.146 | 73.599 ± 0.316 | 0.141 ± 0.006 | – | 0.382 ± 0.037 | **0.008 ± 0.003** | 5.25 |
| | $H_{Exp}$ | 66.430 ± 0.756 | 97.525 ± 0.207 | 72.016 ± 0.729 | 0.113 ± 0.007 | – | 0.332 ± 0.017 | 0.091 ± 0.005 | 4.00 |
| | $H_{LogNorm}$ | **68.767 ± 0.710** | 98.318 ± 0.085 | **75.113 ± 0.540** | **0.106 ± 0.022** | – | 0.341 ± 0.013 | 0.009 ± 0.000 | 2.00 |
| | $H_{Weibull}$ | 66.580 ± 0.684 | 98.380 ± 0.015 | 72.592 ± 0.532 | 0.137 ± 0.004 | – | 0.384 ± 0.020 | 0.009 ± 0.000 | 6.08 |
| SST-2 | $H_{Softmax}$ | 90.969 ± 0.424 | 97.159 ± 0.210 | 97.302 ± 0.309 | 0.054 ± 0.011 | – | 0.153 ± 0.022 | 0.073 ± 0.008 | 5.08 |
| | $H_{ALD}$ | **91.657 ± 0.848** | **97.305 ± 0.513** | **97.503 ± 0.520** | 0.042 ± 0.006 | – | 0.148 ± 0.017 | 0.069 ± 0.013 | 2.33 |
| | $H_{Norm}$ | 90.969 ± 0.570 | 96.774 ± 0.754 | 96.565 ± 1.358 | 0.041 ± 0.009 | – | **0.118 ± 0.007** | 0.071 ± 0.004 | 3.50 |
| | $H_{Logistic}$ | 91.084 ± 0.125 | 97.066 ± 0.351 | 97.301 ± 0.315 | 0.044 ± 0.006 | – | 0.132 ± 0.013 | 0.075 ± 0.011 | 4.75 |
| | $H_{Exp}$ | 91.284 ± 0.693 | 97.039 ± 0.265 | 97.258 ± 0.237 | **0.039 ± 0.008** | – | 0.136 ± 0.012 | 0.078 ± 0.009 | 3.58 |
| | $H_{LogNorm}$ | 90.682 ± 0.784 | 96.843 ± 0.180 | 97.083 ± 0.287 | 0.052 ± 0.009 | – | 0.146 ± 0.016 | 0.080 ± 0.017 | 5.92 |
| | $H_{Weibull}$ | 91.284 ± 0.612 | 97.143 ± 0.248 | 97.396 ± 0.356 | 0.042 ± 0.011 | – | 0.127 ± 0.008 | **0.067 ± 0.008** | **2.25** |
| SST-5 | $H_{Softmax}$ | 53.224 ± 0.595 | 83.055 ± 0.191 | 51.692 ± 1.062 | 0.253 ± 0.074 | – | 0.406 ± 0.073 | **0.125 ± 0.018** | 2.83 |
| | $H_{ALD}$ | 53.839 ± 0.324 | 83.238 ± 0.090 | 51.989 ± 0.516 | **0.147 ± 0.085** | – | 0.382 ± 0.048 | 0.133 ± 0.060 | **2.42** |
| | $H_{Norm}$ | **54.027 ± 0.366** | 82.831 ± 0.482 | 51.978 ± 0.592 | 0.268 ± 0.137 | – | 0.483 ± 0.070 | 0.201 ± 0.081 | 4.33 |
| | $H_{Logistic}$ | 53.439 ± 0.560 | 82.739 ± 0.200 | 51.950 ± 0.470 | 0.369 ± 0.024 | – | 0.552 ± 0.079 | 0.164 ± 0.002 | 6.17 |
| | $H_{Exp}$ | 49.491 ± 2.594 | 80.996 ± 1.730 | 48.235 ± 0.889 | 0.292 ± 0.119 | – | 0.657 ± 0.060 | 0.182 ± 0.023 | 6.75 |
| | $H_{LogNorm}$ | 53.190 ± 0.257 | **83.418 ± 0.129** | **52.484 ± 0.850** | 0.160 ± 0.099 | – | **0.371 ± 0.105** | 0.257 ± 0.054 | 3.92 |
| | $H_{Weibull}$ | 53.552 ± 0.309 | 82.758 ± 0.836 | 51.864 ± 0.855 | 0.335 ± 0.066 | – | 0.501 ± 0.111 | 0.143 ± 0.011 | 4.58 |
| Overall | $H_{Softmax}$ | 5.00 | 3.22 | 4.89 | 4.67 | 4.50 | 5.11 | 3.67 | 4.44 |
| | $H_{ALD}$ | **2.17** | **2.89** | **2.61** | **2.50** | **2.80** | 3.22 | **2.00** | **2.60** |
| | $H_{Norm}$ | 3.94 | 4.22 | 3.89 | 4.06 | 4.40 | 4.33 | 3.94 | 4.11 |
| | $H_{Logistic}$ | 3.72 | 4.22 | 3.28 | 4.83 | 5.10 | **3.11** | 4.00 | 4.04 |
| | $H_{Exp}$ | 5.33 | 5.44 | 5.11 | 3.83 | 3.90 | 4.61 | 5.61 | 4.83 |
| | $H_{LogNorm}$ | 4.00 | 4.44 | 4.00 | 3.39 | 4.10 | 4.17 | 5.33 | 4.20 |
| | $H_{Weibull}$ | 3.83 | 3.56 | 4.22 | 4.72 | 3.20 | 3.44 | 3.44 | 3.77 |

**Quantitative Analysis (Table 5). Discriminative Performance:** $H_{ALD}$ consistently outperforms $H_{Softmax}$ across the majority of datasets. In the overall summary (bottom rows of Table 5), $H_{ALD}$ achieves the best average rank of 2.17 for Accuracy (ACC), 2.61 for Average Precision (AP) and 2.89 for Area Under the ROC Curve (AUC). In contrast, $H_{Softmax}$ lags significantly with an overall ACC rank of 5.00, AP rank of 4.89 and AUC for 3.22.

**Calibration Quality:** The performance gap is most evident in calibration metrics. Table 5 highlights that $H_{Softmax}$ yields poor calibration, with an overall average rank of 4.67 for ECE and 3.67 for KCE. $H_{ALD}$ dominates this category with the best overall ranks of 2.50 (ECE) and 2.00 (KCE). This reduction in calibration error is substantial. For instance, on the SST-5 dataset, $H_{ALD}$ reduces the ECE from 0.253 ($H_{Softmax}$) to 0.147, demonstrating a superior ability to mitigate overconfidence.

**Statistical Significance (Figure 4)** The trends also observed in Table 5 are statistically validated by the Critical Difference diagrams (Demšar, 2006) in Figure 4, which cover the full spectrum of experimental settings ($N = 45$). As shown in Figure 4(b) and (e), $H_{ALD}$ achieves the lowest Friedman average ranks for both ACC (3.18) and ECE (3.07). Crucially, the CD diagrams confirm that $H_{Softmax}$ is statistically significantly worse than $H_{ALD}$ in calibration tasks (ECE rank 4.58 *vs.* 3.07), falling outside the critical difference bar. This reinforces the conclusion that the fixed-scale Gumbel assumption in Softmax limits its uncertainty estimation capabilities.

**Complexity Trade-off (Table 6)** Finally, we evaluate whether the observed performance gains impose a significant computational burden. Practical deployment requires a rigorous assessment of efficiency. Table 6 presents a comparative analysis of parameter complexity and relative computational cost (combined training and inference time) for each classifier, normalized against the standard Softmax baseline ($H_{Softmax}$).

In terms of **Parameter Complexity**, the ratio reflects the number of learnable scalars per class in the final output layer:

- **Single-Parameter Models:** Both the baseline $H_{Softmax}$ and $H_{Exp}$ maintain a ratio of 1, as the Exponential distribution relies solely on the rate parameter $\lambda$, matching the complexity of standard logits.

- **Two-Parameter Models:** The distributions $H_{Norm}$, $H_{Logistic}$, $H_{LogNorm}$, and $H_{Weibull}$ require two parameters per class (*e.g.*, location $\mu$ and scale $\sigma$), resulting in a parameter ratio of 2.

- **Three-Parameter Models:** The $H_{ALD}$ is the most complex, requiring three parameters (*i.e.*, location $\theta$, scale $\sigma$, and asymmetry $\kappa$) per class, yielding a ratio of 3.

Regarding **Time Costs**, despite the increased parameterization in the output head, the relative increase in wall-clock time is marginal. Even for $H_{ALD}$, which triples the output layer complexity, the total computational overhead is only $2.0\%$ ($H_{ALD} \approx 1.020\times$) compared to the baseline. This minimal impact is attributable to the fact that the computational cost of deep learning models is dominated by the backbone feature extractor, rendering the additional operations in the classification head negligible. Given the substantial improvements in both accuracy (Table 5 ACC rank 2.17) and calibration (KCE rank 2.00), these results confirm that $H_{ALD}$ serves as a highly viable and scalable surrogate for $H_{Softmax}$, delivering significant performance gains with virtually no penalty in computational efficiency.

*Table 6.* **Parameter Complexity and Computational Cost Analysis.** Comparison of learnable parameter ratios in the output layer and the relative time costs (Training + Inference) normalized against the Softmax baseline $H_{Softmax}$ with early stopping enabled across all datasets.

| Classifier | Param. Ratio | Train + Inf. Time |
|:---:|:---:|:---:|
| $H_{Exp}$ | 1 | 1.001 |
| $H_{Logistic}$ | 2 | 1.003 |
| $H_{Norm}$ | 2 | 1.010 |
| $H_{LogNorm}$ | 2 | 1.015 |
| $H_{Weibull}$ | 2 | 1.017 |
| $H_{ALD}$ | 3 | 1.020 |

**Why Distribution-based Classifiers Outperform Softmax?** A potential explanation for the empirical superiority of distribution-based classifiers, particularly $H_{ALD}$, over the standard $H_{Softmax}$ baseline lies in the fundamental differences in

their modeling capacity and inductive biases. We can posit that Softmax functions as a restricted *location-only* estimator. In contrast, distribution-based classifiers such as $\mathbf{H}_{\text{ALD}}$ generalize this framework by jointly modeling the *location*, *scale*, and *shape*, thereby offering a comprehensive probabilistic description of the latent space.

**Softmax as a Restricted Fixed-Scale Estimator.** From a probabilistic perspective, our analysis reveals that $\mathbf{H}_{\text{Softmax}}$ essentially functions as a parameter estimator that predicts the *location* parameter ($\mu = z_c$) of a Gumbel distribution (Gumbel, 1935), while implicitly fixing the *scale* parameter ($\beta$) to 1 (see Appendix A.1 for the proof). This theoretical constraint has significant practical implications. By locking the scale parameter, Softmax enforces a strong assumption of *homoscedasticity*—implying that the underlying distribution of every class and every sample shares an identical, constant variance. Consequently, the model lacks the degrees of freedom to capture *heteroscedasticity*, *i.e.*, the varying levels of aleatoric uncertainty inherent in different samples. When facing noisy or ambiguous inputs, an ideal model should increase the predicted scale (widen the distribution) to reflect higher uncertainty. However, Softmax cannot adapt its spread. This inability to adjust the distributional width forces the model to maintain sharp, peaked probability distributions even in uncertain regions of the feature space, directly leading to the phenomenon of overconfidence and the poor calibration (high ECE) observed in our experiments.

**The Expressiveness of Distribution Parameters.** In contrast, distribution-based classifiers decouple the decision boundary into distinct statistical components, offering greater flexibility in data modeling. Taking $\mathbf{H}_{\text{ALD}}$ as a prime example, beyond the standard *location* estimation, the model explicitly incorporates:

- **Adaptive Uncertainty via Scale ($\sigma$):** By predicting a sample-specific scale parameter $\sigma_c(\mathbf{x})$, the model can adaptively "flatten" the probability density for ambiguous or noisy samples. This mechanism naturally yields a high-entropy posterior distribution for uncertain inputs, effectively mitigating overconfidence. This advantage is empirically validated by the significant improvements in calibration metrics (ECE), particularly on datasets with high inherent ambiguity such as DRY-BEAN and SST-5.

- **Fitting Skewed Manifolds via Shape ($\kappa$):** Real-world tabular data often departs from the symmetric Gaussian assumption, exhibiting heavy tails or asymmetry. The Asymmetric Laplace Distribution (ALD) introduces a shape parameter $\kappa$ that allows the learned density to skew left or right. This flexibility enables the decision boundary to better conform to the intrinsic geometry of asymmetric classes, thereby boosting discriminative performance—as evidenced by the superior Accuracy, AP, and AUC scores achieved on the SST-2 dataset.

Fundamentally, while Softmax assumes a rigid, symmetric geometry for all classes, $\mathbf{H}_{\text{ALD}}$ allows the class-conditional densities to deform (via $\sigma$) and skew (via $\kappa$) adaptively. This generalization capability explains why $\mathbf{H}_{\text{ALD}}$ achieves the superior Average Rank across diverse benchmarks. In conclusion, while $\mathbf{H}_{\text{Softmax}}$ remains a lightweight baseline, $\mathbf{H}_{\text{ALD}}$ provides a compelling alternative for applications where high accuracy and reliable uncertainty estimation are paramount, fully justifying the moderate increase in model complexity.

### C.3. Decoupled Dual-Stream Optimization Strategy

Table 7 compares three optimization variants for the ALD classifier. Overall, $\mathbf{H}_{\text{ALD}}^{\text{NLL}}$ often achieves strong discriminative metrics (ACC/AUC/AP), but its calibration can be suboptimal, especially on more complex modalities (*e.g.*, CIFAR and SST), where ECE/KCE and group-level ECE remain relatively high. In contrast, naively incorporating the calibration objective with full backpropagation, $\mathbf{H}_{\text{ALD}}^{\text{Full Grad}}$, tends to exhibit a less favorable trade-off: while it can mildly improve some calibration metrics over NLL-only training, it frequently degrades ACC/AUC/AP, suggesting that calibration gradients may interfere with discriminative representation learning. Finally, the detachment-based DDSO variant $\mathbf{H}_{\text{ALD}}^{\text{Detached}}$ consistently provides the best overall balance across datasets: it matches or recovers the discriminative performance of $\mathbf{H}_{\text{ALD}}^{\text{NLL}}$ in most cases (and even improves it on several tabular/image tasks such as ONLINE-SHOPPERS, ADULT, and CIFAR-100), while yielding uniformly lower ECE, GroupX/GroupY ECE, and often lower KCE. These results support the key motivation of *DDSO*: blocking calibration gradients from flowing into the backbone prevents the model from satisfying low-confidence queries by degrading feature quality, and enables efficient multi-anchor updates to accelerate calibration without compromising class separability. As a consequence, $\mathbf{H}_{\text{ALD}}^{\text{Detached}}$ achieves the strongest average ranks across the 9 datasets, making detachment a practical default for stable and reliable ALD-based training.

*Table 7.* Performance comparison between $H_{ALD}^{NLL}$, $H_{ALD}^{Full\ Grad}$ and $H_{ALD}^{Detached}$ across 9 datasets. Best values within each dataset block are highlighted in **bold yellow**.

| Dataset | Model | ACC | AUC | AP | ECE | GroupX_ECE | GroupY_ECE | KCE | Avg Rank |
|---|---|---|---|---|---|---|---|---|---|
| Breast-Cancer | $H_{ALD}^{NLL}$ | **98.070 ± 1.023** | **99.583 ± 0.340** | **99.429 ± 0.462** | 0.031 ± 0.013 | 0.090 ± 0.031 | 0.057 ± 0.024 | 0.026 ± 0.013 | 2.14 |
| | $H_{ALD}^{Full\ Grad}$ | 96.796 ± 1.792 | 99.234 ± 0.282 | 98.991 ± 0.418 | **0.023 ± 0.012** | 0.084 ± 0.013 | **0.045 ± 0.015** | **0.025 ± 0.007** | 2.14 |
| | $H_{ALD}^{Detached}$ | 97.995 ± 0.895 | 99.565 ± 0.161 | 99.279 ± 0.238 | **0.023 ± 0.007** | **0.080 ± 0.026** | 0.048 ± 0.025 | **0.025 ± 0.007** | **1.71** |
| Heart-Disease | $H_{ALD}^{NLL}$ | **61.667 ± 3.162** | **80.046 ± 2.029** | 41.338 ± 2.967 | 0.097 ± 0.014 | 0.348 ± 0.022 | 0.503 ± 0.112 | 0.272 ± 0.023 | 2.29 |
| | $H_{ALD}^{Full\ Grad}$ | 59.667 ± 2.718 | 78.142 ± 2.087 | 40.921 ± 2.917 | 0.089 ± 0.028 | 0.326 ± 0.128 | 0.421 ± 0.135 | 0.271 ± 0.023 | 2.43 |
| | $H_{ALD}^{Detached}$ | 60.000 ± 4.082 | 79.211 ± 2.983 | **42.023 ± 1.864** | **0.087 ± 0.021** | **0.309 ± 0.044** | **0.404 ± 0.109** | **0.262 ± 0.044** | **1.29** |
| Online-Shoppers | $H_{ALD}^{NLL}$ | 89.757 ± 0.326 | 91.830 ± 0.608 | 70.654 ± 2.239 | 0.016 ± 0.002 | 0.051 ± 0.002 | 0.489 ± 0.024 | 0.211 ± 0.015 | 2.57 |
| | $H_{ALD}^{Full\ Grad}$ | 88.468 ± 0.461 | 89.827 ± 1.026 | 68.546 ± 3.364 | 0.015 ± 0.001 | **0.046 ± 0.004** | 0.485 ± 0.014 | **0.147 ± 0.017** | 2.14 |
| | $H_{ALD}^{Detached}$ | **90.065 ± 0.445** | **92.270 ± 0.609** | **72.142 ± 1.611** | **0.014 ± 0.002** | 0.047 ± 0.003 | **0.477 ± 0.043** | 0.182 ± 0.043 | **1.29** |
| Dry-Bean | $H_{ALD}^{NLL}$ | 92.472 ± 0.463 | 99.428 ± 0.069 | 97.902 ± 0.163 | 0.009 ± 0.004 | 0.032 ± 0.006 | 0.048 ± 0.006 | 0.040 ± 0.002 | 2.71 |
| | $H_{ALD}^{Full\ Grad}$ | 92.521 ± 0.646 | 99.347 ± 0.362 | 97.821 ± 0.263 | **0.008 ± 0.001** | **0.031 ± 0.002** | 0.041 ± 0.014 | 0.036 ± 0.013 | 2.14 |
| | $H_{ALD}^{Detached}$ | **92.545 ± 0.570** | **99.443 ± 0.034** | **97.903 ± 0.153** | **0.008 ± 0.002** | **0.031 ± 0.002** | **0.040 ± 0.010** | **0.035 ± 0.004** | **1.14** |
| Adult | $H_{ALD}^{NLL}$ | 85.404 ± 0.336 | **90.798 ± 0.385** | 78.222 ± 0.918 | 0.015 ± 0.003 | 0.046 ± 0.002 | 0.398 ± 0.012 | 0.236 ± 0.009 | 2.57 |
| | $H_{ALD}^{Full\ Grad}$ | 85.582 ± 0.372 | 88.267 ± 0.441 | 77.563 ± 0.501 | 0.014 ± 0.005 | **0.036 ± 0.005** | 0.333 ± 0.015 | 0.226 ± 0.012 | 2.14 |
| | $H_{ALD}^{Detached}$ | **86.205 ± 0.368** | 90.718 ± 0.285 | **78.268 ± 0.602** | **0.013 ± 0.003** | 0.037 ± 0.007 | **0.312 ± 0.007** | **0.218 ± 0.006** | **1.29** |
| Cifar-10 | $H_{ALD}^{NLL}$ | **90.760 ± 0.573** | **99.415 ± 0.068** | **96.832 ± 0.263** | 0.037 ± 0.006 | – | 0.085 ± 0.006 | 0.005 ± 0.001 | 2.00 |
| | $H_{ALD}^{Full\ Grad}$ | 89.063 ± 0.402 | 98.970 ± 0.036 | 95.728 ± 0.347 | 0.025 ± 0.011 | – | 0.079 ± 0.005 | **0.004 ± 0.002** | 2.42 |
| | $H_{ALD}^{Detached}$ | 90.075 ± 0.296 | 99.051 ± 0.035 | 96.479 ± 0.118 | **0.022 ± 0.006** | – | **0.076 ± 0.012** | **0.004 ± 0.001** | **1.58** |
| Cifar-100 | $H_{ALD}^{NLL}$ | 68.313 ± 0.380 | 98.430 ± 0.187 | 74.880 ± 0.500 | 0.109 ± 0.017 | – | 0.299 ± 0.010 | **0.005 ± 0.001** | 2.17 |
| | $H_{ALD}^{Full\ Grad}$ | 67.101 ± 0.401 | 96.847 ± 0.883 | 73.118 ± 0.827 | 0.083 ± 0.014 | – | 0.252 ± 0.021 | 0.006 ± 0.001 | 2.58 |
| | $H_{ALD}^{Detached}$ | **68.405 ± 0.606** | **98.778 ± 0.039** | **75.114 ± 0.785** | **0.079 ± 0.006** | – | **0.229 ± 0.029** | 0.006 ± 0.001 | **1.25** |
| Sst-2 | $H_{ALD}^{NLL}$ | **91.657 ± 0.848** | **97.305 ± 0.513** | **97.503 ± 0.520** | 0.042 ± 0.006 | – | 0.148 ± 0.017 | 0.069 ± 0.013 | 2.00 |
| | $H_{ALD}^{Full\ Grad}$ | 89.967 ± 0.392 | 95.308 ± 0.184 | 96.395 ± 0.094 | 0.039 ± 0.007 | – | 0.139 ± 0.015 | 0.061 ± 0.012 | 2.50 |
| | $H_{ALD}^{Detached}$ | 90.979 ± 0.516 | 96.990 ± 0.238 | 97.311 ± 0.061 | **0.032 ± 0.007** | – | **0.129 ± 0.013** | **0.055 ± 0.013** | **1.50** |
| Sst-5 | $H_{ALD}^{NLL}$ | **53.839 ± 0.324** | **83.238 ± 0.090** | **51.989 ± 0.516** | 0.147 ± 0.085 | – | 0.382 ± 0.048 | 0.133 ± 0.060 | 2.00 |
| | $H_{ALD}^{Full\ Grad}$ | 51.847 ± 0.251 | 82.532 ± 0.226 | 50.686 ± 0.624 | 0.136 ± 0.097 | – | 0.276 ± 0.038 | 0.118 ± 0.025 | 2.50 |
| | $H_{ALD}^{Detached}$ | 53.364 ± 0.277 | 83.000 ± 0.847 | 51.904 ± 1.635 | **0.128 ± 0.132** | – | **0.264 ± 0.108** | **0.097 ± 0.013** | **1.50** |
| **Overall** | $H_{ALD}^{NLL}$ | 1.67 | **1.33** | 1.56 | 3.00 | 3.00 | 3.00 | 2.78 | 2.27 |
| | $H_{ALD}^{Full\ Grad}$ | 2.78 | 3.00 | 3.00 | 1.89 | **1.50** | 1.89 | 1.83 | 2.33 |
| | $H_{ALD}^{Detached}$ | **1.56** | 1.67 | **1.44** | **1.11** | **1.50** | **1.11** | **1.39** | **1.39** |

## C.4. Full results

We present a comprehensive evaluation of our proposed $\mathbf{H}_{\text{ICALD}}$ framework against the standard Softmax baseline ($\mathbf{H}_{\text{Softmax}}$), representative *post-calibration* methods (TS, Guo et al. 2017 and HB, Zadrozny & Elkan 2001), and *pre-calibration* approaches (MMD, Marx et al. 2023). The aggregate performance across all datasets is summarized in the "Overall" section of Table 8, and the statistical significance of these rankings is visualized in the Critical Difference (CD) diagrams (Figure 5). Notably, it is important to distinguish between the ranking values reported in Table 8 and Figure 5. The "Avg Rank" in Table 8 represents the mean of rankings derived strictly from the 9 representative datasets listed in the table. In contrast, Figure 5 presents the Friedman average ranks calculated across the complete set of $N = 45$ experimental settings (comprising 9 datasets $\times$ 5 random seeds) to facilitate rigorous statistical hypothesis testing (*i.e.*, the Nemenyi test Nemenyi 1963). Although the absolute rank values differ due to the varying sample sizes and calculation scopes, the relative performance trends remain consistent, with $\mathbf{H}_{\text{ICALD}}$ demonstrating dominance in both evaluations.

**Comprehensive Superiority.** Across the aggregated metrics, our proposed *pre-calibration* strategy $\mathbf{H}_{\text{ICALD}}^{\text{Pre}}$ achieves the best (lowest) overall average rank of **2.98**, followed by the *post-calibration* variant $\mathbf{H}_{\text{ICALD}}^{\text{Post}}$ at **3.79**. Both variants significantly outperform the standard $\mathbf{H}_{\text{Softmax}}$ baseline (6.79) and the post-hoc methods $\mathbf{H}_{\text{Softmax+TS}}$ (7.73) and $\mathbf{H}_{\text{Softmax+HB}}$ (9.16). This establishes that incorporating individualized calibration into $\mathbf{H}_{\text{ALD}}$ yields the most robust classifier across diverse modalities.

**Calibration Excellence (Average-Level ECE, Group-Level ECE, KCE).** $\mathbf{H}_{\text{ICALD}}$ demonstrates dominant performance across multi-level calibration metrics, addressing the limitations of standard methods:

- *Average-Level:* $\mathbf{H}_{\text{ICALD}}^{\text{Pre}}$ achieves the top rank in ECE (**2.00**), substantially improving upon the uncalibrated $\mathbf{H}_{\text{ALD}}$ (6.83) and $\mathbf{H}_{\text{Softmax}}$ (7.83).

- *Group-Level:* A key strength of our approach is its reliability across subpopulations. $\mathbf{H}_{\text{ICALD}}^{\text{Pre}}$ secures the best rank in feature-based GroupX_ECE (**1.20**) and class-based GroupY_ECE (**2.39**). This indicates that $\mathbf{H}_{\text{ICALD}}$ does not merely optimize a global average but ensures consistent calibration across different regions of the feature space and minority classes.

- *Distribution-Level:* While the MMD-based baseline $\mathbf{H}_{\text{ALD+MMD}}$ expectedly leads in KCE (**1.39**) due to its direct optimization of the kernel objective, $\mathbf{H}_{\text{ICALD}}^{\text{Pre}}$ remains highly competitive with a rank of **2.00**, far surpassing $\mathbf{H}_{\text{Softmax}}$ (5.89).

**Discriminative Robustness.** Crucially, the improvements in calibration do not come at the expense of discriminative power. As shown in the overall rankings for Accuracy (ACC) and Average Precision (AP), the uncalibrated $\mathbf{H}_{\text{ALD}}$ achieves the highest accuracy rank (**2.94**). However, $\mathbf{H}_{\text{ICALD}}^{\text{Pre}}$ maintains a highly competitive accuracy rank of **3.83** and the best AP rank of **2.67**, significantly outperforming the standard Softmax baseline (ACC: 7.28, AP: 7.17). This contrasts sharply with post-hoc methods like Histogram Binning ($\mathbf{H}_{\text{Softmax+HB}}$), which degrades accuracy significantly (Rank 10.22) to achieve calibration.

**Summary.** The CD diagrams (Figure 5) and Table 8 collectively demonstrate that $\mathbf{H}_{\text{ICALD}}$ offers the most balanced profile. It effectively bridges the gap between high predictive accuracy and rigorous probabilistic reliability, providing a holistic improvement over both standard deep learning baselines and traditional calibration techniques.

*Table 8.* The overall performance comparison of different classifiers. Best values in each column are highlighted in **bold yellow**.

| Dataset | Model | ACC | AUC | AP | ECE | GroupX_ECE | GroupY_ECE | KCE | Avg Rank |
|---|---|---|---|---|---|---|---|---|---|
| BREAST-CANCER | $\mathbf{H}_{\text{Softmax}}$ | $97.719 \pm 0.702$ | $99.286 \pm 0.320$ | $99.205 \pm 0.384$ | $0.030 \pm 0.009$ | $0.093 \pm 0.023$ | $0.060 \pm 0.018$ | $0.027 \pm 0.005$ | 6.07 |
| | $\mathbf{H}_{\text{ALD}}$ | $98.070 \pm 1.023$ | **$99.583 \pm 0.340$** | $99.429 \pm 0.462$ | $0.031 \pm 0.013$ | $0.090 \pm 0.031$ | $0.057 \pm 0.024$ | $0.026 \pm 0.013$ | 4.57 |
| | $\mathbf{H}_{\text{Softmax+TS}}$ | $97.719 \pm 0.702$ | $99.238 \pm 0.154$ | $99.161 \pm 0.276$ | $0.122 \pm 0.041$ | $0.170 \pm 0.043$ | $0.230 \pm 0.074$ | $0.206 \pm 0.060$ | 11.00 |
| | $\mathbf{H}_{\text{Softmax+HB}}$ | $95.211 \pm 0.719$ | $97.286 \pm 0.473$ | $92.806 \pm 0.649$ | $0.057 \pm 0.018$ | $0.182 \pm 0.049$ | $0.111 \pm 0.067$ | $0.041 \pm 0.019$ | 9.86 |
| | $\mathbf{H}_{\text{ICSoftmax}}^{\text{Post}}$ | $97.368 \pm 0.961$ | $98.910 \pm 0.806$ | $99.268 \pm 0.120$ | $0.024 \pm 0.005$ | $0.086 \pm 0.026$ | $0.052 \pm 0.018$ | $0.025 \pm 0.006$ | 5.36 |
| | $\mathbf{H}_{\text{ALD+TS}}$ | $98.070 \pm 1.023$ | $99.541 \pm 0.426$ | **$99.706 \pm 0.107$** | $0.064 \pm 0.009$ | $0.155 \pm 0.025$ | $0.108 \pm 0.021$ | $0.066 \pm 0.022$ | 8.21 |
| | $\mathbf{H}_{\text{ALD+HB}}$ | $97.561 \pm 0.744$ | $96.134 \pm 2.545$ | $93.126 \pm 5.405$ | $0.055 \pm 0.018$ | $0.149 \pm 0.023$ | $0.103 \pm 0.027$ | $0.042 \pm 0.006$ | 8.93 |
| | $\mathbf{H}_{\text{ICALD}}^{\text{Post}}$ | $98.368 \pm 0.468$ | $99.034 \pm 0.797$ | $98.992 \pm 0.830$ | | $0.086 \pm 0.031$ | $0.049 \pm 0.027$ | $0.024 \pm 0.010$ | 4.64 |
| | $\mathbf{H}_{\text{Softmax+MMD}}$ | $97.609 \pm 0.609$ | $99.210 \pm 0.657$ | $99.210 \pm 0.657$ | **$0.023 \pm 0.009$** | $0.089 \pm 0.024$ | $0.056 \pm 0.021$ | $0.023 \pm 0.006$ | 5.21 |
| | $\mathbf{H}_{\text{ICSoftmax}}^{\text{Pre}}$ | $97.768 \pm 0.961$ | $99.517 \pm 0.357$ | $99.314 \pm 0.474$ | $0.025 \pm 0.004$ | $0.083 \pm 0.025$ | $0.053 \pm 0.022$ | $0.024 \pm 0.019$ | 4.86 |
| | $\mathbf{H}_{\text{ALD+MMD}}$ | **$98.395 \pm 0.853$** | $99.504 \pm 0.325$ | $99.339 \pm 0.427$ | $0.032 \pm 0.014$ | $0.094 \pm 0.034$ | $0.055 \pm 0.022$ | **$0.020 \pm 0.014$** | 4.93 |
| | $\mathbf{H}_{\text{ICALD}}^{\text{Pre}}$ | $97.995 \pm 0.895$ | $99.565 \pm 0.161$ | $99.279 \pm 0.238$ | **$0.023 \pm 0.007$** | **$0.080 \pm 0.026$** | **$0.048 \pm 0.025$** | $0.025 \pm 0.007$ | **3.43** |
| HEART-DISEASE | $\mathbf{H}_{\text{Softmax}}$ | $59.333 \pm 3.590$ | $79.141 \pm 1.374$ | $41.130 \pm 2.677$ | $0.100 \pm 0.020$ | $0.315 \pm 0.039$ | $0.443 \pm 0.097$ | $0.295 \pm 0.024$ | 6.57 |
| | $\mathbf{H}_{\text{ALD}}$ | **$61.667 \pm 3.162$** | $80.046 \pm 2.029$ | $41.338 \pm 2.967$ | $0.097 \pm 0.014$ | $0.348 \pm 0.022$ | $0.503 \pm 0.112$ | $0.272 \pm 0.023$ | 5.79 |
| | $\mathbf{H}_{\text{Softmax+TS}}$ | $59.333 \pm 3.590$ | $76.175 \pm 6.810$ | $40.993 \pm 4.606$ | $0.194 \pm 0.055$ | $0.430 \pm 0.079$ | $0.513 \pm 0.027$ | $0.358 \pm 0.105$ | 10.86 |
| | $\mathbf{H}_{\text{Softmax+HB}}$ | $55.000 \pm 4.944$ | $71.621 \pm 4.224$ | $35.422 \pm 4.380$ | $0.169 \pm 0.038$ | $0.457 \pm 0.097$ | $0.566 \pm 0.082$ | $0.293 \pm 0.066$ | 11.79 |
| | $\mathbf{H}_{\text{ICSoftmax}}^{\text{Post}}$ | $60.667 \pm 1.091$ | $79.273 \pm 1.415$ | $41.290 \pm 2.701$ | $0.092 \pm 0.051$ | $0.345 \pm 0.046$ | $0.408 \pm 0.090$ | $0.282 \pm 0.026$ | 5.29 |
| | $\mathbf{H}_{\text{ALD+TS}}$ | **$61.667 \pm 3.162$** | $75.549 \pm 10.471$ | $41.579 \pm 6.709$ | $0.205 \pm 0.084$ | $0.473 \pm 0.167$ | $0.577 \pm 0.166$ | $0.366 \pm 0.086$ | 11.64 |
| | $\mathbf{H}_{\text{ALD+HB}}$ | $48.667 \pm 17.994$ | $63.866 \pm 11.014$ | $35.246 \pm 6.860$ | $0.200 \pm 0.082$ | $0.410 \pm 0.111$ | $0.517 \pm 0.191$ | $0.282 \pm 0.131$ | 11.36 |
| | $\mathbf{H}_{\text{ICALD}}^{\text{Post}}$ | $61.000 \pm 2.906$ | **$81.497 \pm 1.810$** | $44.670 \pm 2.289$ | **$0.082 \pm 0.025$** | $0.350 \pm 0.047$ | $0.454 \pm 0.114$ | $0.262 \pm 0.039$ | **3.79** |
| | $\mathbf{H}_{\text{Softmax+MMD}}$ | $59.667 \pm 1.944$ | $79.844 \pm 0.879$ | $42.809 \pm 2.450$ | $0.091 \pm 0.009$ | $0.317 \pm 0.018$ | $0.484 \pm 0.030$ | $0.272 \pm 0.024$ | 5.64 |
| | $\mathbf{H}_{\text{ICSoftmax}}^{\text{Pre}}$ | $60.000 \pm 2.108$ | $79.602 \pm 2.345$ | $41.757 \pm 3.059$ | $0.088 \pm 0.017$ | **$0.305 \pm 0.052$** | **$0.403 \pm 0.081$** | $0.280 \pm 0.030$ | 4.29 |
| | $\mathbf{H}_{\text{ALD+MMD}}$ | $60.667 \pm 1.700$ | $80.382 \pm 3.215$ | **$44.890 \pm 1.793$** | $0.094 \pm 0.062$ | $0.355 \pm 0.078$ | $0.502 \pm 0.122$ | **$0.257 \pm 0.091$** | 5.57 |
| | $\mathbf{H}_{\text{ICALD}}^{\text{Pre}}$ | $60.000 \pm 4.082$ | $79.211 \pm 2.983$ | $42.023 \pm 1.864$ | $0.087 \pm 0.021$ | $0.309 \pm 0.044$ | $0.404 \pm 0.109$ | $0.262 \pm 0.044$ | 4.07 |
| ONLINE-SHOPPERS | $\mathbf{H}_{\text{Softmax}}$ | $89.749 \pm 0.339$ | $91.915 \pm 0.534$ | $70.777 \pm 1.511$ | $0.018 \pm 0.002$ | $0.055 \pm 0.005$ | $0.491 \pm 0.010$ | $0.211 \pm 0.011$ | 7.36 |
| | $\mathbf{H}_{\text{ALD}}$ | $89.757 \pm 0.326$ | $91.830 \pm 0.608$ | $70.654 \pm 2.239$ | $0.016 \pm 0.002$ | $0.051 \pm 0.002$ | $0.489 \pm 0.024$ | $0.211 \pm 0.015$ | 6.43 |
| | $\mathbf{H}_{\text{Softmax+TS}}$ | $89.749 \pm 0.339$ | $91.783 \pm 0.848$ | $70.623 \pm 2.528$ | $0.017 \pm 0.006$ | $0.056 \pm 0.010$ | $0.499 \pm 0.018$ | $0.260 \pm 0.023$ | 8.29 |
| | $\mathbf{H}_{\text{Softmax+HB}}$ | $89.594 \pm 0.532$ | $90.389 \pm 1.306$ | $68.494 \pm 3.254$ | $0.017 \pm 0.007$ | $0.049 \pm 0.013$ | $0.485 \pm 0.025$ | $0.243 \pm 0.013$ | 8.21 |
| | $\mathbf{H}_{\text{ICSoftmax}}^{\text{Post}}$ | $89.676 \pm 0.596$ | $91.454 \pm 1.242$ | $70.918 \pm 2.405$ | **$0.013 \pm 0.004$** | $0.052 \pm 0.013$ | $0.473 \pm 0.015$ | $0.194 \pm 0.027$ | 4.21 |
| | $\mathbf{H}_{\text{ALD+TS}}$ | $89.757 \pm 0.326$ | $91.985 \pm 0.453$ | $71.777 \pm 2.191$ | $0.015 \pm 0.009$ | $0.079 \pm 0.012$ | $0.482 \pm 0.023$ | $0.267 \pm 0.027$ | 7.64 |
| | $\mathbf{H}_{\text{ALD+HB}}$ | $89.043 \pm 0.891$ | $90.289 \pm 0.662$ | $68.044 \pm 3.224$ | $0.015 \pm 0.006$ | $0.049 \pm 0.011$ | $0.479 \pm 0.025$ | $0.241 \pm 0.021$ | 8.93 |
| | $\mathbf{H}_{\text{ICALD}}^{\text{Post}}$ | $89.797 \pm 0.403$ | $91.876 \pm 0.899$ | $71.877 \pm 2.707$ | $0.019 \pm 0.003$ | $0.055 \pm 0.005$ | **$0.465 \pm 0.024$** | $0.213 \pm 0.022$ | 6.50 |
| | $\mathbf{H}_{\text{Softmax+MMD}}$ | $89.862 \pm 0.373$ | $91.748 \pm 0.967$ | $70.726 \pm 3.176$ | $0.017 \pm 0.002$ | $0.054 \pm 0.006$ | $0.473 \pm 0.021$ | $0.180 \pm 0.012$ | 6.14 |
| | $\mathbf{H}_{\text{ICSoftmax}}^{\text{Pre}}$ | $89.651 \pm 0.547$ | $91.865 \pm 0.923$ | $70.537 \pm 2.644$ | $0.016 \pm 0.006$ | $0.052 \pm 0.014$ | $0.475 \pm 0.017$ | $0.192 \pm 0.025$ | 5.86 |
| | $\mathbf{H}_{\text{ALD+MMD}}$ | $89.968 \pm 0.449$ | $91.855 \pm 0.721$ | $71.700 \pm 2.684$ | $0.016 \pm 0.003$ | $0.050 \pm 0.006$ | $0.478 \pm 0.026$ | **$0.176 \pm 0.014$** | 4.79 |
| | $\mathbf{H}_{\text{ICALD}}^{\text{Pre}}$ | **$90.065 \pm 0.445$** | **$92.270 \pm 0.609$** | **$72.142 \pm 1.611$** | $0.014 \pm 0.002$ | **$0.047 \pm 0.003$** | $0.477 \pm 0.043$ | $0.182 \pm 0.043$ | **2.29** |
| DRY-BEAN | $\mathbf{H}_{\text{Softmax}}$ | $92.295 \pm 0.383$ | $99.438 \pm 0.022$ | $97.817 \pm 0.099$ | $0.017 \pm 0.004$ | $0.045 \pm 0.007$ | $0.083 \pm 0.022$ | $0.053 \pm 0.002$ | 9.00 |
| | $\mathbf{H}_{\text{ALD}}$ | $92.472 \pm 0.463$ | $99.428 \pm 0.069$ | $97.902 \pm 0.163$ | $0.009 \pm 0.004$ | $0.032 \pm 0.006$ | $0.048 \pm 0.006$ | $0.040 \pm 0.002$ | 3.07 |
| | $\mathbf{H}_{\text{Softmax+TS}}$ | $92.295 \pm 0.383$ | $99.393 \pm 0.040$ | $97.659 \pm 0.194$ | $0.016 \pm 0.003$ | $0.073 \pm 0.016$ | $0.053 \pm 0.015$ | $0.069 \pm 0.005$ | 9.79 |
| | $\mathbf{H}_{\text{Softmax+HB}}$ | $91.891 \pm 0.664$ | $99.227 \pm 0.085$ | $96.969 \pm 0.275$ | $0.016 \pm 0.005$ | $0.050 \pm 0.015$ | $0.070 \pm 0.017$ | $0.042 \pm 0.003$ | 9.64 |
| | $\mathbf{H}_{\text{ICSoftmax}}^{\text{Post}}$ | $92.413 \pm 0.489$ | $99.452 \pm 0.014$ | $97.847 \pm 0.032$ | $0.011 \pm 0.002$ | $0.037 \pm 0.005$ | $0.055 \pm 0.010$ | $0.044 \pm 0.003$ | 4.64 |
| | $\mathbf{H}_{\text{ALD+TS}}$ | $92.472 \pm 0.463$ | $99.423 \pm 0.055$ | $97.851 \pm 0.092$ | $0.009 \pm 0.013$ | $0.038 \pm 0.014$ | $0.049 \pm 0.012$ | $0.062 \pm 0.008$ | 5.50 |
| | $\mathbf{H}_{\text{ALD+HB}}$ | $91.179 \pm 0.924$ | $98.949 \pm 0.328$ | $96.145 \pm 1.358$ | **$0.008 \pm 0.011$** | $0.033 \pm 0.022$ | $0.044 \pm 0.016$ | $0.051 \pm 0.014$ | 5.93 |
| | $\mathbf{H}_{\text{ICALD}}^{\text{Post}}$ | $92.538 \pm 0.529$ | **$99.473 \pm 0.034$** | **$97.964 \pm 0.161$** | **$0.008 \pm 0.002$** | $0.032 \pm 0.007$ | $0.042 \pm 0.013$ | $0.032 \pm 0.007$ | **1.86** |
| | $\mathbf{H}_{\text{Softmax+MMD}}$ | $92.376 \pm 0.401$ | $99.435 \pm 0.029$ | $97.824 \pm 0.164$ | $0.017 \pm 0.005$ | $0.046 \pm 0.012$ | $0.061 \pm 0.020$ | **$0.029 \pm 0.004$** | 8.21 |
| | $\mathbf{H}_{\text{ICSoftmax}}^{\text{Pre}}$ | **$92.716 \pm 1.557$** | $99.360 \pm 0.079$ | $97.438 \pm 0.323$ | $0.012 \pm 0.005$ | $0.049 \pm 0.024$ | $0.056 \pm 0.023$ | $0.044 \pm 0.023$ | 6.36 |
| | $\mathbf{H}_{\text{ALD+MMD}}$ | $92.340 \pm 0.929$ | $99.417 \pm 0.068$ | $97.736 \pm 0.220$ | $0.013 \pm 0.009$ | $0.037 \pm 0.013$ | $0.048 \pm 0.057$ | $0.027 \pm 0.018$ | 5.21 |
| | $\mathbf{H}_{\text{ICALD}}^{\text{Pre}}$ | $92.545 \pm 0.570$ | $99.443 \pm 0.034$ | $97.903 \pm 0.153$ | **$0.008 \pm 0.002$** | **$0.031 \pm 0.002$** | **$0.040 \pm 0.010$** | $0.035 \pm 0.004$ | 2.43 |
| ADULT | $\mathbf{H}_{\text{Softmax}}$ | $85.351 \pm 0.157$ | $90.791 \pm 0.259$ | $78.118 \pm 0.569$ | $0.016 \pm 0.003$ | $0.048 \pm 0.006$ | $0.409 \pm 0.011$ | $0.243 \pm 0.006$ | 8.14 |
| | $\mathbf{H}_{\text{ALD}}$ | $85.404 \pm 0.336$ | $90.798 \pm 0.385$ | $78.222 \pm 0.918$ | $0.015 \pm 0.003$ | $0.046 \pm 0.002$ | $0.398 \pm 0.012$ | $0.236 \pm 0.009$ | 7.64 |
| | $\mathbf{H}_{\text{Softmax+TS}}$ | $85.351 \pm 0.157$ | **$90.867 \pm 0.273$** | **$78.286 \pm 0.574$** | $0.015 \pm 0.009$ | $0.041 \pm 0.011$ | $0.408 \pm 0.008$ | $0.272 \pm 0.011$ | 8.07 |
| | $\mathbf{H}_{\text{Softmax+HB}}$ | $85.268 \pm 0.346$ | $90.434 \pm 0.374$ | $77.344 \pm 0.574$ | $0.014 \pm 0.002$ | $0.043 \pm 0.007$ | $0.398 \pm 0.008$ | $0.231 \pm 0.011$ | 7.07 |
| | $\mathbf{H}_{\text{ICSoftmax}}^{\text{Post}}$ | $85.132 \pm 0.185$ | $90.756 \pm 0.244$ | $78.172 \pm 0.634$ | $0.014 \pm 0.003$ | $0.042 \pm 0.003$ | $0.399 \pm 0.011$ | $0.217 \pm 0.010$ | 6.57 |
| | $\mathbf{H}_{\text{ALD+TS}}$ | $85.404 \pm 0.336$ | $90.752 \pm 0.259$ | $78.085 \pm 0.585$ | $0.014 \pm 0.004$ | $0.060 \pm 0.009$ | $0.401 \pm 0.006$ | $0.278 \pm 0.009$ | 9.14 |
| | $\mathbf{H}_{\text{ALD+HB}}$ | $85.162 \pm 0.251$ | $90.304 \pm 0.317$ | $77.247 \pm 0.577$ | **$0.013 \pm 0.006$** | $0.051 \pm 0.005$ | $0.396 \pm 0.006$ | $0.247 \pm 0.010$ | 7.79 |
| | $\mathbf{H}_{\text{ICALD}}^{\text{Post}}$ | $85.264 \pm 0.359$ | $90.814 \pm 0.354$ | $78.061 \pm 0.766$ | $0.014 \pm 0.003$ | $0.038 \pm 0.006$ | $0.324 \pm 0.007$ | $0.214 \pm 0.008$ | 3.86 |
| | $\mathbf{H}_{\text{Softmax+MMD}}$ | $85.337 \pm 0.321$ | $90.842 \pm 0.321$ | $78.063 \pm 0.636$ | $0.014 \pm 0.002$ | $0.044 \pm 0.003$ | $0.409 \pm 0.011$ | **$0.211 \pm 0.005$** | 5.86 |
| | $\mathbf{H}_{\text{ICSoftmax}}^{\text{Pre}}$ | $85.367 \pm 0.228$ | $90.829 \pm 0.368$ | $78.187 \pm 0.623$ | $0.015 \pm 0.003$ | $0.044 \pm 0.004$ | $0.407 \pm 0.010$ | $0.224 \pm 0.009$ | 6.86 |
| | $\mathbf{H}_{\text{ALD+MMD}}$ | $85.304 \pm 0.366$ | $90.762 \pm 0.403$ | $78.029 \pm 0.891$ | $0.014 \pm 0.002$ | $0.043 \pm 0.002$ | $0.402 \pm 0.012$ | $0.212 \pm 0.006$ | 5.86 |
| | $\mathbf{H}_{\text{ICALD}}^{\text{Pre}}$ | **$86.205 \pm 0.368$** | $90.718 \pm 0.285$ | $78.268 \pm 0.602$ | **$0.013 \pm 0.003$** | **$0.037 \pm 0.007$** | **$0.312 \pm 0.007$** | $0.218 \pm 0.006$ | **2.93** |

| Dataset | Model | ACC | AUC | AP | ECE | GroupX_ECE | GroupY_ECE | KCE | Avg Rank |
|---|---|---|---|---|---|---|---|---|---|
| CIFAR-10 | $\mathbf{H}_{Softmax}$ | $90.485 \pm 0.359$ | $99.417 \pm 0.049$ | $96.458 \pm 0.246$ | $0.041 \pm 0.003$ | – | $0.087 \pm 0.003$ | $0.007 \pm 0.002$ | $8.10$ |
| | $\mathbf{H}_{ALD}$ | $90.760 \pm 0.573$ | $99.415 \pm 0.068$ | $96.832 \pm 0.263$ | $0.037 \pm 0.006$ | – | $0.085 \pm 0.006$ | $0.005 \pm 0.001$ | $6.70$ |
| | $\mathbf{H}_{Softmax+TS}$ | $90.485 \pm 0.359$ | $99.332 \pm 0.059$ | $95.982 \pm 0.340$ | $0.032 \pm 0.002$ | – | $0.079 \pm 0.003$ | $0.005 \pm 0.000$ | $6.50$ |
| | $\mathbf{H}_{Softmax+HB}$ | $89.460 \pm 0.597$ | $99.118 \pm 0.068$ | $94.560 \pm 0.257$ | $0.034 \pm 0.004$ | – | $0.081 \pm 0.005$ | $0.005 \pm 0.000$ | $9.00$ |
| | $\mathbf{H}_{ICSoftmax}^{Post}$ | $90.890 \pm 0.150$ | $99.153 \pm 0.325$ | $95.162 \pm 1.679$ | $0.025 \pm 0.009$ | – | $0.077 \pm 0.013$ | $0.004 \pm 0.001$ | $5.80$ |
| | $\mathbf{H}_{ALD+TS}$ | $90.760 \pm 0.573$ | $99.275 \pm 0.030$ | $95.614 \pm 0.104$ | $0.030 \pm 0.006$ | – | $0.073 \pm 0.005$ | $0.005 \pm 0.000$ | $5.60$ |
| | $\mathbf{H}_{ALD+HB}$ | $89.083 \pm 0.056$ | $99.100 \pm 0.079$ | $94.956 \pm 0.100$ | $0.033 \pm 0.006$ | – | $0.075 \pm 0.001$ | $0.005 \pm 0.000$ | $8.30$ |
| | $\mathbf{H}_{ICALD}^{Post}$ | $90.860 \pm 0.270$ | $99.152 \pm 0.284$ | $96.077 \pm 0.641$ | $0.025 \pm 0.008$ | – | $0.061 \pm 0.015$ | $0.004 \pm 0.001$ | $4.40$ |
| | $\mathbf{H}_{Softmax+MMD}$ | $90.587 \pm 0.248$ | $99.406 \pm 0.035$ | $96.449 \pm 0.175$ | $0.041 \pm 0.005$ | – | $0.115 \pm 0.018$ | $0.003 \pm 0.000$ | $7.20$ |
| | $\mathbf{H}_{ICSoftmax}^{Pre}$ | $90.005 \pm 0.545$ | $99.055 \pm 0.065$ | $95.173 \pm 0.246$ | $0.024 \pm 0.004$ | – | $0.076 \pm 0.009$ | $0.004 \pm 0.001$ | $6.50$ |
| | $\mathbf{H}_{ALD+MMD}$ | $89.875 \pm 0.569$ | $99.351 \pm 0.061$ | $96.079 \pm 0.281$ | $0.034 \pm 0.005$ | – | $0.096 \pm 0.022$ | $0.003 \pm 0.001$ | $8.40$ |
| | $\mathbf{H}_{ICALD}^{Pre}$ | $90.075 \pm 0.296$ | $99.051 \pm 0.035$ | $96.479 \pm 0.118$ | $0.022 \pm 0.006$ | – | $0.076 \pm 0.012$ | $0.004 \pm 0.001$ | $5.60$ |
| CIFAR-100 | $\mathbf{H}_{Softmax}$ | $67.405 \pm 0.484$ | $98.599 \pm 0.039$ | $73.403 \pm 0.546$ | $0.119 \pm 0.006$ | – | $0.365 \pm 0.036$ | $0.008 \pm 0.002$ | $8.20$ |
| | $\mathbf{H}_{ALD}$ | $68.313 \pm 0.380$ | $98.430 \pm 0.187$ | $74.880 \pm 0.500$ | $0.109 \pm 0.017$ | – | $0.299 \pm 0.010$ | $0.005 \pm 0.001$ | $5.70$ |
| | $\mathbf{H}_{Softmax+TS}$ | $67.405 \pm 0.484$ | $98.707 \pm 0.119$ | $72.560 \pm 1.505$ | $0.087 \pm 0.005$ | – | $0.268 \pm 0.024$ | $0.011 \pm 0.000$ | $6.40$ |
| | $\mathbf{H}_{Softmax+HB}$ | $65.260 \pm 1.174$ | $98.639 \pm 0.102$ | $71.179 \pm 1.422$ | $0.083 \pm 0.007$ | – | $0.240 \pm 0.026$ | $0.011 \pm 0.000$ | $6.80$ |
| | $\mathbf{H}_{ICSoftmax}^{Post}$ | $67.440 \pm 1.960$ | $98.105 \pm 0.141$ | $71.302 \pm 1.326$ | $0.091 \pm 0.014$ | – | $0.297 \pm 0.012$ | $0.005 \pm 0.002$ | $6.90$ |
| | $\mathbf{H}_{ALD+TS}$ | $68.313 \pm 0.380$ | $98.661 \pm 0.221$ | $74.688 \pm 0.262$ | $0.088 \pm 0.014$ | – | $0.216 \pm 0.008$ | $0.009 \pm 0.000$ | $4.70$ |
| | $\mathbf{H}_{ALD+HB}$ | $65.917 \pm 0.327$ | $98.279 \pm 0.461$ | $73.032 \pm 0.428$ | $0.090 \pm 0.012$ | – | $0.213 \pm 0.007$ | $0.009 \pm 0.000$ | $5.80$ |
| | $\mathbf{H}_{ICALD}^{Post}$ | $67.685 \pm 1.185$ | $98.096 \pm 0.076$ | $74.931 \pm 0.866$ | $0.081 \pm 0.015$ | – | $0.236 \pm 0.015$ | $0.005 \pm 0.002$ | $4.60$ |
| | $\mathbf{H}_{Softmax+MMD}$ | $66.532 \pm 0.304$ | $98.615 \pm 0.050$ | $73.001 \pm 0.457$ | $0.115 \pm 0.009$ | – | $0.352 \pm 0.052$ | $0.005 \pm 0.001$ | $6.70$ |
| | $\mathbf{H}_{ICSoftmax}^{Pre}$ | $67.245 \pm 0.415$ | $98.697 \pm 0.082$ | $73.558 \pm 0.210$ | $0.094 \pm 0.016$ | – | $0.291 \pm 0.049$ | $0.007 \pm 0.001$ | $6.10$ |
| | $\mathbf{H}_{ALD+MMD}$ | $67.920 \pm 0.870$ | $98.413 \pm 0.093$ | $73.972 \pm 0.363$ | $0.106 \pm 0.052$ | – | $0.283 \pm 0.081$ | $0.005 \pm 0.001$ | $5.90$ |
| | $\mathbf{H}_{ICALD}^{Pre}$ | $68.405 \pm 0.606$ | $98.778 \pm 0.039$ | $75.114 \pm 0.785$ | $0.079 \pm 0.006$ | – | $0.229 \pm 0.029$ | $0.006 \pm 0.001$ | $3.20$ |
| SST-2 | $\mathbf{H}_{Softmax}$ | $90.969 \pm 0.424$ | $97.159 \pm 0.210$ | $97.302 \pm 0.309$ | $0.054 \pm 0.011$ | – | $0.153 \pm 0.022$ | $0.073 \pm 0.008$ | $7.50$ |
| | $\mathbf{H}_{ALD}$ | $91.657 \pm 0.848$ | $97.305 \pm 0.513$ | $97.503 \pm 0.520$ | $0.042 \pm 0.006$ | – | $0.148 \pm 0.017$ | $0.069 \pm 0.013$ | $4.90$ |
| | $\mathbf{H}_{Softmax+TS}$ | $90.969 \pm 0.424$ | $97.054 \pm 0.140$ | $97.314 \pm 0.173$ | $0.059 \pm 0.015$ | – | $0.200 \pm 0.035$ | $0.113 \pm 0.021$ | $10.30$ |
| | $\mathbf{H}_{Softmax+HB}$ | $90.138 \pm 0.468$ | $96.216 \pm 0.283$ | $95.838 \pm 0.436$ | $0.038 \pm 0.020$ | – | $0.181 \pm 0.032$ | $0.100 \pm 0.010$ | $8.90$ |
| | $\mathbf{H}_{ICSoftmax}^{Post}$ | $90.970 \pm 0.516$ | $96.508 \pm 3.223$ | $97.263 \pm 0.515$ | $0.035 \pm 0.013$ | – | $0.148 \pm 0.004$ | $0.061 \pm 0.009$ | $5.30$ |
| | $\mathbf{H}_{ALD+TS}$ | $91.657 \pm 0.848$ | $96.637 \pm 0.150$ | $96.995 \pm 0.103$ | $0.048 \pm 0.007$ | – | $0.190 \pm 0.011$ | $0.075 \pm 0.032$ | $9.10$ |
| | $\mathbf{H}_{ALD+HB}$ | $90.838 \pm 0.919$ | $95.125 \pm 0.338$ | $94.622 \pm 0.578$ | $0.037 \pm 0.004$ | – | $0.185 \pm 0.008$ | $0.060 \pm 0.024$ | $9.40$ |
| | $\mathbf{H}_{ICALD}^{Post}$ | $90.766 \pm 0.747$ | $96.470 \pm 0.210$ | $97.363 \pm 0.501$ | $0.035 \pm 0.006$ | – | $0.137 \pm 0.007$ | $0.059 \pm 0.011$ | $5.10$ |
| | $\mathbf{H}_{Softmax+MMD}$ | $91.170 \pm 0.654$ | $96.988 \pm 0.305$ | $97.264 \pm 0.241$ | $0.043 \pm 0.009$ | – | $0.137 \pm 0.010$ | $0.055 \pm 0.006$ | $5.20$ |
| | $\mathbf{H}_{ICSoftmax}^{Pre}$ | $90.749 \pm 0.301$ | $97.095 \pm 0.217$ | $97.359 \pm 0.190$ | $0.034 \pm 0.009$ | – | $0.134 \pm 0.008$ | $0.057 \pm 0.008$ | $4.40$ |
| | $\mathbf{H}_{ALD+MMD}$ | $91.198 \pm 0.188$ | $96.910 \pm 0.083$ | $97.049 \pm 0.205$ | $0.036 \pm 0.011$ | – | $0.139 \pm 0.022$ | $0.049 \pm 0.030$ | $5.30$ |
| | $\mathbf{H}_{ICALD}^{Pre}$ | $90.979 \pm 0.516$ | $96.990 \pm 0.238$ | $97.311 \pm 0.061$ | $0.032 \pm 0.007$ | – | $0.129 \pm 0.013$ | $0.055 \pm 0.013$ | $4.00$ |
| SST-5 | $\mathbf{H}_{Softmax}$ | $53.224 \pm 0.595$ | $83.055 \pm 0.191$ | $51.692 \pm 1.062$ | $0.253 \pm 0.074$ | – | $0.406 \pm 0.073$ | $0.125 \pm 0.018$ | $10.00$ |
| | $\mathbf{H}_{ALD}$ | $53.839 \pm 0.324$ | $83.238 \pm 0.090$ | $51.989 \pm 0.516$ | $0.147 \pm 0.085$ | – | $0.382 \pm 0.048$ | $0.133 \pm 0.060$ | $6.20$ |
| | $\mathbf{H}_{Softmax+TS}$ | $53.224 \pm 0.595$ | $82.001 \pm 0.415$ | $51.647 \pm 1.782$ | $0.120 \pm 0.116$ | – | $0.519 \pm 0.192$ | $0.343 \pm 0.300$ | $9.80$ |
| | $\mathbf{H}_{Softmax+HB}$ | $51.086 \pm 0.736$ | $82.070 \pm 0.348$ | $51.518 \pm 1.631$ | $0.138 \pm 0.006$ | – | $0.387 \pm 0.240$ | $0.370 \pm 0.291$ | $10.00$ |
| | $\mathbf{H}_{ICSoftmax}^{Post}$ | $52.774 \pm 0.090$ | $82.670 \pm 1.978$ | $51.117 \pm 1.086$ | $0.116 \pm 0.002$ | – | $0.216 \pm 0.021$ | $0.102 \pm 0.007$ | $4.90$ |
| | $\mathbf{H}_{ALD+TS}$ | $53.839 \pm 0.324$ | $82.655 \pm 0.039$ | $51.099 \pm 0.630$ | $0.135 \pm 0.020$ | – | $0.328 \pm 0.036$ | $0.188 \pm 0.034$ | $7.50$ |
| | $\mathbf{H}_{ALD+HB}$ | $49.279 \pm 0.832$ | $81.250 \pm 0.343$ | $45.771 \pm 1.991$ | $0.126 \pm 0.012$ | – | $0.323 \pm 0.095$ | $0.192 \pm 0.024$ | $11.00$ |
| | $\mathbf{H}_{ICALD}^{Post}$ | $52.633 \pm 0.452$ | $82.378 \pm 2.219$ | $51.219 \pm 1.398$ | $0.123 \pm 0.002$ | – | $0.252 \pm 0.027$ | $0.088 \pm 0.004$ | $4.40$ |
| | $\mathbf{H}_{Softmax+MMD}$ | $53.416 \pm 0.345$ | $82.695 \pm 0.272$ | $51.297 \pm 0.764$ | $0.209 \pm 0.053$ | – | $0.324 \pm 0.066$ | $0.081 \pm 0.009$ | $7.00$ |
| | $\mathbf{H}_{ICSoftmax}^{Pre}$ | $53.552 \pm 0.023$ | $82.715 \pm 0.041$ | $51.143 \pm 0.831$ | $0.118 \pm 0.006$ | – | $0.297 \pm 0.066$ | $0.107 \pm 0.008$ | $6.20$ |
| | $\mathbf{H}_{ALD+MMD}$ | $52.828 \pm 0.825$ | $82.641 \pm 0.743$ | $51.094 \pm 1.239$ | $0.132 \pm 0.130$ | – | $0.330 \pm 0.117$ | $0.078 \pm 0.046$ | $6.70$ |
| | $\mathbf{H}_{ICALD}^{Pre}$ | $53.364 \pm 0.277$ | $83.000 \pm 0.847$ | $51.904 \pm 1.635$ | $0.128 \pm 0.132$ | – | $0.264 \pm 0.108$ | $0.097 \pm 0.013$ | $4.40$ |
| Overall | $\mathbf{H}_{Softmax}$ | $7.28$ | $4.33$ | $7.17$ | $7.83$ | $8.00$ | $7.06$ | $5.89$ | $6.79$ |
| | $\mathbf{H}_{ALD}$ | $2.94$ | $3.78$ | $3.50$ | $6.83$ | $3.60$ | $6.72$ | $4.83$ | $4.60$ |
| | $\mathbf{H}_{Softmax+TS}$ | $7.28$ | $5.89$ | $7.89$ | $7.11$ | $9.40$ | $8.22$ | $8.33$ | $7.73$ |
| | $\mathbf{H}_{Softmax+HB}$ | $10.22$ | $10.11$ | $11.11$ | $6.94$ | $10.40$ | $8.28$ | $6.06$ | $9.16$ |
| | $\mathbf{H}_{ICSoftmax}^{Post}$ | $4.89$ | $7.89$ | $7.17$ | $4.44$ | $6.60$ | $4.11$ | $3.56$ | $5.52$ |
| | $\mathbf{H}_{ALD+TS}$ | $2.94$ | $6.67$ | $5.83$ | $6.72$ | $9.40$ | $7.56$ | $7.67$ | $6.97$ |
| | $\mathbf{H}_{ALD+HB}$ | $10.67$ | $9.22$ | $11.33$ | $5.89$ | $4.80$ | $6.56$ | $6.72$ | $7.89$ |
| | $\mathbf{H}_{ICALD}^{Post}$ | $4.94$ | $5.67$ | $3.89$ | $3.39$ | $4.20$ | $2.50$ | $1.94$ | $3.79$ |
| | $\mathbf{H}_{Softmax+MMD}$ | $7.00$ | $4.78$ | $7.00$ | $6.83$ | $7.20$ | $6.89$ | $2.28$ | $6.00$ |
| | $\mathbf{H}_{ICSoftmax}^{Pre}$ | $6.61$ | $7.33$ | $8.72$ | $4.72$ | $5.40$ | $5.78$ | $3.22$ | $5.97$ |
| | $\mathbf{H}_{ALD+MMD}$ | $5.39$ | $5.56$ | $4.72$ | $6.28$ | $5.60$ | $6.06$ | $1.39$ | $5.00$ |
| | $\mathbf{H}_{ICALD}^{Pre}$ | $3.83$ | $4.78$ | $2.67$ | $2.00$ | $1.20$ | $2.39$ | $2.00$ | $2.98$ |

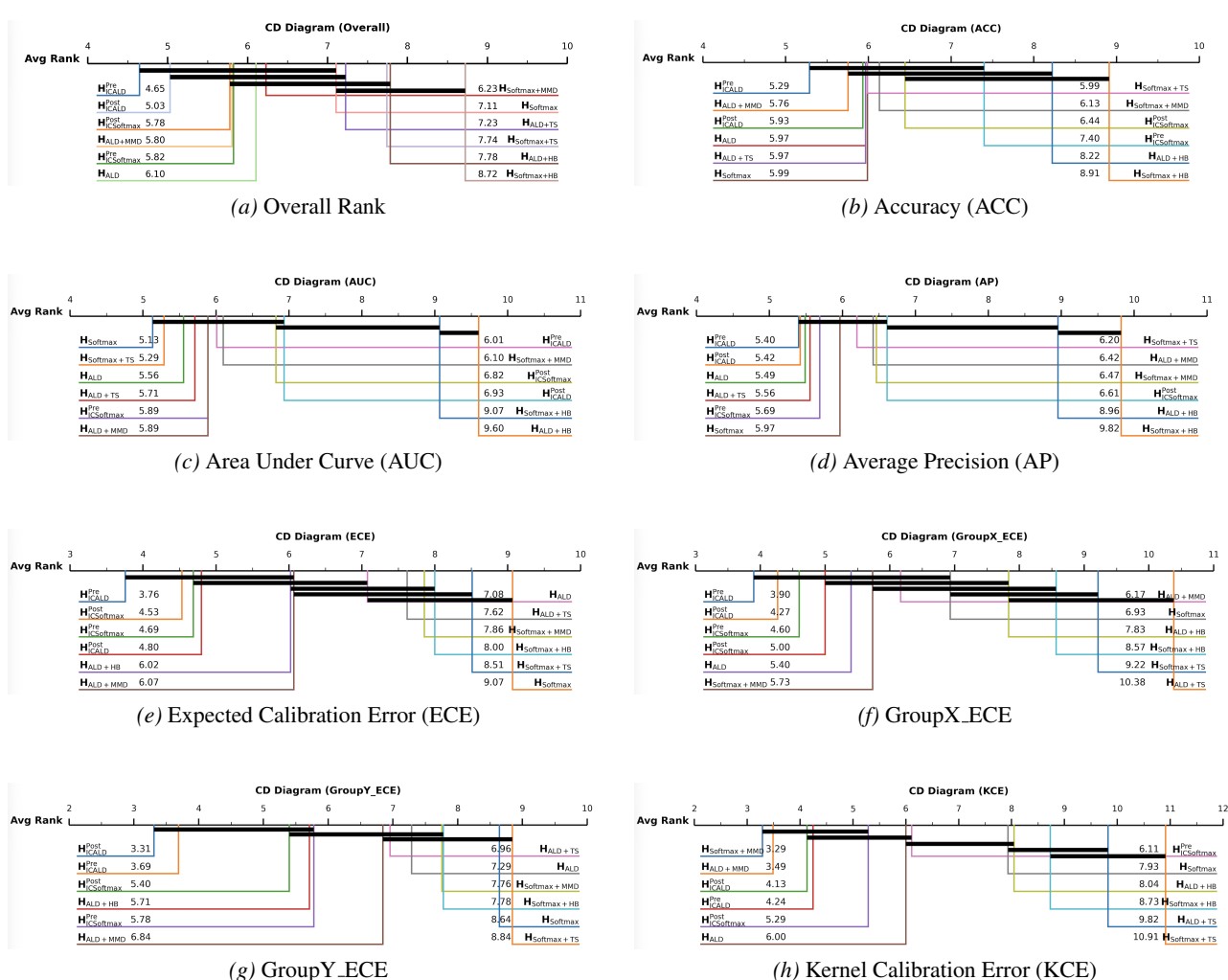

*Figure 5.* **Critical Difference (CD) diagrams comparing classifiers across** $N = 45$ **experimental settings.** Each subplot reports the Friedman average rank of each method (lower is better) under a specific metric: (a) overall rank aggregated across metrics, (b–d) *discriminative* performance (ACC, AUC, AP), and (e–h) *calibration* quality (ECE, GroupX_ECE, GroupY_ECE, KCE). The critical difference (CD = 2.48) is computed via the Nemenyi test at $\alpha = 0.05$. Thick horizontal bars connect methods that are ***not*** statistically significantly different (*i.e.*, their rank difference is within the critical difference threshold). Notably, our proposed *pre-calibration* ($\mathbf{H}_{\text{ICALD}}^{\text{Pre}}$) and *post-calibration* ($\mathbf{H}_{\text{ICALD}}^{\text{Post}}$) strategies consistently secure the top rankings, statistically outperforming the standard Softmax baseline and various post-hoc calibration methods across both reliability and accuracy dimensions.

## C.5. Additional results

We provide an expanded comparison with eight additional baselines to further assess the effectiveness of $\mathbf{H}_{\text{ICALD}}^{\text{Pre}}$. The additional methods include recent calibration-aware training objectives, post-hoc multiclass calibration, and uncertainty-estimation models: $\mathbf{H}_{\text{CWLS}}$ (Jung et al., 2023), $\mathbf{H}_{\text{TNA}}$ (Cho & Youn, 2024), $\mathbf{H}_{\text{FC}}$ (Tao et al., 2025), $\mathbf{H}_{\text{DFL}}$ (Tao et al., 2023), $\mathbf{H}_{\text{BSCE-GRA}}$ (Lin et al., 2025), $\mathbf{H}_{\text{EDL}}$ (Sensoy et al., 2018), $\mathbf{H}_{\text{NatPN}}$ (Charpentier et al., 2022), and $\mathbf{H}_{\text{Dirichlet}}$ (Kull et al., 2019). All methods are evaluated on the five tabular datasets using the same train/validation/test protocol as in the main experiments. We report three predictive metrics, *i.e.*, ACC, AUC, and AP, and six calibration or uncertainty-sensitive metrics, *i.e.*, ECE, GroupX_ECE, GroupY_ECE, Classwise_ECE, Brier score, and KCE. For ACC, AUC, and AP, higher values are better; for all calibration and error metrics, lower values are better.

Table 9 reports the full quantitative results. We observe that $\mathbf{H}_{\text{ICALD}}^{\text{Pre}}$ achieves the best average rank on every dataset block: 2.67 on BREAST-CANCER, 2.89 on HEART-DISEASE, 2.50 on ONLINE-SHOPPERS, 1.33 on DRY-BEAN, and 2.56 on ADULT. In the overall comparison, $\mathbf{H}_{\text{ICALD}}^{\text{Pre}}$ obtains the best aggregate average rank of 2.17, outperforming $\mathbf{H}_{\text{Softmax}}$ (7.23) and all additional baselines. Metric-wise, $\mathbf{H}_{\text{ICALD}}^{\text{Pre}}$ ranks first on ACC, AUC, AP, ECE, GroupX_ECE, Brier score, and KCE. Although $\mathbf{H}_{\text{EDL}}$ achieves the best overall rank on GroupY_ECE and $\mathbf{H}_{\text{Dirichlet}}$ achieves the best overall rank on Classwise ECE, their aggregate ranks are worse than $\mathbf{H}_{\text{ICALD}}^{\text{Pre}}$. This suggests that these baselines can be strong on specific calibration criteria, whereas $\mathbf{H}_{\text{ICALD}}^{\text{Pre}}$ provides a more balanced improvement across accuracy, average calibration, group-level calibration, and distribution-level calibration.

Figure 6 provides the corresponding reliability diagrams. The blue bars represent empirical accuracy within confidence bins, the diagonal line denotes perfect calibration, and the orange gaps visualize the confidence-accuracy discrepancy. Compared with the additional baselines, $\mathbf{H}_{\text{ICALD}}^{\text{Pre}}$ generally produces smaller and more stable gaps across confidence ranges, especially on larger datasets such as ONLINE-SHOPPERS, DRY-BEAN, and ADULT. These qualitative results are consistent with the quantitative improvements in ECE, Brier score, and KCE reported in Table 9.

Overall, the expanded comparison confirms that the advantage of $\mathbf{H}_{\text{ICALD}}^{\text{Pre}}$ is not limited to the baselines considered in the main text. By combining a flexible distributional parameterization with individualized calibration, our method provides a robust trade-off between discriminative performance and probabilistic reliability across multiple complementary evaluation criteria.

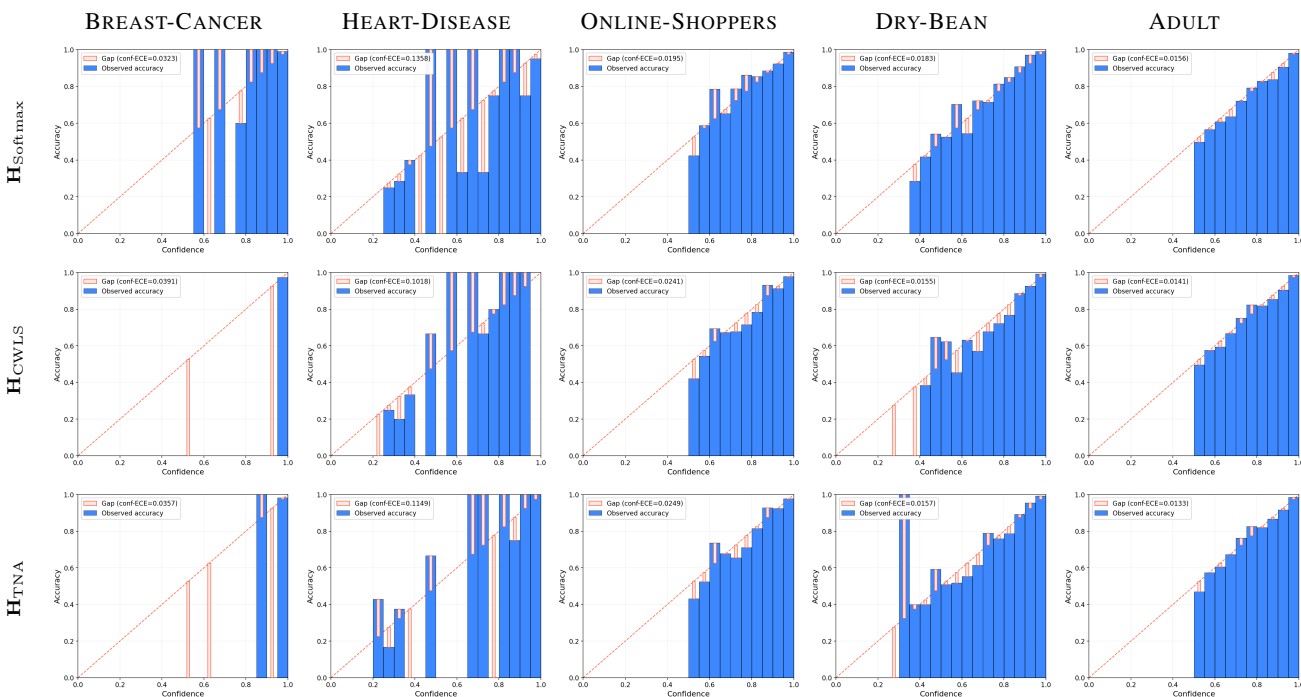

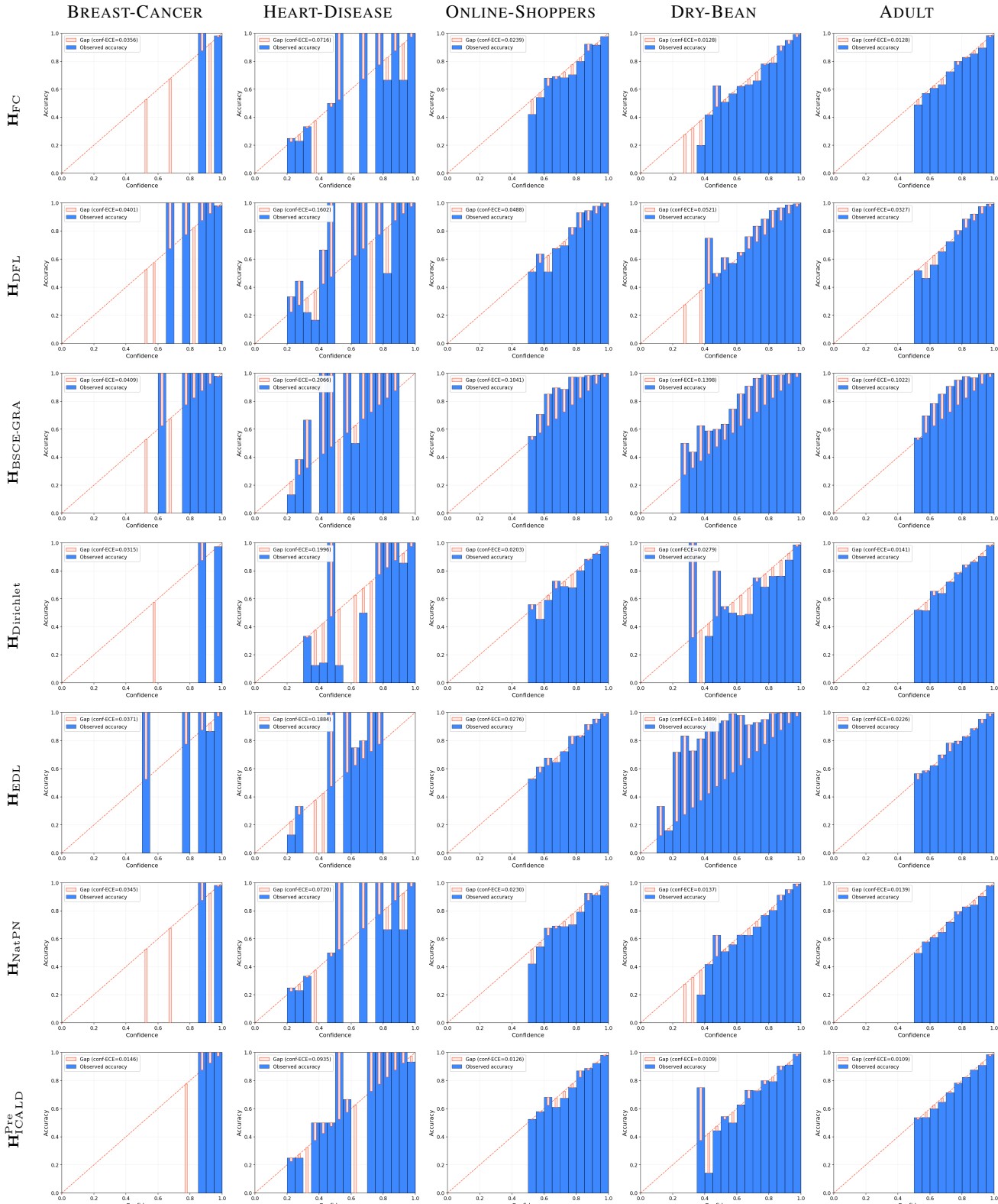

*Figure 6.* Reliability diagram comparison, including 8 additional baselines across five tabular datasets.

*Table 9.* Overall performance comparison, including 8 additional baselines and 3 additional evaluation metrics. Best values in each column are highlighted by **bold yellow**.

| Dataset | Model | ACC | AUC | AP | ECE | GroupX.ECE | GroupY.ECE | Classwise.ECE | Brier.score | KCE | Avg Rank |
|---|---|---|---|---|---|---|---|---|---|---|---|
| BREAST-CANCER | H_Softmax | 97.719 ± 0.702 | 99.286 ± 0.320 | 99.205 ± 0.384 | 0.030 ± 0.009 | 0.093 ± 0.023 | 0.060 ± 0.018 | 0.036 ± 0.010 | 0.052 ± 0.019 | 0.027 ± 0.005 | 7.06 |
| | H_CWLS | 96.667 ± 1.441 | 98.955 ± 0.596 | 98.838 ± 0.654 | 0.028 ± 0.012 | 0.085 ± 0.024 | 0.055 ± 0.017 | **0.033 ± 0.010** | 0.048 ± 0.023 | 0.026 ± 0.012 | 4.72 |
| | H_TNA | 96.491 ± 1.641 | 98.948 ± 0.595 | 98.834 ± 0.662 | 0.027 ± 0.010 | 0.084 ± 0.026 | 0.057 ± 0.021 | **0.033 ± 0.010** | 0.049 ± 0.019 | 0.027 ± 0.011 | 5.17 |
| | H_FC | 96.491 ± 1.641 | 98.968 ± 0.586 | 98.840 ± 0.656 | 0.028 ± 0.011 | 0.084 ± 0.026 | 0.058 ± 0.023 | 0.034 ± 0.008 | 0.048 ± 0.019 | 0.026 ± 0.007 | 4.94 |
| | H_DFL | 96.316 ± 1.144 | 98.750 ± 0.600 | 98.697 ± 0.434 | 0.029 ± 0.008 | 0.089 ± 0.022 | 0.059 ± 0.019 | 0.034 ± 0.008 | 0.050 ± 0.012 | 0.034 ± 0.016 | 8.39 |
| | H_BSCE-GRA | 96.842 ± 1.000 | 98.849 ± 0.736 | 98.776 ± 0.675 | 0.028 ± 0.004 | 0.090 ± 0.023 | 0.059 ± 0.022 | **0.033 ± 0.016** | 0.050 ± 0.017 | 0.050 ± 0.028 | 6.83 |
| | H_Dirichlet | 96.667 ± 0.961 | 98.277 ± 1.150 | 98.068 ± 1.373 | 0.024 ± 0.009 | 0.085 ± 0.027 | 0.058 ± 0.018 | **0.033 ± 0.010** | 0.050 ± 0.017 | 0.027 ± 0.006 | 6.28 |
| | H_EDL | 97.544 ± 0.734 | 99.438 ± 0.482 | 99.242 ± 0.617 | 0.026 ± 0.013 | 0.097 ± 0.010 | 0.055 ± 0.006 | 0.035 ± 0.010 | **0.043 ± 0.013** | **0.024 ± 0.010** | 3.67 |
| | H_NatPN | 96.491 ± 1.641 | 98.763 ± 0.766 | 98.727 ± 0.750 | 0.028 ± 0.010 | 0.084 ± 0.025 | 0.056 ± 0.022 | **0.033 ± 0.010** | 0.048 ± 0.019 | 0.027 ± 0.009 | 5.50 |
| | $H^{Pre}_{ICALD}$ | **97.995 ± 0.895** | **99.565 ± 0.161** | **99.279 ± 0.238** | **0.023 ± 0.007** | **0.080 ± 0.026** | **0.048 ± 0.025** | 0.035 ± 0.014 | 0.049 ± 0.024 | 0.025 ± 0.007 | **2.67** |
| HEART-DISEASE | H_Softmax | 59.333 ± 3.590 | 79.141 ± 1.374 | 41.130 ± 2.677 | 0.100 ± 0.020 | 0.315 ± 0.039 | 0.443 ± 0.097 | 0.076 ± 0.008 | 0.475 ± 0.018 | 0.295 ± 0.024 | 7.67 |
| | H_CWLS | 56.667 ± 3.118 | 79.073 ± 0.908 | **42.853 ± 1.275** | 0.093 ± 0.022 | 0.305 ± 0.021 | 0.426 ± 0.085 | 0.068 ± 0.010 | 0.465 ± 0.025 | 0.275 ± 0.037 | 4.50 |
| | H_TNA | 59.667 ± 3.206 | 79.120 ± 1.187 | 41.603 ± 1.843 | 0.092 ± 0.019 | 0.309 ± 0.028 | 0.479 ± 0.127 | 0.069 ± 0.008 | 0.473 ± 0.018 | 0.268 ± 0.040 | 5.44 |
| | H_FC | 58.667 ± 3.613 | **79.298 ± 1.073** | 42.153 ± 1.245 | 0.094 ± 0.028 | 0.308 ± 0.035 | 0.462 ± 0.112 | 0.064 ± 0.007 | 0.474 ± 0.016 | 0.265 ± 0.027 | 4.67 |
| | H_DFL | 58.333 ± 4.249 | 78.616 ± 2.384 | 41.242 ± 2.827 | 0.095 ± 0.035 | 0.308 ± 0.032 | 0.437 ± 0.115 | 0.072 ± 0.007 | **0.462 ± 0.040** | 0.274 ± 0.030 | 5.44 |
| | H_BSCE-GRA | 59.000 ± 3.028 | 78.182 ± 2.967 | 42.341 ± 4.406 | 0.097 ± 0.050 | 0.309 ± 0.051 | 0.436 ± 0.061 | **0.063 ± 0.010** | 0.465 ± 0.059 | 0.283 ± 0.025 | 5.28 |
| | H_Dirichlet | 56.333 ± 1.826 | 76.910 ± 5.052 | 40.992 ± 3.747 | 0.089 ± 0.016 | **0.301 ± 0.024** | 0.423 ± 0.096 | 0.065 ± 0.005 | 0.467 ± 0.030 | 0.285 ± 0.036 | 5.67 |
| | H_EDL | 57.333 ± 4.944 | 78.264 ± 2.717 | 40.812 ± 4.230 | 0.096 ± 0.042 | 0.304 ± 0.010 | **0.403 ± 0.183** | 0.074 ± 0.011 | 0.472 ± 0.078 | 0.284 ± 0.052 | 6.50 |
| | H_NatPN | 58.667 ± 3.613 | 78.530 ± 1.785 | 40.744 ± 2.642 | 0.096 ± 0.026 | 0.310 ± 0.031 | 0.455 ± 0.106 | 0.072 ± 0.007 | 0.466 ± 0.017 | 0.376 ± 0.043 | 7.50 |
| | $H^{Pre}_{ICALD}$ | **60.000 ± 4.082** | 79.211 ± 2.983 | 42.023 ± 1.864 | **0.087 ± 0.021** | 0.309 ± 0.044 | 0.404 ± 0.109 | 0.067 ± 0.015 | 0.463 ± 0.028 | **0.262 ± 0.044** | **2.89** |
| ONLINE-SHOPPERS | H_Softmax | 89.749 ± 0.339 | 91.915 ± 0.534 | 70.777 ± 1.511 | 0.018 ± 0.002 | 0.055 ± 0.005 | 0.491 ± 0.005 | 0.023 ± 0.003 | 0.150 ± 0.008 | 0.211 ± 0.011 | 6.61 |
| | H_CWLS | 89.440 ± 0.495 | 91.692 ± 1.183 | 70.299 ± 3.489 | 0.017 ± 0.005 | 0.049 ± 0.010 | 0.453 ± 0.028 | 0.022 ± 0.002 | 0.150 ± 0.008 | 0.196 ± 0.032 | 6.06 |
| | H_TNA | 89.457 ± 0.477 | 91.694 ± 1.185 | 70.304 ± 3.493 | 0.017 ± 0.005 | 0.051 ± 0.008 | 0.456 ± 0.025 | 0.022 ± 0.003 | 0.150 ± 0.008 | 0.194 ± 0.027 | 5.72 |
| | H_FC | 89.440 ± 0.495 | 91.703 ± 1.162 | 70.395 ± 3.373 | 0.017 ± 0.004 | 0.050 ± 0.008 | 0.456 ± 0.026 | 0.023 ± 0.003 | 0.150 ± 0.008 | 0.196 ± 0.034 | 6.11 |
| | H_DFL | 89.716 ± 0.529 | 92.185 ± 0.737 | **72.317 ± 2.252** | **0.013 ± 0.006** | 0.085 ± 0.009 | 0.459 ± 0.044 | 0.059 ± 0.007 | 0.152 ± 0.006 | 0.195 ± 0.026 | 5.44 |
| | H_BSCE-GRA | 89.667 ± 0.498 | 92.227 ± 0.604 | 72.058 ± 1.989 | 0.014 ± 0.008 | 0.153 ± 0.016 | 0.469 ± 0.039 | 0.104 ± 0.008 | 0.179 ± 0.005 | 0.189 ± 0.013 | 5.89 |
| | H_Dirichlet | 89.619 ± 0.447 | 91.692 ± 1.183 | 70.299 ± 3.489 | 0.016 ± 0.004 | 0.051 ± 0.008 | 0.438 ± 0.042 | 0.022 ± 0.003 | 0.150 ± 0.007 | 0.185 ± 0.022 | 4.83 |
| | H_EDL | 89.732 ± 0.372 | 91.424 ± 0.980 | 70.953 ± 2.241 | 0.016 ± 0.004 | 0.062 ± 0.009 | **0.420 ± 0.026** | 0.038 ± 0.006 | 0.150 ± 0.006 | 0.189 ± 0.071 | 5.06 |
| | H_NatPN | 89.440 ± 0.495 | 91.665 ± 1.227 | 70.232 ± 3.343 | 0.017 ± 0.004 | 0.051 ± 0.008 | 0.453 ± 0.023 | 0.022 ± 0.002 | 0.150 ± 0.008 | 0.194 ± 0.025 | 6.39 |
| | $H^{Pre}_{ICALD}$ | **90.065 ± 0.445** | **92.270 ± 0.609** | 72.142 ± 1.611 | 0.014 ± 0.002 | **0.047 ± 0.003** | 0.477 ± 0.043 | **0.020 ± 0.003** | **0.147 ± 0.005** | **0.182 ± 0.043** | **2.50** |
| DRY-BEAN | H_Softmax | 92.295 ± 0.383 | 99.438 ± 0.022 | 97.817 ± 0.099 | 0.017 ± 0.004 | 0.045 ± 0.007 | 0.083 ± 0.022 | 0.011 ± 0.002 | 0.115 ± 0.006 | 0.053 ± 0.002 | 7.00 |
| | H_CWLS | 92.038 ± 0.512 | 99.441 ± 0.037 | 97.786 ± 0.185 | 0.016 ± 0.006 | 0.038 ± 0.005 | 0.064 ± 0.013 | 0.009 ± 0.001 | 0.114 ± 0.006 | 0.050 ± 0.001 | 5.17 |
| | H_TNA | 92.134 ± 0.586 | 99.427 ± 0.049 | 97.737 ± 0.275 | 0.014 ± 0.002 | 0.043 ± 0.009 | 0.085 ± 0.017 | 0.010 ± 0.001 | 0.114 ± 0.008 | 0.045 ± 0.004 | 6.00 |
| | H_FC | 92.281 ± 0.601 | 99.441 ± 0.031 | 97.820 ± 0.159 | 0.014 ± 0.004 | 0.044 ± 0.008 | 0.077 ± 0.020 | 0.011 ± 0.002 | 0.114 ± 0.006 | 0.044 ± 0.003 | 4.94 |
| | H_DFL | 92.266 ± 0.538 | 99.401 ± 0.018 | 97.725 ± 0.149 | 0.015 ± 0.008 | 0.042 ± 0.017 | 0.082 ± 0.021 | 0.011 ± 0.002 | 0.113 ± 0.006 | 0.043 ± 0.006 | 5.83 |
| | H_BSCE-GRA | 91.891 ± 0.392 | 99.384 ± 0.022 | 97.593 ± 0.114 | 0.016 ± 0.013 | 0.178 ± 0.012 | 0.082 ± 0.029 | 0.011 ± 0.003 | 0.113 ± 0.005 | 0.044 ± 0.009 | 7.44 |
| | H_Dirichlet | 91.972 ± 0.400 | 99.417 ± 0.057 | 97.595 ± 0.280 | 0.009 ± 0.006 | 0.038 ± 0.006 | 0.067 ± 0.015 | 0.009 ± 0.001 | 0.106 ± 0.006 | 0.042 ± 0.002 | 4.17 |
| | H_EDL | 79.177 ± 6.902 | 94.476 ± 2.439 | 77.457 ± 9.223 | 0.016 ± 0.049 | 0.289 ± 0.040 | 0.072 ± 0.120 | 0.009 ± 0.004 | 0.114 ± 0.037 | 0.039 ± 0.023 | 6.94 |
| | H_NatPN | 92.281 ± 0.601 | 99.414 ± 0.034 | 97.798 ± 0.169 | 0.014 ± 0.004 | 0.045 ± 0.008 | 0.078 ± 0.022 | 0.011 ± 0.002 | 0.115 ± 0.006 | 0.045 ± 0.003 | 6.17 |
| | $H^{Pre}_{ICALD}$ | **92.545 ± 0.570** | **99.443 ± 0.034** | **97.903 ± 0.153** | **0.008 ± 0.002** | **0.031 ± 0.002** | **0.040 ± 0.010** | **0.007 ± 0.002** | **0.105 ± 0.027** | **0.035 ± 0.004** | **1.33** |
| ADULT | H_Softmax | 85.351 ± 0.157 | 90.791 ± 0.259 | 78.118 ± 0.569 | 0.016 ± 0.003 | 0.048 ± 0.006 | 0.409 ± 0.011 | 0.021 ± 0.004 | 0.205 ± 0.005 | 0.243 ± 0.006 | 7.83 |
| | H_CWLS | 85.074 ± 0.257 | **90.922 ± 0.138** | 78.394 ± 0.478 | 0.014 ± 0.003 | 0.045 ± 0.004 | 0.408 ± 0.004 | 0.019 ± 0.003 | 0.204 ± 0.002 | 0.222 ± 0.003 | 5.17 |
| | H_TNA | 85.102 ± 0.509 | 90.830 ± 0.402 | 78.126 ± 0.889 | 0.014 ± 0.003 | 0.041 ± 0.005 | 0.404 ± 0.011 | 0.020 ± 0.003 | 0.201 ± 0.005 | 0.237 ± 0.014 | 6.06 |
| | H_FC | 85.155 ± 0.480 | 90.867 ± 0.360 | **78.475 ± 0.756** | **0.011 ± 0.003** | **0.035 ± 0.003** | 0.405 ± 0.012 | 0.018 ± 0.004 | 0.199 ± 0.005 | 0.228 ± 0.013 | 3.17 |
| | H_DFL | 84.993 ± 0.309 | 90.523 ± 0.481 | 77.440 ± 0.874 | 0.014 ± 0.006 | 0.049 ± 0.013 | 0.445 ± 0.018 | 0.020 ± 0.009 | 0.202 ± 0.003 | 0.224 ± 0.005 | 7.94 |
| | H_BSCE-GRA | 84.847 ± 0.240 | 90.544 ± 0.329 | 77.512 ± 0.792 | 0.014 ± 0.004 | 0.038 ± 0.005 | 0.347 ± 0.014 | 0.018 ± 0.003 | 0.204 ± 0.003 | 0.225 ± 0.007 | 6.50 |
| | H_Dirichlet | 85.132 ± 0.362 | 90.832 ± 0.400 | 78.129 ± 0.908 | 0.012 ± 0.005 | 0.037 ± 0.006 | 0.346 ± 0.029 | **0.017 ± 0.002** | 0.198 ± 0.004 | 0.224 ± 0.019 | 3.00 |
| | H_EDL | 84.946 ± 0.720 | 90.530 ± 0.601 | 77.403 ± 1.605 | 0.015 ± 0.002 | 0.046 ± 0.005 | **0.306 ± 0.023** | 0.019 ± 0.003 | 0.200 ± 0.007 | 0.226 ± 0.010 | 6.89 |
| | H_NatPN | 85.152 ± 0.462 | 90.731 ± 0.382 | 77.584 ± 0.767 | 0.013 ± 0.003 | 0.037 ± 0.004 | 0.409 ± 0.018 | 0.020 ± 0.003 | 0.202 ± 0.005 | 0.226 ± 0.013 | 5.89 |
| | $H^{Pre}_{ICALD}$ | **86.205 ± 0.368** | 90.718 ± 0.285 | 78.268 ± 0.602 | 0.013 ± 0.003 | 0.037 ± 0.007 | 0.312 ± 0.007 | **0.017 ± 0.003** | **0.196 ± 0.005** | **0.218 ± 0.006** | **2.56** |
| **Overall** | H_Softmax | 2.20 | 3.80 | 4.80 | 10.00 | 8.50 | 8.90 | 8.90 | 8.90 | 9.10 | 7.23 |
| | H_CWLS | 7.50 | 4.20 | 4.30 | 6.50 | 4.00 | 3.80 | 4.60 | 5.30 | 5.60 | 5.09 |
| | H_TNA | 5.80 | 5.00 | 5.80 | 5.00 | 5.20 | 7.10 | 5.00 | 6.00 | 6.30 | 5.69 |
| | H_FC | 5.80 | 2.90 | 3.20 | 4.80 | 3.50 | 6.40 | 5.20 | 5.30 | 5.50 | 4.73 |
| | H_DFL | 6.80 | 7.20 | 6.40 | 5.70 | 6.90 | 7.80 | 7.70 | 5.60 | 5.50 | 6.62 |
| | H_BSCE-GRA | 6.40 | 7.00 | 5.80 | 6.50 | 7.80 | 6.60 | 5.00 | 6.70 | 6.00 | 6.42 |
| | H_Dirichlet | 6.90 | 7.30 | 7.70 | 2.50 | 3.40 | 3.50 | **2.70** | 4.60 | 4.60 | 4.80 |
| | H_EDL | 6.60 | 7.80 | 7.00 | 6.40 | 7.60 | **1.90** | 6.80 | 4.70 | 4.00 | 5.87 |
| | H_NatPN | 6.00 | 7.40 | 7.80 | 5.80 | 5.50 | 6.00 | 5.90 | 5.80 | 7.20 | 6.38 |
| | $H^{Pre}_{ICALD}$ | **1.00** | **2.40** | **2.20** | **1.80** | **2.60** | 3.00 | 3.20 | **2.10** | **1.20** | **2.17** |

## C.6. Case studies

**Case Study I: Sample-wise Gumbel Scale and Fixed-scale bias of Softmax.**

Although $\mathbf{H}_{\text{Softmax}}$ can represent arbitrary categorical distributions through a sufficiently expressive logit function, its random-utility interpretation still imposes a fixed-scale latent noise model. Specifically, (1) shows that $\mathbf{H}_{\text{Softmax}}$ corresponds to a Gumbel utility model with $\beta = 1$, where only the location parameters are learned. Therefore, the limitation we study here is not a strict representational impossibility, but rather a parameterization and optimization bias: the logits must simultaneously encode both class discrimination and predictive confidence, without an explicit sample-wise uncertainty parameter. This coupling can make it difficult to model heteroscedastic uncertainty and may contribute to *miscalibration* under cross-entropy training.

To examine the effect of this fixed-scale assumption in isolation, we introduce a minimal diagnostic variant, denoted $\mathbf{H}_{\text{Gumbel}}$. This model preserves the same classification pipeline as $\mathbf{H}_{\text{Softmax}}$, but relaxes only the homoscedastic scale constraint by learning a sample-wise Gumbel scale $\beta(x) > 0$. Specifically, for each input $x$, we define

$$U_c(x) = z_c(x) + \epsilon_c, \qquad \epsilon_c \sim \text{Gumbel}(0, \beta(x)), \tag{90}$$

which yields the predictive probability

$$\hat{p}_c(x) = \frac{\exp\left(z_c(x)/\beta(x)\right)}{\sum_{j=1}^{C} \exp\left(z_j(x)/\beta(x)\right)}. \tag{91}$$

Here, $z_c(x)$ models the relative class utility, while $\beta(x)$ acts as a sample-specific uncertainty or temperature parameter. Larger $\beta(x)$ produces a flatter predictive distribution, corresponding to higher uncertainty, whereas smaller $\beta(x)$ sharpens the distribution. By changing only the scale parameterization while keeping the rest of the classifier unchanged, $\mathbf{H}_{\text{Gumbel}}$ allows us to directly test whether an explicit sample-wise uncertainty degree of freedom improves calibration beyond the standard fixed-scale Softmax formulation.

*Table 10.* Performance comparison between $\mathbf{H}_{\text{Softmax}}$ and $\mathbf{H}_{\text{Gumbel}}$ across five tabular datasets. Best values in each dataset block are highlighted by **bold yellow**.

| Dataset | Model | ACC | AUC | AP | ECE | GroupX_ECE | GroupY_ECE | Classwise_ECE | Brier_score | KCE |
|---|---|---|---|---|---|---|---|---|---|---|
| Breast-Cancer | $\mathbf{H}_{\text{Softmax}}$ | $97.719 \pm 0.702$ | $99.286 \pm 0.320$ | $99.205 \pm 0.384$ | $0.030 \pm 0.009$ | $0.093 \pm 0.023$ | $\mathbf{0.060 \pm 0.018}$ | $0.036 \pm 0.010$ | $0.052 \pm 0.019$ | $0.027 \pm 0.005$ |
| | $\mathbf{H}_{\text{Gumbel}}$ | $\mathbf{97.967 \pm 0.509}$ | $\mathbf{99.340 \pm 0.526}$ | $\mathbf{99.216 \pm 0.662}$ | $\mathbf{0.026 \pm 0.010}$ | $\mathbf{0.089 \pm 0.025}$ | $0.061 \pm 0.018$ | $\mathbf{0.034 \pm 0.012}$ | $\mathbf{0.051 \pm 0.021}$ | $\mathbf{0.026 \pm 0.003}$ |
| Heart-Disease | $\mathbf{H}_{\text{Softmax}}$ | $59.333 \pm 3.590$ | $\mathbf{79.141 \pm 1.374}$ | $41.130 \pm 2.677$ | $0.100 \pm 0.020$ | $0.315 \pm 0.039$ | $\mathbf{0.443 \pm 0.097}$ | $0.076 \pm 0.008$ | $0.475 \pm 0.018$ | $0.295 \pm 0.024$ |
| | $\mathbf{H}_{\text{Gumbel}}$ | $\mathbf{59.667 \pm 3.496}$ | $78.871 \pm 2.644$ | $\mathbf{41.189 \pm 3.084}$ | $\mathbf{0.098 \pm 0.019}$ | $\mathbf{0.310 \pm 0.017}$ | $0.449 \pm 0.064$ | $\mathbf{0.074 \pm 0.008}$ | $\mathbf{0.472 \pm 0.021}$ | $\mathbf{0.289 \pm 0.059}$ |
| Online-Shoppers | $\mathbf{H}_{\text{Softmax}}$ | $\mathbf{89.749 \pm 0.339}$ | $\mathbf{91.915 \pm 0.534}$ | $\mathbf{70.777 \pm 1.511}$ | $0.018 \pm 0.002$ | $0.055 \pm 0.005$ | $0.491 \pm 0.010$ | $0.023 \pm 0.003$ | $\mathbf{0.150 \pm 0.008}$ | $0.211 \pm 0.011$ |
| | $\mathbf{H}_{\text{Gumbel}}$ | $89.651 \pm 0.359$ | $91.398 \pm 0.884$ | $69.811 \pm 2.916$ | $\mathbf{0.017 \pm 0.003}$ | $\mathbf{0.046 \pm 0.005}$ | $\mathbf{0.264 \pm 0.020}$ | $\mathbf{0.022 \pm 0.002}$ | $0.152 \pm 0.006$ | $\mathbf{0.206 \pm 0.023}$ |
| Dry-Bean | $\mathbf{H}_{\text{Softmax}}$ | $92.295 \pm 0.383$ | $99.438 \pm 0.022$ | $\mathbf{97.817 \pm 0.099}$ | $0.017 \pm 0.004$ | $0.045 \pm 0.007$ | $0.083 \pm 0.022$ | $0.011 \pm 0.002$ | $0.115 \pm 0.006$ | $0.053 \pm 0.002$ |
| | $\mathbf{H}_{\text{Gumbel}}$ | $\mathbf{92.314 \pm 0.324}$ | $\mathbf{99.443 \pm 0.021}$ | $97.817 \pm 0.086$ | $\mathbf{0.015 \pm 0.005}$ | $\mathbf{0.040 \pm 0.005}$ | $\mathbf{0.075 \pm 0.019}$ | $\mathbf{0.010 \pm 0.001}$ | $\mathbf{0.114 \pm 0.003}$ | $\mathbf{0.052 \pm 0.002}$ |
| Adult | $\mathbf{H}_{\text{Softmax}}$ | $\mathbf{85.351 \pm 0.157}$ | $90.791 \pm 0.259$ | $78.118 \pm 0.569$ | $0.016 \pm 0.003$ | $0.048 \pm 0.006$ | $0.409 \pm 0.011$ | $0.021 \pm 0.004$ | $0.205 \pm 0.005$ | $0.243 \pm 0.006$ |
| | $\mathbf{H}_{\text{Gumbel}}$ | $85.251 \pm 0.254$ | $\mathbf{90.818 \pm 0.284}$ | $\mathbf{78.399 \pm 0.631}$ | $\mathbf{0.012 \pm 0.002}$ | $\mathbf{0.033 \pm 0.003}$ | $\mathbf{0.313 \pm 0.014}$ | $\mathbf{0.018 \pm 0.005}$ | $\mathbf{0.202 \pm 0.003}$ | $\mathbf{0.233 \pm 0.016}$ |

Table 10 shows that learning a sample-wise Gumbel scale consistently improves calibration over $\mathbf{H}_{\text{Softmax}}$ across the five tabular datasets. In particular, $\mathbf{H}_{\text{Gumbel}}$ reduces ECE, GroupX_ECE, Classwise_ECE, and KCE on all five datasets, while maintaining comparable discriminative performance in ACC, AUC, and AP. These results suggest that the benefit is not merely due to a large increase in model capacity, but rather to providing the optimizer with a dedicated degree of freedom for confidence adjustment.

Figure 7 further supports this interpretation by visualizing the learned sample-wise $\log(\beta)$ values. The learned scales do not collapse to the homoscedastic baseline $\log(\beta) = 0$; instead, they exhibit substantial variation both within and across datasets. This indicates that the model actively uses the additional scale parameter to adapt uncertainty at the sample level. Figure 8 provides a complementary view through reliability diagrams. Compared with $\mathbf{H}_{\text{Softmax}}$, $\mathbf{H}_{\text{Gumbel}}$ generally yields smaller confidence-accuracy gaps, especially in high-confidence regions where overconfidence is most common.

Overall, this case study supports the interpretation that the fixed-scale Softmax parameterization imposes an optimization bias rather than a strict representational impossibility. The sample-wise Gumbel scale provides a simple uncertainty-specific

## Distribution of learned sample-wise log(β) across datasets

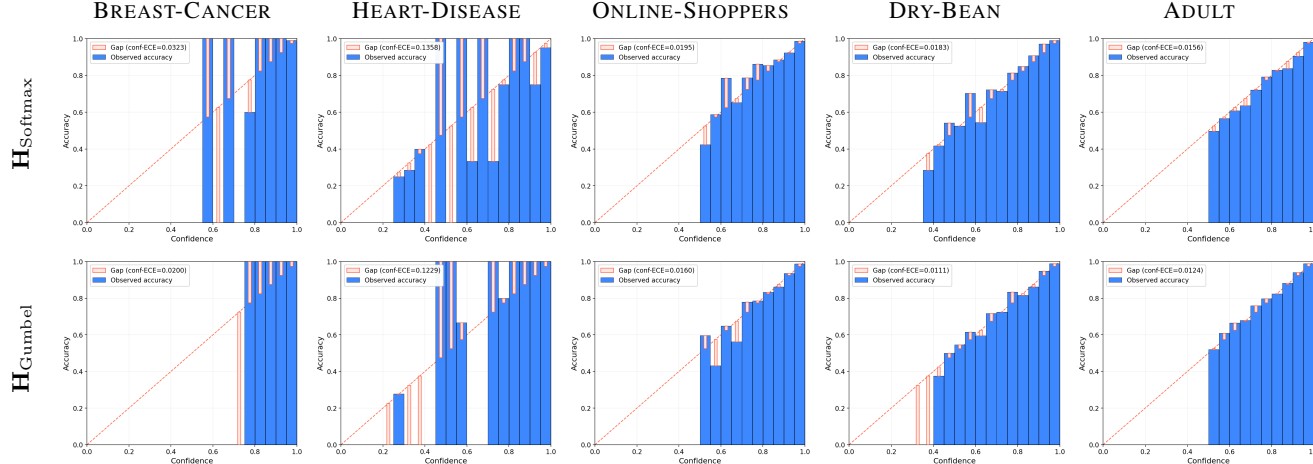

*Figure 7.* Distribution of learned sample-wise $\log(\beta)$ in $\mathbf{H}_{\mathrm{Gumbel}}$ across five tabular datasets. The dashed horizontal line marks the homoscedastic baseline $\log(\beta) = 0$, corresponding to the fixed-scale assumption in $\mathbf{H}_{\mathrm{Softmax}}$.

*Figure 8.* Comparison of reliability diagrams between $\mathbf{H}_{\mathrm{Softmax}}$ and $\mathbf{H}_{\mathrm{Gumbel}}$ on five datasets.

parameter that helps decouple confidence calibration from class discrimination. This finding also motivates the use of more flexible distribution-based classifiers such as $\mathbf{H}_{\mathrm{ALD}}$, where scale and asymmetry parameters provide an even richer inductive bias for modeling heteroscedasticity and skewness.

**Case Study II: Analysis of Parameter Sensitivity.**

In this case study, we analyze the sensitivity of the proposed framework to four key hyperparameters: the ALD threshold $t_0$ in Eq. (2), the MMD regularization weight $\alpha$ in Eq. (11), the individualized calibration weight $\beta$ in Eq. (12), and the RBF kernel bandwidth factor $\tau$ in Eq. (10). The goal is to verify that the reported performance is not driven by a narrow or overly tuned hyperparameter choice. For each parameter, we compare the default setting used in our main experiments against several alternative values on the five tabular datasets. Each entry in Tables 11–14 reports the percentage of datasets for which the default value outperforms the corresponding comparator under a given metric. For predictive metrics, *i.e.*, ACC, AUC, and AP, higher values are better; for calibration and error metrics, *i.e.*, ECE, GroupX_ECE, GroupY_ECE, Classwise_ECE, Brier score, and KCE, lower values are better. Therefore, a value larger than $50\%$ indicates that the default setting wins more often than the comparator across datasets.

**Sensitivity to the ALD threshold $t_0$.** The threshold $t_0$ determines the fixed evaluation point of the ALD CDF in Eq. (2), and therefore serves as an anchor in the latent utility space. As discussed in Section 2.2, fixing $t_0$ avoids the translation non-identifiability that would arise from learning both $t_0$ and the location parameter simultaneously. Table 11 compares the default value $t_0 = 0.5$ against $t_0 \in \{-2.0, -1.0, -0.5, 0.0, 1.0, 2.0\}$. Overall, the default setting wins in more than half of the comparisons against all alternatives, with total win rates ranging from $51.1\%$ to $68.9\%$. The advantage is especially clear against the extreme threshold $t_0 = 2.0$, where the default wins in $68.9\%$ of all metric-dataset comparisons. This suggests that $t_0 = 0.5$ provides a stable anchor that avoids overly saturated CDF evaluations while preserving discriminative and calibration performance.

*Table 11.* Comparison across different $t_0$ values in Eq. (2), using $t_0 = 0.5$ as the default. Each column reports the percentage of datasets for which the default setting outperforms a comparator $t_0 \in \{-2.0, -1.0, -0.5, 0.0, 1.0, 2.0\}$ on a given metric. Higher percentages indicate that the default setting performs **better** more consistently across five datasets.

| Metric | 0.5 vs -2.0 | 0.5 vs -1.0 | 0.5 vs -0.5 | 0.5 vs 0.0 | 0.5 vs 1.0 | 0.5 vs 2.0 |
|---|---|---|---|---|---|---|
| ACC | 60.0% | 20.0% | 80.0% | 60.0% | 60.0% | 40.0% |
| AUC | 80.0% | 60.0% | 80.0% | 100.0% | 60.0% | 80.0% |
| AP | 80.0% | 80.0% | 60.0% | 60.0% | 60.0% | 80.0% |
| ECE | 20.0% | 60.0% | 60.0% | 40.0% | 60.0% | 80.0% |
| GroupX_ECE | 40.0% | 40.0% | 40.0% | 40.0% | 40.0% | 60.0% |
| GroupY_ECE | 40.0% | 40.0% | 40.0% | 20.0% | 40.0% | 80.0% |
| Classwise_ECE | 60.0% | 40.0% | 40.0% | 60.0% | 80.0% | 40.0% |
| Brier_score | 60.0% | 80.0% | 40.0% | 40.0% | 60.0% | 80.0% |
| KCE | 60.0% | 60.0% | 40.0% | 40.0% | 60.0% | 80.0% |
| **Total** | **55.6%** | **53.3%** | **53.3%** | **51.1%** | **57.8%** | **68.9%** |

**Sensitivity to the MMD regularization weight $\alpha$.** The coefficient $\alpha$ controls the strength of the distribution-matching calibration objective in Eq. (11). A larger $\alpha$ places more emphasis on kernel calibration, but may also introduce stronger optimization tension with the discriminative NLL objective. Table 12 shows that the default value $\alpha = 0.1$ is a robust choice: it outperforms larger alternatives in $57.8\%$–$68.9\%$ of all comparisons. In particular, $\alpha = 0.1$ performs consistently well on ECE, GroupY_ECE, Classwise_ECE, and Brier score, indicating that a relatively mild MMD penalty is sufficient to improve calibration without overly distorting the classifier. Although larger $\alpha$ values can sometimes improve KCE directly, they are less favorable when considering the full set of predictive and calibration metrics. This supports the use of $\alpha = 0.1$ as a balanced default.

**Sensitivity to the individualized calibration weight $\beta$.** The coefficient $\beta$ controls the strength of the anchor-confidence alignment term in Eq. (12). Similar to $\alpha$, increasing $\beta$ can encourage stronger calibration alignment, but an overly large value may compete with the NLL objective and affect discriminative learning. Table 13 shows that the default value $\beta = 0.1$ consistently outperforms larger alternatives, with total win rates between $62.2\%$ and $64.4\%$. The default setting is

*Table 12.* Comparison across different $\alpha$ values in Eq. (11), using $\alpha = 0.1$ as the default. Each column reports the percentage of datasets for which the default setting outperforms a comparator $\alpha \in \{0.3, 0.5, 0.7, 0.9\}$ on a given metric. Higher percentages indicate that the default setting performs **better** more consistently across five datasets.

| Metric | 0.1 vs 0.3 | 0.1 vs 0.5 | 0.1 vs 0.7 | 0.1 vs 0.9 |
|---|---|---|---|---|
| ACC | 40.0% | 40.0% | 20.0% | 20.0% |
| AUC | 100.0% | 80.0% | 60.0% | 60.0% |
| AP | 60.0% | 60.0% | 40.0% | 60.0% |
| ECE | 80.0% | 80.0% | 80.0% | 100.0% |
| GroupX_ECE | 40.0% | 80.0% | 80.0% | 80.0% |
| GroupY_ECE | 80.0% | 80.0% | 80.0% | 80.0% |
| Classwise_ECE | 100.0% | 80.0% | 80.0% | 80.0% |
| Brier_score | 100.0% | 80.0% | 60.0% | 60.0% |
| KCE | 20.0% | 20.0% | 20.0% | 20.0% |
| **Total** | 68.9% | 66.7% | 57.8% | 62.2% |

particularly strong on AUC, ECE, Classwise_ECE, and Brier score, while also preserving competitive ACC and AP. These results suggest that a small calibration weight is sufficient to guide the model toward reliable confidence estimates, whereas larger values may overemphasize confidence alignment at the expense of a more balanced predictive-calibration trade-off.

*Table 13.* Comparison across different $\beta$ values in Eq. (12), using $\beta = 0.1$ as the default. Each column reports the percentage of datasets for which the default setting outperforms a comparator $\beta \in \{0.3, 0.5, 0.7, 0.9\}$ on a given metric. Higher percentages indicate that the default setting performs **better** more consistently across five datasets.

| Metric | 0.1 vs 0.3 | 0.1 vs 0.5 | 0.1 vs 0.7 | 0.1 vs 0.9 |
|---|---|---|---|---|
| ACC | 80.0% | 60.0% | 60.0% | 40.0% |
| AUC | 100.0% | 80.0% | 80.0% | 60.0% |
| AP | 60.0% | 60.0% | 60.0% | 40.0% |
| ECE | 80.0% | 80.0% | 80.0% | 100.0% |
| GroupX_ECE | 20.0% | 40.0% | 60.0% | 80.0% |
| GroupY_ECE | 20.0% | 40.0% | 60.0% | 80.0% |
| Classwise_ECE | 100.0% | 80.0% | 80.0% | 80.0% |
| Brier_score | 100.0% | 100.0% | 60.0% | 60.0% |
| KCE | 20.0% | 20.0% | 20.0% | 20.0% |
| **Total** | 64.4% | 62.2% | 62.2% | 62.2% |

**Sensitivity to the RBF kernel bandwidth factor $\tau$.** Finally, we evaluate the sensitivity of the kernel calibration objective to the RBF bandwidth factor $\tau$ in Eq. (10). Kernel-based objectives can be sensitive to bandwidth selection: too small a bandwidth may make the kernel overly local and noisy, whereas too large a bandwidth may oversmooth the distributional discrepancy. Following standard practice, we use the median heuristic (Gretton et al., 2012) as the default bandwidth. Table 14 shows that the median heuristic is consistently competitive against fixed bandwidth factors $\tau \in \{0.25, 0.5, 1.0, 2.0\}$, with total win rates from 62.2% to 66.7%. This indicates that the median heuristic provides a robust data-adaptive bandwidth choice and avoids the need for expensive bandwidth tuning.

**Summary.** Across all four sensitivity analyses, the default hyperparameter choices are consistently competitive and usually outperform alternative settings in the majority of metric-dataset comparisons. The results also reveal a common pattern: stronger calibration weights or more aggressive kernel settings may improve certain calibration-specific quantities, but they do not always provide the best overall trade-off across predictive accuracy, average calibration, group-level calibration, and distribution-level calibration. Therefore, our default configuration is not selected to optimize a single metric in isolation; rather, it provides a stable and balanced choice across heterogeneous datasets and complementary evaluation criteria.

**Case Study III: Sensitivity to Calibration Gradient Scale in the *DDSO*.**

*Table 14.* Comparison across different bandwidth factor $\tau$ values in Eq. (10), using the median heuristic (Gretton et al., 2012) as the default. Each column reports the percentage of datasets for which the default setting outperforms a comparator bandwidth factor $\tau \in \{0.25, 0.5, 1.0, 2.0\}$ on a given metric. Higher percentages indicate that the default setting performs **better** more consistently across five datasets.

| Metric | $\tau_{\mathbf{median}}$ vs $\mathbf{0.25}$ | $\tau_{\mathbf{median}}$ vs $\mathbf{0.5}$ | $\tau_{\mathbf{median}}$ vs $\mathbf{1.0}$ | $\tau_{\mathbf{median}}$ vs $\mathbf{2.0}$ |
|---|---|---|---|---|
| ACC | 40.0% | 60.0% | 60.0% | 80.0% |
| AUC | 80.0% | 80.0% | 60.0% | 80.0% |
| AP | 80.0% | 40.0% | 60.0% | 80.0% |
| ECE | 40.0% | 80.0% | 80.0% | 80.0% |
| GroupX_ECE | 60.0% | 40.0% | 40.0% | 40.0% |
| GroupY_ECE | 60.0% | 100.0% | 80.0% | 40.0% |
| Classwise_ECE | 60.0% | 40.0% | 60.0% | 60.0% |
| Brier_score | 80.0% | 60.0% | 80.0% | 80.0% |
| KCE | 80.0% | 60.0% | 80.0% | 40.0% |
| **Total** | 64.4% | 62.2% | 66.7% | 64.4% |

We further analyze the effect of the calibration gradient scale $s$ in the *DDSO* strategy. This parameter controls how much of the calibration-stream gradient is allowed to flow back into the feature extractor. Concretely, if $h(x)$ denotes the backbone feature, we implement the scaled-gradient feature as

$$\tilde{h}_s(x) = sh(x) + (1 - s)\,\mathrm{sg}(h(x)), \tag{92}$$

where $\mathrm{sg}(\cdot)$ denotes the stop-gradient operator. This construction preserves the same forward feature value, while scaling the calibration gradient passed to the backbone by $s$. Therefore, $s = 0$ corresponds to full detachment, *i.e.*, $\mathbf{H}_{\mathrm{ICALD}}^{\mathrm{Detached}}$, whereas $s = 1$ corresponds to the full-gradient variant. Intermediate values partially couple the calibration objective with backbone representation learning.

Table 15 compares the default detached setting $s = 0$ against $s \in \{0.05, 0.1, 0.2, 0.5, 1.0\}$ across five tabular datasets. Each entry reports the percentage of datasets for which the detached setting outperforms the corresponding comparator under a given metric. For predictive metrics, *i.e.*, ACC, AUC, and AP, higher values are better; for calibration and error metrics, *i.e.*, ECE, GroupX_ECE, GroupY_ECE, Classwise_ECE, Brier score, and KCE, lower values are better.

Overall, $s = 0$ provides the most stable trade-off across metrics. The detached setting wins in $57.8\%$–$73.3\%$ of all metric-dataset comparisons, with the strongest advantage against the full-gradient setting $s = 1.0$. This supports the central motivation of *DDSO*: directly propagating calibration gradients into the backbone can interfere with discriminative representation learning and lead to less reliable confidence estimates. In contrast, full detachment isolates calibration updates to the lightweight calibration pathway, thereby preserving the discriminative backbone while still enabling effective confidence alignment.

The improvement is particularly clear on calibration-sensitive metrics. Compared with all nonzero gradient scales, $s = 0$ wins on ECE for $80.0\%$ of datasets, and it also performs strongly on GroupX_ECE and Classwise_ECE. Against the full-gradient setting $s = 1.0$, the detached model wins on AP, GroupX_ECE, and Classwise_ECE for $100.0\%$ of datasets, and on ECE and Brier score for $80.0\%$ of datasets. Although the detached model does not uniformly dominate ACC, its predictive performance remains competitive, while its calibration behavior is more consistent. These results indicate that the gains from *DDSO* are not merely due to adding a calibration loss, but are closely tied to how calibration gradients are routed during optimization.

**Case Study IV: Robustness under Corruption-based Covariate Shift.**

Beyond in-distribution calibration, a reliable classifier should maintain both predictive accuracy and calibrated confidence under covariate shift. To evaluate this property, we construct corruption-based covariate shifts on the five tabular datasets. Specifically, we consider three types of perturbations: Gaussian noise with severity levels $0.10/0.25/0.50$, feature masking with severity levels $0.10/0.20/0.40$, and mean shift with severity levels $0.10/0.25/0.50$. These corruptions perturb the input distribution while keeping the prediction task unchanged, allowing us to assess whether the model remains reliable

*Table 15.* Comparison across different calibration gradient scales, using $s = 0$ (*i.e.*, $\mathbf{H}_{\text{ICALD}}^{\text{Detached}}$) as the default. Each column reports the percentage of datasets for which the default setting outperforms a comparator $s \in \{0.05, 0.1, 0.2, 0.5, 1.0\}$ on a given metric. Higher percentages indicate that the default setting performs **better** more consistently across five datasets.

| Metric | 0.0 vs 0.05 | 0.0 vs 0.1 | 0.0 vs 0.2 | 0.0 vs 0.5 | 0.0 vs 1.0 |
|---|---|---|---|---|---|
| ACC | 40.0% | 20.0% | 40.0% | 40.0% | 40.0% |
| AUC | 60.0% | 60.0% | 60.0% | 80.0% | 60.0% |
| AP | 80.0% | 60.0% | 60.0% | 80.0% | 100.0% |
| ECE | 80.0% | 80.0% | 80.0% | 80.0% | 80.0% |
| GroupX_ECE | 80.0% | 100.0% | 80.0% | 80.0% | 100.0% |
| GroupY_ECE | 20.0% | 40.0% | 40.0% | 20.0% | 60.0% |
| Classwise_ECE | 60.0% | 60.0% | 60.0% | 60.0% | 100.0% |
| Brier_score | 40.0% | 60.0% | 60.0% | 80.0% | 80.0% |
| KCE | 60.0% | 40.0% | 60.0% | 40.0% | 40.0% |
| **Total** | 57.8% | 57.8% | 60.0% | 62.2% | 73.3% |

when test inputs deviate from the clean training distribution.

Table 16 reports the average performance over all datasets and corruption settings. We include both the corrupted-test performance and the degradation relative to the clean setting. Specifically, $\Delta$ACC measures the change in accuracy under corruption, where a less negative value indicates better robustness. For calibration and error metrics, $\Delta$ECE, $\Delta$Brier, and $\Delta$KCE measure the increase in error under corruption, where smaller values indicate less degradation.

The results show that $\mathbf{H}_{\text{ICALD}}$ is consistently more robust than both $\mathbf{H}_{\text{Softmax}}$ and $\mathbf{H}_{\text{ALD}}$. Under corruption, $\mathbf{H}_{\text{ICALD}}$ achieves the best ACC, ECE, Brier score, and KCE. More importantly, it also exhibits the smallest degradation across all four $\Delta$ metrics. Compared with $\mathbf{H}_{\text{Softmax}}$, $\mathbf{H}_{\text{ICALD}}$ improves the average corrupted accuracy from $81.61$ to $82.36$, reduces ECE from $0.0629$ to $0.0517$, reduces Brier score from $0.2472$ to $0.2301$, and reduces KCE from $0.2114$ to $0.1945$. The degradation is also smaller: $\Delta$ACC improves from $-2.80$ to $-2.50$, while $\Delta$Brier and $\Delta$KCE are reduced from $0.0479$ to $0.0381$ and from $0.0231$ to $0.0159$, respectively.

These results suggest that the proposed individualized calibration mechanism does not merely improve calibration on clean in-distribution data. By explicitly modeling sample-dependent uncertainty, $\mathbf{H}_{\text{ICALD}}$ also provides a more stable confidence estimate under corrupted inputs. This supports the practical value of distribution-based calibration in settings where test-time covariates may be noisy, partially missing, or shifted from the training distribution.

*Table 16.* Robustness under corruption-based covariate shift on five tabular datasets. Results are averaged over all corruption settings, including Gaussian noise $(0.10/0.25/0.50)$, feature masking $(0.10/0.20/0.40)$, and mean shift $(0.10/0.25/0.50)$. Best values are highlighted by **bold yellow**.

| Model | ACC ↑ | ECE ↓ | Brier ↓ | KCE ↓ | $\Delta$ACC ↑ | $\Delta$ECE ↓ | $\Delta$Brier ↓ | $\Delta$KCE ↓ |
|---|---|---|---|---|---|---|---|---|
| $\mathbf{H}_{\text{Softmax}}$ | 81.61 | 0.0629 | 0.2472 | 0.2114 | -2.80 | 0.0226 | 0.0479 | 0.0231 |
| $\mathbf{H}_{\text{ALD}}$ | 81.99 | 0.0616 | 0.2403 | 0.2095 | -2.70 | 0.0217 | 0.0452 | 0.0192 |
| $\mathbf{H}_{\text{ICALD}}$ | **82.36** | **0.0517** | **0.2301** | **0.1945** | **-2.50** | **0.0210** | **0.0381** | **0.0159** |

