# OpenReview forum: "From Individual Calibration to Reliable Classifiers: ALD Parameterization with mPAIC Guarantees"
_ICML.cc/2026/Conference — ICML 2026 regular_

### Official Review · Reviewer_Q4na · 2026-03-11

**Soundness:** 3
**Presentation:** 2
**Significance:** 3
**Originality:** 3
**Overall Recommendation:** 4
**Confidence:** 3

**Summary:**

This paper studies the problem of probability calibration in neural classifiers, focusing on improving the reliability of predicted class probabilities while maintaining strong predictive accuracy. The authors argue that a key limitation of standard Softmax classifiers arises from their probabilistic interpretation under random utility theory: Softmax implicitly corresponds to a Gumbel distribution with a fixed scale parameter, which restricts the model’s ability to represent heteroscedastic uncertainty across samples.


To address this limitation, the paper proposes a distribution based classification framework that generalizes Softmax by modeling class utilities using flexible parametric distributions. In particular, the authors introduce the HALD classifier, which parameterizes class scores using the asymmetric Laplace distribution (ALD). This allows the model to capture both heteroscedastic uncertainty and asymmetric class distributions.



The paper further introduces HICALD, an individualized calibration variant that enforces alignment between predicted confidence and a randomly sampled calibration level. The training objective encourages monotone Probably Approximately Individually Calibrated (mPAIC) predictors, which theoretically implies Probably Approximately Calibrated Classifiers (PACC) under kernel calibration error metrics.


To address optimization challenges caused by conflicting objectives between discrimination and calibration, the authors propose a Decoupled Dual Stream Optimization (DDSO) strategy. DDSO separates discriminative learning and calibration learning through gradient detachment, allowing the calibration head to improve reliability without degrading the learned feature representations.


Experiments on nine datasets across tabular, image, and text domains compare the proposed methods against standard classifiers and several calibration baselines such as temperature scaling, histogram binning, and MMD based calibration. The results show that the proposed ALD based models achieve improved calibration metrics while maintaining competitive predictive performance.

**Compliance With Llm Reviewing Policy:**

Affirmed.

**Final Justification:**

The authors have addressed my concerns. I keep my score as is.

**Key Questions For Authors:**

Why specifically choose the asymmetric Laplace distribution?
The empirical comparison shows improvements, but it would be helpful to understand whether ALD provides theoretical advantages compared to other flexible distributions such as mixtures or learned variance models.

How sensitive is the method to hyperparameters such as the calibration weight and kernel bandwidth?
Calibration methods are often sensitive to such parameters, and it would be useful to understand the robustness of the proposed approach.

How does the method scale to large scale deep learning models?
The experiments mainly involve moderate sized datasets. Could the authors comment on the computational overhead when applied to large scale vision or language models?

How does the method behave under distribution shift or out of distribution evaluation?
Since calibration is especially important under shift, additional experiments would strengthen the claims.

Can the framework be extended to structured prediction tasks or regression problems?
The distribution based formulation suggests that such extensions might be possible.

**Limitations:**

Yes.
The paper explicitly discusses limitations including the focus on top label calibration and the additional computational overhead introduced by calibration training.

**Strengths And Weaknesses:**

Strengths:
The paper provides a clear conceptual motivation by linking probabilistic modeling with calibration. By interpreting Softmax through random utility theory as a distribution based classifier with fixed variance, the authors offer an intuitive explanation for systematic miscalibration and motivate the use of richer parameterizations such as the asymmetric Laplace distribution. The proposed framework treats classification as estimating parameters of a latent distribution whose cumulative distribution function defines class scores, making it flexible and potentially applicable to other parametric families. The work also connects individualized calibration guarantees (PAIC and mPAIC) with kernel based calibration metrics, offering a theoretical bridge between different calibration notions in the literature. From a practical perspective, the DDSO training strategy is well motivated and supported by experiments showing that detaching calibration gradients can preserve feature separability. Finally, the paper includes reasonably broad empirical evaluation across multiple datasets, metrics, and ablations.

Weaknesses:
The overall novelty appears somewhat limited, as several components build directly on existing ideas such as MMD based calibration and individualized calibration from prior work, with the main contribution being their combination with ALD parameterization. The justification for choosing the asymmetric Laplace distribution is largely empirical, and it is unclear whether it provides fundamental advantages over other flexible distributions. The paper also offers limited discussion of scalability to very large models or datasets. In addition, the evaluation relies heavily on ECE and kernel calibration metrics, and further analysis such as reliability diagrams or out of distribution testing could strengthen the claims. Finally, the presentation introduces multiple related concepts (PACC, PAIC, mPAIC, and kernel calibration) which makes the overall narrative somewhat dense.

---

> ### Author Rebuttal · Authors · 2026-03-30
>
> We sincerely thank you for your constructive feedback and reading of our submission.
>
> **R1 [W]:** We added 8 additional baselines and 3 additional evaluation metrics. The overall results are reported in Table 1 [https://anonymous.4open.science/r/ICML-Re/Exp1.png ], and the corresponding reliability-diagram comparisons are shown in Fig.1 [https://anonymous.4open.science/r/ICML-Re/Exp1.png ]. These additions substantially strengthen the empirical evaluation.
>
> **R2 [Q1]:** Beyond its strong empirical performance, we choose ALD because it provides a simple yet expressive parameterization with three degrees of freedom: location, scale, and asymmetry. Compared with $\mathbf{H}_{\rm Softmax}$, which under the random-utility interpretation corresponds to a fixed-scale Gumbel model, ALD offers a more flexible latent-utility model while remaining lightweight and analytically convenient through its closed-form CDF.
>
> To further support this motivation, we added a case study to isolate the effect of the fixed-scale assumption. Specifically, compared with $\mathbf{H}\_{\rm Softmax}$, the classifier $\mathbf{H}\_{\rm Gumbel}$ keeps the same classification setting but relaxes only the fixed-scale assumption in Eq. (1): $\mathbf{H}\_{\rm Softmax}$ fixes $\beta=1$, whereas $\mathbf{H}\_{\rm Gumbel}$ learns a sample-dependent $\beta(x)$. As shown in Table 2 and Figs.2~3 [https://anonymous.4open.science/r/ICML-Re/Exp2.png ], $\mathbf{H}\_{\rm Gumbel}$ consistently improves calibration metrics over $\mathbf{H}\_{\rm Softmax}$, reduces confidence-ECE in reliability diagrams across all five datasets, and exhibits substantial sample-wise variation in the learned $\log(\beta)$ away from the homoscedastic baseline $\log(\beta)=0$. Taken together, these results suggest that moving beyond the fixed-scale assumption and adopting a more flexible distributional choice such as ALD is beneficial for calibration.
>
> **R3 [Q2]:** We added case studies on the calibration weights and kernel bandwidth. Specifically, we varied the weights in Eq. (11) and Eq. (12), as well as the bandwidth factor relative to the median heuristic. As shown in Tables 7~9 [https://anonymous.4open.science/r/ICML-Re/Exp7.png ], the default setting remains reasonably robust and consistently provides a good trade-off across datasets and metrics. These results suggest that the method is not overly sensitive to these hyperparameters.
>
> **R4 [Q3]:** We further added corruption-based distribution-shift experiments on five tabular datasets, including Gaussian noise, feature masking, and mean shift. As shown in Table 10 [https://anonymous.4open.science/r/ICML-Re/Exp8.png ], ICALD remains the most robust model under shift, achieving the strongest post-shift calibration performance and the smallest degradation relative to the in-distribution baseline.
>
> **R5 [Q4]:** While our experiments are conducted on moderate-scale datasets, the computational pattern of our method suggests favorable scalability to larger vision or language models. In particular, ICALD in the pre-calibration setting can be accelerated through the DDSO strategy. Alternatively, ICALD can also be used in a post-calibration manner using Eq. (13), where it is attached to an already-trained model as a lightweight calibration module, making the framework practical for large-scale settings.
>
> **R6 [Q5]:** Our method can be naturally extended to regression, while extension to structured prediction also seems possible, but would require additional careful design choices and is therefore left for future work.
> For regression, our framework can be extended from top-label confidence calibration to distribution-level calibration (see also o **R1 for Reviewer tVsS**). Specifically, given an input $x$ and a random variable $q \sim U(0,1)$, the model outputs a predictive distribution for $Y$ (e.g., through the three ALD parameters \{$\kappa,\sigma,\theta$\} = $M_\Theta(x,q)$). One can then directly penalize
> $$
> \frac{1}{N}\sum_{i=1}^N | F_{\rm ALD} (y_i; M_\Theta(x_i,q_i))- q_i |.
> $$
> This yields a calibrated predictive distribution over $Y$, which can be interpreted as the mixture
> $$
> \int dq \ p(q) f_{\rm ALD} (y; M_\Theta(x,q)), p(q)=U(0,1),
> $$
> where the feature extractor captures the overall shape of the predictive distribution, while the $q$-branch acts as a lightweight adapter for local calibration adjustment. Here, $f_{\rm ALD}$ and $F_{\rm ALD}$ denote the PDF and CDF of the ALD, respectively. In this setting, the corresponding theoretical implication becomes $\mathrm{mPAIC} \Rightarrow \mathrm{PAIC}$, and point predictions can then be obtained from summaries of the calibrated predictive distribution, such as the mean, median, mode, or quantiles.
>
> For structured prediction, we believe a similar idea may be applicable by defining suitable structured confidence or distributional calibration targets, but this would require a task-specific formulation and is an interesting direction for future work.

---

> > ### Author Rebuttal · Reviewer_Q4na · 2026-04-04
> >
> > Thanks for the response. I am satisfied with the response and keep my score as is.

---

> > > ### Author Response · Authors · 2026-04-04
> > >
> > > We sincerely thank you for your careful reading of our paper and for your thoughtful follow-up comments. We greatly appreciate that you found our response satisfactory and are grateful for your time and consideration throughout the review process. Thank you again for your constructive feedback and support.

---

### Official Review · Reviewer_yAuh · 2026-03-12

**Soundness:** 3
**Presentation:** 3
**Significance:** 3
**Originality:** 3
**Overall Recommendation:** 5
**Confidence:** 3

**Summary:**

This paper proposes replacing the standard Softmax classifier with one based on the Asymmetric Laplace Distribution (ALD), which predicts three parameters per class (location, scale, asymmetry) rather than a single logit. The authors argue that Softmax is equivalent to a Gumbel distribution with a fixed scale parameter, enforcing homoscedasticity and preventing the model from capturing sample-specific uncertainty. The ALD's additional parameters allow the model to adapt both the spread and skew of class distributions per input, which the authors show improves both accuracy and calibration across seven distribution families tested.

On top of the ALD classifier, the authors introduce ICALD, an individualized calibration framework. During training, a random anchor q ~ U(0,1) is fed alongside the input, and the ALD parameters become functions of (x, q). The model is penalized for deviation between its top-label confidence and q, forcing it to produce calibrated predictions at every confidence level. They prove a chain of implications: mPAIC => PAIC => PACC, connecting the confidence-alignment objective to kernel calibration error bounds. To address the tension between discriminative and calibration objectives, they propose Decoupled Dual-Stream Optimization (DDSO), which runs two parallel streams per iteration: a discriminative stream with q = 1 that updates the full backbone via cross-entropy, and a calibration stream with random q that only updates a lightweight calibration head via gradient detachment. This prevents calibration gradients from degrading learned feature representations. The framework supports both pre-calibration (end-to-end) and post-calibration (lightweight adapter) deployment. Experiments across 9 datasets (tabular, image, text) with 7 metrics and 5 random splits show consistent improvements over Softmax, Temperature Scaling, Histogram Binning, and MMD-based baselines.

**Compliance With Llm Reviewing Policy:**

Affirmed.

**Final Justification:**

See rebuttal answer

**Key Questions For Authors:**

1. **Missing baselines**: How does ICALD compare to Dirichlet calibration (Kull et al., NeurIPS 2019) and evidential deep learning (Sensoy et al., NeurIPS 2018)? Both are distribution-based approaches that address the same Softmax limitations. Dirichlet calibration in particular is a strong post-hoc baseline that generalizes Temperature Scaling, and evidential DL similarly predicts per-class uncertainty parameters. Including these comparisons (even on a subset of datasets) would substantially strengthen the experimental contribution and could raise my overall recommendation.

2. **DDSO trade-off**: By blocking calibration gradients from the backbone, DDSO prevents the calibration objective from improving feature representations. Have you measured cases where allowing some calibration gradient flow (e.g., with a small scaling factor rather than full detachment) could improve both objectives? The binary choice between full gradient and full detachment may not be optimal.

3. **Softmax expressiveness vs. optimization dynamics**: The paper argues that Softmax "lacks the necessary degrees of freedom" to capture heteroscedasticity due to its fixed Gumbel scale parameter. However, Softmax can represent arbitrary categorical distributions - a sufficiently powerful backbone can produce any logit configuration, including small gaps for uncertain inputs and large gaps for confident ones. The fixed scale in the latent utility interpretation does not constrain the output probability distribution. A more defensible argument would be that the ALD's separate scale parameter provides an architectural inductive bias that decouples confidence from discrimination, making it easier for the optimizer to avoid the overconfidence driven by cross-entropy training (Guo et al. 2017). Can the authors clarify whether their claim is about fundamental representational limitations of Softmax or about optimization dynamics? If the latter, can they provide evidence (e.g., analysis of learned scale parameters, or comparison of optimization trajectories) showing that the ALD's extra parameters specifically help the optimizer find better-calibrated solutions rather than merely increasing model capacity?

4. **Inference-time anchor protocol**: The training procedure clearly specifies that the discriminative stream uses q = 1 and the calibration stream samples q ~ U(0,1), with K anchors per mini-batch. However, the paper does not state what value of q is used at test time. This is a significant omission because the choice has major implications. Setting q = 1 asks the model for maximum confidence, which would seemingly encourage overconfident predictions - the opposite of the calibration goal. Averaging over multiple q values would be expensive and seems inconsistent with the reported inference cost of 0.554x relative to full-gradient ICALD. Selecting a specific q on validation introduces another hyperparameter. Can the authors clarify the exact inference protocol and explain why the chosen value of q produces well-calibrated predictions at test time?

5. **t0 sensitivity**: The threshold t0 shows non-trivial sensitivity across distributions (e.g., ALD ranks range from 3.11 to 4.56 in Table 4). Is the optimal t0 consistent across datasets, or should it be cross-validated per dataset? How sensitive are the main results to this choice?

**Limitations:**

The authors discuss two main limitations: the focus on top-label calibration (canonical calibration remains open) and computational overhead. Both are honest and appropriate. However, the paper could also acknowledge: (1) the monotonicity gap between theory and practice discussed above; and (2) the dependence on the t0 hyperparameter, which adds a tuning dimension absent from standard Softmax training. The societal impact discussion is adequate but brief.

**Strengths And Weaknesses:**

**Strengths**

*Soundness.* The proofs of the mPAIC => PAIC => PACC chain (Theorems 2.4 and 2.5) are correct and follow standard techniques (Kantorovich-Rubinstein duality, U-statistic concentration), though the key monotonicity assumption is not enforced during training (discussed further in weaknesses). On the empirical side, the evaluation protocol is robust: 5 random splits with Nemenyi tests and CD diagrams exceeds the standard in calibration papers. The comparison across 7 distribution families (Table 4, Table 5) is thorough and the DDSO ablation (Figure 3, Table 7) convincingly demonstrates the feature collapse problem and its solution via gradient detachment.

*Presentation.* The paper covers a lot of ground - distributional classifiers, the calibration framework, DDSO, and experiments across 9 datasets - while remaining mostly clear. The Gumbel-Softmax connection in Section 2.1 is well-explained, the DDSO schematic (Figure 1) is helpful, and the CD diagrams provide an effective visual summary of results. The appendix is comprehensive, with full proofs, detailed dataset descriptions, and extensive ablations.

*Significance.* The paper addresses a genuinely important open problem. Zhao et al. (2020) explicitly listed extending the PAIC framework to classification as future work, noting the lack of a natural CDF for discrete random variables. The authors' solution - using top-label confidence as the calibration quantity - is clean and opens a new connection between individual calibration and distribution-matching calibration (KCE/MMD). The ALD classifier's negligible overhead (1.020x, Table 6) and the framework's dual pre/post-calibration flexibility make it practically deployable. The ALD parameterization also provides a useful architectural inductive bias: by giving the optimizer a separate per-sample scale parameter for controlling confidence, it decouples calibration from discrimination in a way that cross-entropy training over Softmax logits does not naturally afford.

*Originality.* The ALD as a classifier parameterization replacing Softmax has no direct precedent. The mPAIC => PACC bridge is a genuine theoretical contribution connecting two previously separate calibration paradigms. The DDSO gradient detachment strategy is a novel application of stop-gradient techniques to the calibration-discrimination tension, distinct from prior multi-task gradient methods (PCGrad, GradNorm) which modify gradients rather than blocking them entirely. The systematic evaluation of 7 distribution families as Softmax alternatives is itself a useful empirical contribution.

**Weaknesses**

*Soundness.*
- The mPAIC => PAIC proof (Theorem 2.4) requires monotonicity of q -> p_hat_y*(x,q), but the authors acknowledge in Appendix A.2 that monotonicity is not enforced during training and is not relied upon when reporting results. The experiments thus optimize an upper bound surrogate without verifying the condition under which the bound is tight.
- The paper's motivating claim that Softmax "lacks the necessary degrees of freedom" to capture heteroscedasticity could be more carefully stated. Since Softmax can represent arbitrary categorical distributions via logit magnitudes, the fixed scale in the Gumbel interpretation does not obviously constrain the output probabilities. It is possible I am misunderstanding the authors' intended argument, but as written, the distinction between a limitation of the latent utility model and a limitation of the output distribution is not clearly drawn. If the intended argument is about optimization dynamics rather than representational capacity, that would be more convincing but would need to be framed and tested explicitly.
- Several common calibration evaluation metrics are not reported, including reliability diagrams, individual calibration metrics (which would directly validate the mPAIC claims), Brier score, and classwise ECE.

*Presentation.*
- The inference-time protocol for the anchor q is never specified. Since the entire ICALD framework is built around conditioning on q, the choice at test time is fundamental yet omitted from both the main text and appendix.
- The transition from Section 2.2 (MMD-based calibration) to Section 2.3 (ICALD) is abrupt; it is unclear when one approach is preferred over the other.

*Significance.*
- The absence of comparisons to strong distribution-based baselines - Dirichlet calibration (Kull et al., NeurIPS 2019), evidential deep learning (Sensoy et al., NeurIPS 2018), Natural Posterior Networks (Charpentier et al., ICLR 2022) - makes it difficult to assess whether the improvements come from the ALD parameterization specifically or from having more parameters in general.

*Originality.*
- The paper's positioning against the broader landscape of Softmax alternatives is incomplete. Evidential DL, Dirichlet calibration, Natural Posterior Networks, and bi-tempered logistic loss all replace or augment Softmax with richer distributional parameterizations for uncertainty-aware classification, yet none are cited or discussed. This makes it harder to assess what is specifically new about the ALD approach beyond the particular choice of distribution.

---

> ### Author Rebuttal · Authors · 2026-03-30
>
> We sincerely thank you for your time and effort devoted to evaluating our work in detail. Your constructive comments on the Weaknesses and Questions are highly valuable to us, and will help us further improve the manuscript.
>
> **R1 [W1; Limitations(1)]:** We will make this distinction more explicit in both the main text and the limitations section: Theorem 2.4 characterizes the monotone case and provides the tightness condition, whereas the current experiments optimize a practical surrogate without explicitly enforcing monotonicity.
>
> **R2 [W2; Q3]:** Our claim is *not* that Softmax cannot represent arbitrary categorical distributions. Rather, under the random-utility view, $\mathbf{H}_{\rm Softmax}$ corresponds to a fixed-scale Gumbel model with $\beta=1$ and thus lacks an explicit sample-specific uncertainty parameter. By contrast, $\mathbf{H}\_{\rm ALD}$ introduces separate scale $\sigma(x)$ and asymmetry $\kappa(x)$ parameters, providing a more direct inductive bias for heteroscedasticity and skewness. We will revise the paper to emphasize that our argument concerns latent-utility parameterization and optimization bias, not strict representational impossibility.
>
> To test this directly, we added a case study with $\mathbf{H}\_{\rm Gumbel}$, which keeps the classification setting of $\mathbf{H}\_{\rm Softmax}$ but relaxes only the fixed-scale assumption in Eq. (1) by learning a sample-dependent $\beta(x)$. As shown in Table 2 and Figs. 2~3 [https://anonymous.4open.science/r/ICML-Re/Exp2.png ], $\mathbf{H}\_{\rm Gumbel}$ improves calibration metrics over $\mathbf{H}\_{\rm Softmax}$, reduces confidence-ECE on all five datasets, and shows substantial variation in learned $\log(\beta)$ away from the homoscedastic baseline $\log(\beta)=0$. This supports the view that the gain comes from relaxing the fixed-scale assumption and giving the optimizer a dedicated uncertainty degree of freedom, rather than simply increasing capacity.
>
> **R3 [W3, 6, 7; Q1]:** We expanded the study by adding 8 baselines and 3 evaluation metrics. The updated overall results are shown in Table 1, and the corresponding reliability diagrams in Fig. 1 [https://anonymous.4open.science/r/ICML-Re/Exp1.png ]. Overall, our methods remain competitive with other baselines. We will expand the Related Work section to provide a more detailed discussion of these additional baselines, and update the Appendix with the full implementation details of all added baselines and evaluation metrics.
>
> **R4 [W4; Q4]:** We sincerely apologize for omitting this detail. We do not tune a special test-time anchor $q$ on the validation set; the held-out split is used only for early stopping. During training, the discriminative stream uses $q=1$ for the NLL term, while the calibration stream samples $K$ *i.i.d.* anchors $q \sim U(0,1)$ per mini-batch and averages the calibration loss. At inference, we sample 200 anchors, compute the corresponding predictive probabilities, and average them.
>
> When normalizing $\mathbf{H}^\text{Full Grad}\_\text{ICALD}$ to $1\times$, $\mathbf{H}^\text{Detached}\_\text{ICALD}$ requires only $0.554\times$ total train+inference time while still achieving calibration gains. Since inference is only a small fraction of total cost, training dominates efficiency; DDSO therefore yields better practical efficiency.
>
> **R5 [W5]:** We will improve the transition from Section 2.2 to 2.3. Empirically, ICALD achieves stronger overall performance and naturally supports both pre-calibration and post-calibration, making it flexible across deployment settings.
>
> **R6 [Q2]:** We added a case study on intermediate calibration-gradient scales, varying $s \in \\{0.05, 0.1, 0.2, 0.5, 1.0\\}$ and comparing them with the default fully detached variant ($s=0$), i.e., $\mathbf{H}^\text{Detached}_{\text{ICALD}}$. Here, $s$ controls how much gradient from the calibration branch is allowed to flow back into the shared feature extractor. As shown in Table 5 [https://anonymous.4open.science/r/ICML-Re/Exp5.png ], the fully detached setting remains the strongest and most consistent overall. Its advantage becomes more pronounced as $s$ increases. Small nonzero scales can help on some metrics, but these gains are not consistent across datasets.
>
> **R7 [Q5; Limitations(2)]:** In our experiments, $t_0$ is fixed across datasets and is not cross-validated separately. To test sensitivity, we added a case study varying $t_0 \in \\{-2.0,-1.0,-0.5,0.0,0.5,1.0,2.0\\}$ with $t_0=0.5$ as default. As shown in Table 6 [https://anonymous.4open.science/r/ICML-Re/Exp6.png ], the default remains robust overall; its advantage is especially clear over more distant choices such as $t_0=2.0$ (overall win rate $68.9\%$). These results suggest that while $t_0$ affects performance, the main conclusions are not overly sensitive to it. We agree that $t_0$ introduces an extra tuning dimension relative to standard Softmax training, and we will state this explicitly as a limitation.

---

> > ### Author Rebuttal · Reviewer_yAuh · 2026-04-03
> >
> > I thank the authors for a thorough and substantive rebuttal. Several of my key concerns have been addressed:
> >
> > **R2/Q3 (Softmax expressiveness) - Resolved.** The authors agree to reframe the argument around latent-utility parameterization and optimization bias rather than representational impossibility, which is what I was asking for. More importantly, the H_Gumbel(beta) ablation is the right experiment that provides evidence that the fixed-scale assumption is the bottleneck.
> >
> > **R3/Q1 (Missing baselines) - Largely resolved.** The addition of 8 baselines and 3 metrics (including reliability diagrams) is a significant effort.
> >
> > **R4/Q4 (Inference protocol) - Resolved.** The clarification that inference averages 200 anchor samples fills a real gap. This must be included in the paper.
> >
> > **R6/Q2 (DDSO partial gradient) - Resolved.** New experiments strengthens confidence in the DDSO design choice.
> >
> > **R7/Q5 (t0 sensitivity) - Resolved.** The sensitivity analysis shows robustness around the default. Acknowledging t0 as a limitation is appropriate.
> >
> > **R1 (Monotonicity) - Partially resolved.** Agreeing to clarify the theory-practice distinction is helpful but the underlying gap remains. This is a minor residual concern.
> >
> > Overall, the rebuttal is good: authors ran substantive new experiments (H_Gumbel(beta) ablation, gradient scale sweep, t0 sensitivity, 8 new baselines) that fill in the gaps. I am raising my recommendation to 5 (Accept), contingent on the promised revisions being incorporated into the final version - particularly the reframed expressiveness argument, the inference protocol description, and the expanded baselines.

---

> > > ### Author Response · Authors · 2026-04-03
> > >
> > > We sincerely thank you for carefully reconsidering our paper and for taking the time to read our rebuttal in detail. We also greatly appreciate your decision to raise the score after considering our response. Thank you for your thoughtful follow-up comments. We're encouraged that the additional experiments and clarifications addressed most of your concerns.
> > >
> > > In the final version, we will incorporate all promised revisions from the rebuttal, including the revised framing of the Softmax expressiveness discussion with the $\mathbf{H}_{\mathrm{Gumbel}}(\beta)$ case study, the expanded baselines and metrics with reliability diagrams, explicitly describing the inference protocol, the DDSO partial gradient case study, the $t_0$ sensitivity analysis, and the clarified discussion of the monotonicity gap and limitations.
> > >
> > > Thank you again for your constructive feedback and for helping improve the paper.

---

### Official Review · Reviewer_hB4R · 2026-03-13

**Soundness:** 2
**Presentation:** 1
**Significance:** 3
**Originality:** 3
**Overall Recommendation:** 4
**Confidence:** 2

**Summary:**

The paper proposes a new calibration framework for learning distribution-based classifiers. The main method called ALD is based on the asymmetric Laplace distribution but also allows for more flexible output distributions. The paper extends this method from distributional calibration matching to a method called ICALD which introduces an individual calibration guarantee. The paper also proposes a method to encourage predictive performance and calibration at the same time. The paper then evaluates the proposed methods on simple tabular, image, and text classification benchmarks.

**Compliance With Llm Reviewing Policy:**

Affirmed.

**Final Justification:**

I found the rebuttal by the author's convincing and trust that they will make the relevant changes in the paper before publication.

**Key Questions For Authors:**

Most question included above. In addition:

1. Is there a reason why Vector/Dirichlet scaling was not tested in the empirical results section?
2. Do you envision similar ideas as proposed in this work generalizing to regression or sequence-based models?
3. Why are the only tested datasets relatively simple datasets? If the claim of scalability is true, then there should be no concerns on testing it on higher dimensional, more versatile data distributions.

**Limitations:**

The paper does include a limitations paragraph in the conclusion section. I would also argue that the presentation of the work could be significantly improved.

**Strengths And Weaknesses:**

Overall, I found the paper hard to read at times and somewhat inaccessible. I would strongly encourage the authors to add additional intuition.

### __Soundness__:
- In general, the proposed approach seems sound to me.
- In the intro the paper claims that TS acts as a monotone re-scaler that preserves ranking. This is a common but wrong misconception. Temperature scaling can in fact change the ranking of data points. While TS leads to a monotone rescaling of the *logits*, it can lead to a non-monotone rescaling of the *softmax probabilities*. Since the softmax function is non-linear with respect to the temperature parameter (it enters both in the exponential of the numerator but also in the sum of exponentials in the denominator), temperature scaling can therefore change the ranking of samples by confidence. See [1,2] who have studied this effect.
- The usage of a kernel function for yielding a continuous calibration measure is not assumption free. While it might be differentiable, it still uses hyper-parameters for the kernel to determine smoothness. This does not really side-step the subjectivity of the binning resolution in ECE.
- The procedure to make $H_\\text{ALD}$ PACC seems pretty standard to me.
- I had some trouble parsing some steps in section 2.3 (as described below).

### __Presentation__:
- While the paper does talk about some related work, it is somewhat scattered throughout the paper. Therese is no related work section in the main paper which contextualizes all of this work in a single section. There is a forward pointer to the appendix but I would have strongly preferred to have a at least a paragraph or two included in the main paper.
- Little intuition is provided then ALD is initially introduced. The paragraph in lines 93-101 (right column) are where the paper should have provided intuition, not formalism. What do the parameters represent? What is the threshold for? A visual depiction might have been helpful.
- What is the kernel calibration error (line 150)? This term is introduced here but not defined ahead of time. It is only defined in the later paragraph and also not formally introduced there either.
- What is the random anchor in line 192? What does it do and why is it necessary? Again, the paper introduces a term and then goes on to use it later which forces a lot of cognitive load onto the reader. At the time of introduction, it is unclear why this is even needed. Ideally, intuition for the need of a particular variable should be established first.
- Frankly, despite being familiar with the uncertainty and calibration literature, I lack the understanding to understand much of what is happening in Section 2.3. I hope that more theoretically trained reviewers can assess this part of the paper better than me.
- It seems like the paper is trying to do too much at the same time. I wonder whether a focus on one central contribution would have helped to improve the presentation of this work. There are a lot of approaches and ideas introduced and there are frequent references to the appendix that should provide more detail.
- I had a hard time understanding Figure 2. I apologize if this is standard in the community but it took me a while to interpret the findings. The main paper does not introduce what critical distance (CD) measures.
- The paper is using some very noticeable vspace tricks to fit the paper into 8 pages. This is not a great look and further contributes to the previous point.

### __Significance__:
- Building flexible yet well calibrated models is important for trustworthy ML. As such, the paper provides a valuable contribution to this important effort.
- Condition on soundness of the approach, this approach could help to establish other calibration approaches based on learning more flexible output distributions.
- I wonder why only very simple datasets are used for the empirical evaluation.

### __Originality__:
- I think the paper proposes an interesting extension on how we can train more flexible and calibrated classifiers.
- Related work is discussed inline but I would have preferred to see a dedicated related work section in the main body of the paper.

__References__

[1]: Galil, Ido, Mohammed Dabbah, and Ran El-Yaniv. "What can we learn from the selective prediction and uncertainty estimation performance of 523 imagenet classifiers." arXiv preprint arXiv:2302.11874 (2023).

[2]: Rabanser, Stephan, and Nicolas Papernot. "What Does It Take to Build a Performant Selective Classifier?." arXiv preprint arXiv:2510.20242 (2025).

---

> ### Author Rebuttal · Authors · 2026-03-30
>
> Thank you for your careful reading and feedback. Below, we will respond to each point.
>
> **R1 [S2]:** Thanks for pointing this out and for the references. Our original wording was imprecise. Temperature scaling preserves the within-sample ordering of logits, and thus the predicted class and accuracy, but not necessarily the cross-sample ranking by softmax confidence. This clarification does not affect the main narrative, and we will revise the introduction accordingly.
>
> **R2 [S3]:** We agree that KCE is not assumption-free. While KCE avoids the hard discretization used in ECE and provides a differentiable objective, it still depends on kernel design choices, especially the bandwidth parameter controlling smoothness. Our intended point was only that, compared with ECE, KCE offers a continuous and differentiable surrogate that is more convenient for end-to-end optimization (Marx et al., 2023). We also added a sensitivity analysis for this kernel parameter in Table 9 [https://anonymous.4open.science/r/ICML-Re/Exp7.png ]. We will make this distinction explicit and avoid overstating KCE as a bias-free alternative to ECE.
>
> **R3 [P1, 2, 6, 8; O2; Limitations]:** We acknowledge the presentation issues and will address them in the final version. Using the 9-page allowance, we will move the Related Work (Appendix A.4) and the introduction of ALD (Appendix C.1) back into the main paper, clarify our contributions, and move non-essential details to the appendix.
>
> **R4 [P3]:** We will add the reference (Marx et al., 2023) for kernel calibration error.
>
> **R5 [P4]:** Intuitively, $q \sim U(0,1)$ is a sampled target confidence level. It makes calibration queryable and level-conditioned: instead of producing a single fixed confidence per input, the model is trained to align its top-label confidence with different target levels $q$ across $(0,1)$. The anchor is also theoretically important because it connects our construction to the mPAIC formulation and the implication chain ${\rm mPAIC} \Rightarrow {\rm PAIC} \Rightarrow {\rm PACC}$. We will revise the paper to explain this intuition earlier and connect it more explicitly to both the confidence-alignment loss and the mPAIC theory.
>
> **R5 [P7]:** We used the CD diagrams to provide a compact statistical summary across methods and settings. As **Reviewer yAuh** also noted in the Strengths section, the CD diagrams can serve as an effective visual summary.
> Concretely, the critical difference is the significance threshold from the Nemenyi post-hoc test after Friedman ranking, and classifiers connected by a thick horizontal bar should be interpreted as statistically tied. We agree that the current draft assumes too much familiarity with CD diagrams, and we will explain this more clearly in the final version.
>
> **R6 [Q1]:** We expanded the experimental section by adding 8 additional baselines and 3 additional evaluation metrics. The updated overall results are reported in Table 1 [https://anonymous.4open.science/r/ICML-Re/Exp1.png ], and the corresponding reliability-diagram comparisons are shown in Fig.1 [https://anonymous.4open.science/r/ICML-Re/Exp1.png ]. These additions substantially strengthen our empirical evaluation, and we remain open to including more baselines if needed.
>
> **R7 [Q2]:** For regression, our framework can be extended from top-label confidence calibration to distribution-level calibration. Specifically, given an input $x$ and a random variable $q \sim U(0,1)$, the model outputs a predictive distribution for $Y$ (e.g., through the three ALD parameters \{$\kappa,\sigma,\theta$\}=$M_\Theta(x,q)$). One can then directly penalize
> $$
> \frac{1}{N}\sum_{i=1}^N |F_{\rm ALD} (y_i; M_\Theta(x_i,q_i))-q_i|.
> $$
> This yields a calibrated predictive distribution over $Y$, interpretable as the mixture
> $$
> \int dq \ p(q) f_{\rm ALD} (y; M_\Theta(x,q)), p(q)=U(0,1),
> $$
> where the feature extractor captures the overall shape of the predictive distribution, while the $q$-branch acts as a lightweight adapter for local calibration adjustments. In this regression setting, the corresponding theoretical implication becomes $\{\rm mPAIC} \Rightarrow {\rm PAIC}$, and point prediction can be obtained from summaries of the calibrated predictive distribution. For sequence-based models, the calibration mechanism is agnostic to the choice of feature extractor, so RNNs, Transformers, or other autoregressive architectures can be used as the backbone encoder.
>
> **R8 [Q3]:** We did not intentionally choose simple datasets. For the tabular setting, we adopted the same benchmark datasets as Marx et al. (2023) to ensure a fair comparison. We additionally included image and text benchmarks to assess scalability to higher-dimensional inputs, covering 9 datasets in total. We have now added one more high-dimensional dataset, as shown in Table 4 [https://anonymous.4open.science/r/ICML-Re/Exp4.png ], and remain open to considering other datasets in the final version pending suggestions from the reviewer.

---

> > ### Author Rebuttal · Reviewer_hB4R · 2026-04-03
> >
> > I thank the authors for their response to my review. I think my concerns have been reasonably well addressed. As such, I am willing to recommend weak acceptance of the paper. However, since it is hard to verify the proposed changes (in particular the presentation issues) due to paper updates not being allowed, I don't think I will be able to strongly advocate for the paper.

---

> > > ### Author Response · Authors · 2026-04-03
> > >
> > > We sincerely thank you for your careful reconsideration of our paper and for your positive update. We greatly appreciate your willingness to raise the score after considering our rebuttal. We are also especially grateful for your constructive feedback on the presentation of the paper. In the final version, we will carefully revise and polish the manuscript to improve its clarity, intuition, and overall readability, and to better present the key ideas and contributions. Thank you again for your helpful comments and support.

---

### Official Review · Reviewer_tVsS · 2026-03-15

**Soundness:** 3
**Presentation:** 3
**Significance:** 2
**Originality:** 2
**Overall Recommendation:** 5
**Confidence:** 4

**Summary:**

The paper addresses a limitation of softmax-based classifiers, which rely on a fixed scale parameter and are therefore vulnerable to heteroscedasticity. To address this issue, the authors propose a distribution-based classifier based on the Asymmetric Laplace Distribution (ALD).

The training objective jointly considers prediction accuracy and calibration quality. Specifically, the model is trained using the negative log-likelihood (NLL) for classification performance, while Maximum Mean Discrepancy (MMD) is incorporated to encourage calibration. Building on this formulation, the authors further propose an individualized calibration method based on ALD, termed ICALD.

The authors also provide theoretical analysis showing that the proposed framework induces the implication chain mPAIC ⇒ PAIC ⇒ PACC, thereby providing a distributional justification for the confidence-alignment objective.

**Compliance With Llm Reviewing Policy:**

Affirmed.

**Final Justification:**

The authors have addressed all major concerns during the rebuttal process.

**Key Questions For Authors:**

1. It would be helpful if the authors could clarify how the proposed approach differs from the method introduced in “Distribution Calibration for Regression.”

2. Could the authors provide more direct evidence linking homoscedasticity to calibration performance? For example, it would be helpful to quantify this relationship using appropriate metrics.

3. Could the authors include comparisons with the recent methods mentioned in the Weakness section?

* While evaluating pre-hoc methods may require additional training time, post-hoc methods can likely be evaluated relatively quickly. It would also be interesting to investigate whether existing post-hoc calibration methods perform well with distribution-based classifiers, and how their performance differs from that of conventional softmax-based classifiers.

**I will primarily base my decision on the author's response to Question 1, with the response to Question 2 also taken into consideration. The response to Question 3 will mainly help resolve my hesitation between the two scores. Depending on the author's responses, my final score may range from weak reject to accept.**

**Limitations:**

Yes. The authors provide the limitations of their method.

**Strengths And Weaknesses:**

**Strengths**
1. The paper is well written and easy to follow. The methodology is clearly described, and the proposed method appears to be reproducible based on the descriptions provided in the paper.

2. The paper presents solid theoretical analysis, effectively leveraging and connecting prior work in the literature. The theoretical derivations supporting the proposed method appear well motivated and technically sound.

3. The empirical evaluation is conducted across multiple data modalities, including tabular, image, and text datasets (five tabular, two image, and two text datasets). The results demonstrate that the proposed method performs consistently well across diverse domains.

**Weaknesses**
1. The level of conceptual novelty appears somewhat limited. Individualized calibration [1] has been explored in prior work, and related studies have also considered distribution-based and CDF-based formulations for calibration [2].

2. While the paper argues that the homoscedasticity assumption of softmax-based classifiers leads to calibration issues, it does not provide sufficient analysis to justify that this is a primary cause of miscalibration.

3. The experimental evaluation lacks comparisons with some recent related works, which makes it somewhat difficult to assess the competitiveness of the proposed method with respect to the current state of the art.

* Related to Weakness 2, it is possible that recent calibration methods already mitigate the homoscedasticity issue. A more in-depth analysis of the role of homoscedasticity in calibration would help clarify this point.

* Recent post-hoc [3,4,5] and pre-hoc calibration methods [6, 7] for softmax-based classifiers.

[1] Zhao, S., Ma, T., and Ermon, S. Individual calibration with randomized forecasting. In International Conference on Machine Learning, pp. 11387–11397. PMLR, 2020.

[2] Song, H., Diethe, T., Kull, M., & Flach, P. (2019, May). Distribution calibration for regression. In International Conference on Machine Learning (pp. 5897-5906). PMLR.

[3] Jung, S., Seo, S., Jeong, Y., & Choi, J. (2023, July). Scaling of class-wise training losses for post-hoc calibration. In International Conference on Machine Learning (pp. 15421-15434). PMLR.

[4] Cho, G. and Youn, C.-H. Tilt and average: geometric adjustment of the last layer for recalibration. In International Conference on Machine Learning, 2024.

[5] Tao, L., Dong, M., and Xu, C. Feature clipping for uncertainty calibration. In AAAI Conference on Artificial Intelligence, pp. 20841–20849, 2025.

[6] Tao, L., Dong, M., & Xu, C. (2023, July). Dual focal loss for calibration. In International Conference on Machine Learning (pp. 33833-33849). PMLR.

[7] Lin, J., Tao, L., Dong, M., and Xu, C. (2025). Uncertainty weighted gradients for model calibration. In Proceedings of the IEEE/CVF Conference on Computer Vision and Pattern Recognition (pp. 15497-15507).

**Suggestion**

Some references may need to be updated and should be checked. Although I did not conduct a comprehensive search, I noticed that some papers that have already been accepted at conferences are still cited as arXiv papers. An example is as follows:

* Jang, E., Gu, S., and Poole, B. Categorical reparameterization with gumbel-softmax. ~~arXiv preprint arXiv:1611.01144, 2016.~~ $\rightarrow$ International Conference on Learning Representations. 2017.

---

> ### Author Rebuttal · Authors · 2026-03-30
>
> Thank you for your thoughtful and constructive feedback. We sincerely appreciate your time and careful reading of our submission.
>
> **R1 [W1; Q1]:** Our method differs from [2] in two main aspects.
>
> **1. Different calibration mechanisms.** Both our method and [2] can calibrate the predictive distribution produced by an already-trained model $M$ at the distribution level, but in different ways. Specifically, [2] takes an input feature vector $x$ and outputs a predictive distribution for $Y$ (e.g., Gaussian parameters $\mu$ and $\sigma$). It then uses these predicted parameters together with the observed target $y$ to learn a transformation (GP-BETA) that calibrates the original predictive distribution. Regression is then performed using summaries of the calibrated distribution, such as the mean, mode, median, or quantiles. In principle, [2] could also be extended to classification, although doing so in the multiclass setting would require careful modification.
>
> In contrast, our method takes as input both the feature vector $x$ and a random variable $q \sim U(0,1)$, and outputs a predictive distribution for $Y$ (e.g., through the three ALD parameters \{$\kappa, \sigma, \theta$\} = $M_\Theta(x,q)$). In the classification setting, we directly penalize the deviation between the top-label confidence $\hat{p}_{y^*}$ and the target confidence level $q$, thereby achieving individualized calibration at the top-label confidence level, with the theoretical implication chain $\mathrm{mPAIC} \Rightarrow \mathrm{PAIC} \Rightarrow \mathrm{PACC}$.
>
> Our framework can also be extended to distribution-level calibration in regression. Given $x$ and $q \sim U(0,1)$, the model outputs a predictive distribution for $Y$, and one can directly penalize
> $$
> \frac{1}{N}\sum_{i=1}^{N} | F_{\rm ALD} (y_i; M_{\Theta}(x_i,q_i))- q_i |.
> $$
> This yields a calibrated predictive distribution over $Y$ that can be interpreted as a mixture
> $$
> \int dq \ p(q) f_{\rm ALD} (y; M_{\Theta}(x,q)), p(q)=U(0,1),
> $$
> where the feature extractor captures the overall distribution shape, while the $q$-branch acts as an adapter for local calibration adjustment. Here, $f_{\rm ALD}$ and $F_{\rm ALD}$ denote the ALD's PDF and CDF. In this regression setting, the theoretical implication becomes $\mathrm{mPAIC} \Rightarrow \mathrm{PAIC}$. Regression can then again be performed using summaries of the calibrated distribution as in [2].
>
> **2. Greater flexibility in training and deployment.** Our approach can be trained end-to-end for optimal feature-calibration alignment (*pre-calibration*), or attached as a lightweight adapter to correct an already-trained model (*post-calibration*). By contrast, [2] is only applicable in the post-calibration setting.
>
> **R2 [W2; Q2]:** We added a case study showing that the homoscedasticity assumption itself contributes to miscalibration. Compared with $\mathbf{H}\_{\rm Softmax}$, $\mathbf{H}\_{\rm Gumbel}$ keeps the same overall classification setting but relaxes the fixed-scale assumption in Eq. (1): $\mathbf{H}\_{\rm Softmax}$ fixes $\beta=1$, while $\mathbf{H}_{\rm Gumbel}$ learns a sample-dependent $\beta(x)$.
>
> As shown in Table 2 [https://anonymous.4open.science/r/ICML-Re/Exp2.png ], $\mathbf{H}\_{\mathrm{Gumbel}}$ consistently improves calibration-related metrics over $\mathbf{H}\_{\rm Softmax}$. Fig.2 [https://anonymous.4open.science/r/ICML-Re/Exp2.png ] further shows that the confidence-ECE from the reliability diagrams decreases on all five datasets. Fig.3 [https://anonymous.4open.science/r/ICML-Re/Exp2.png ] also shows substantial sample-wise variation in the learned $\log(\beta)$ away from the homoscedastic baseline $\log(\beta)=0$, indicating that the optimal scale is not constant across samples. We will include this case study in the final version.
>
> **R3 [W3; Q3]:** We added 8 additional baselines and 3 additional evaluation metrics. The updated overall results are reported in Table 1 [https://anonymous.4open.science/r/ICML-Re/Exp1.png ], and the corresponding reliability-diagram comparisons are shown in Fig.1 [https://anonymous.4open.science/r/ICML-Re/Exp1.png ]. These additional experiments further strengthen our empirical evaluation. At the same time, we acknowledge that it is difficult to compare exhaustively against every possible baseline and will note this limitation in the Weakness section of the final version.
>
> Moreover, Table 3 [https://anonymous.4open.science/r/ICML-Re/Exp3.png ] compares [3,4,5] on conventional softmax-based classifiers $\mathbf{H}\_{\rm Softmax+}$ and distribution-based ALD classifiers $\mathbf{H}\_{\rm ALD+}$ across five tabular datasets. These results show that existing post-hoc calibration methods transfer well to distribution-based classifiers, and that the ALD-based post-hoc variants are overall slightly better, or at least comparable, across most metrics and datasets.
>
> **R4 [Suggestion]:** We will update the references and replace arXiv citations with conference versions where available.

---

> > ### Author Rebuttal · Reviewer_tVsS · 2026-04-04
> >
> > I appreciate the authors’ responses and thank them for the helpful clarifications.
> >
> > **R1**
> > The authors clearly distinguish their method from [2] in terms of both mechanism and flexibility.
> >
> > **R2**
> > Through extensive experiments, the authors provide convincing evidence that the homoscedasticity assumption can contribute to miscalibration.
> >
> > **R3** The addition of baselines, evaluation metrics, and reliability diagrams substantially improves the rigor and completeness of the empirical evaluation.
> >
> > Overall, the rebuttal effectively addresses my major concerns. The remaining issues are minor and should be addressed in the camera-ready version along with the rebuttal clarifications.
> >
> > Based on the rebuttal, I will raise my score to **Accept**.

---

> > > ### Author Response · Authors · 2026-04-04
> > >
> > > We sincerely thank you for your careful reconsideration of our paper and greatly appreciate your decision to raise the score after considering our rebuttal. We are encouraged that our responses and additional experiments helped address your major concerns. In the final version, we will incorporate all relevant rebuttal clarifications and carefully address the remaining minor issues. Thank you again for your constructive feedback and support.

---

### Decision · Program_Chairs · 2026-04-30

**Decision:**

Accept (regular)

**Comment:**

The paper addresses the problem of calibrating softmax based classifiers.

Some of the core strengths highlighted by the reviewers are:
- the paper is well-written and easy to follow
- extensive empirical evaluation across nine datasets
- theoretical derivations are well supported and technically sound
- the paper provides a valuable contribution to model calibration literature
- the motivation is interesting for studying softmax based classifiers through random utility theory

Among important concerns mentioned by the reviewers are:
- the novelty seems to be somewhat limited given individualized calibration and CDF formulation works
- experiments lack evaluation with recent baselines
- hard to read at times and the intuition needs further improvement
- paper's position in literature w.r.t, broad set of techniques on softmax based classifier calibration is weak
- limited discussion on scalability to large datasets and metrics other than ECE

In the post-rebuttal phase, all reviewers acknowledged that their concerns have been resolved satisfactorily and there are no left over concerns. The final ratings for the paper are two accepts and two weak accepts. The authors did a good job on providing a comprehensive rebuttal that satisfied all major concerns adequately. Therefore , the AC choses to recommend acceptance.